# Mitigating Object Hallucination in MLLMs via Data-augmented Phrase-level Alignment

**Pritam Sarkar**♥♦,∗, **Sayna Ebrahimi**♣, **Ali Etemad**♥,†, **Ahmad Beirami**♣,
**Sercan Ö. Arık**♠, **Tomas Pfister**♠
♥Queen's University, ♦Vector Institute, ♣Google DeepMind, ♠Google Cloud AI Research
{pritam.sarkar,ali.etemad}@queensu.ca
{saynae,beirami,soarik,tpfister}@google.com
https://github.com/pritamqu/HALVA

## Abstract

Despite their significant advancements, Multimodal Large Language Models (MLLMs) often generate factually inaccurate information, referred to as hallucination. In this work, we address object hallucinations in MLLMs, where information is generated about an object not present in the input image. We introduce Data-augmented Phrase-level Alignment (DPA), a novel loss which can be applied to instruction-tuned off-the-shelf MLLMs to mitigate hallucinations, while preserving their general vision-language capabilities. To fine-tune MLLMs with DPA, we first generate a set of 'hallucinated' and 'correct' response pairs through generative data augmentation by selectively altering the ground-truth information of the correct responses at a phrase level. The DPA loss is then used to train MLLMs to reduce the likelihood of hallucinated phrases compared to the correct ones. In contrast, existing alignment techniques act at the sequence level and often lead to a sharp trade off between mitigating hallucinations and preserving model capabilities. Our thorough evaluation on various benchmarks confirms the effectiveness of DPA in mitigating hallucination while retaining the out-of-the-box performance of the MLLMs on general tasks. For instance, MLLMs finetuned with DPA, which we refer to as Hallucination Attenuated Language and Vision Assistant (HALVA), improve F1 by up to $13.4\%$ on hallucination visual question-answering and reduce the hallucination rate by up to $4.2\%$ on image description tasks.

## 1 Introduction

Recent advancements in Large Language Models (LLMs) (Chowdhery et al., 2023; Anil et al., 2023; Raffel et al., 2020; Touvron et al., 2023a;b; Team et al., 2023; Brown et al., 2020) have laid the foundation for the development of highly capable multimodal LLMs (MLLMs) (Team et al., 2023; Liu et al., 2024; 2023c; Dai et al., 2023; Li et al., 2023c; Achiam et al., 2023). MLLMs can process additional modalities such as image or video, while retaining language understanding and generation capabilities. Despite their impressive performance across a variety of tasks, the issue of *object hallucination* in MLLMs presents a significant challenge to their widespread and reliable use (Wang et al., 2023c; Hu et al., 2023; Rohrbach et al., 2018; Bai et al., 2024).

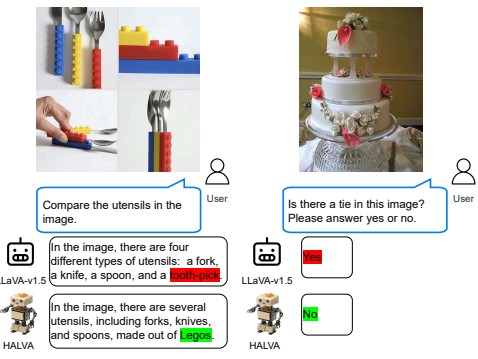

Figure 1: Examples of object hallucinations.

Object hallucination refers to generated language that includes descriptions of objects or their attributes that are not present in, or cannot be verified by, the given input. We illustrate a few examples

---

∗This work was partially done when PS was an intern at Google Cloud AI Research.
†This work was partially done when AE was a visiting faculty researcher at Google Research.

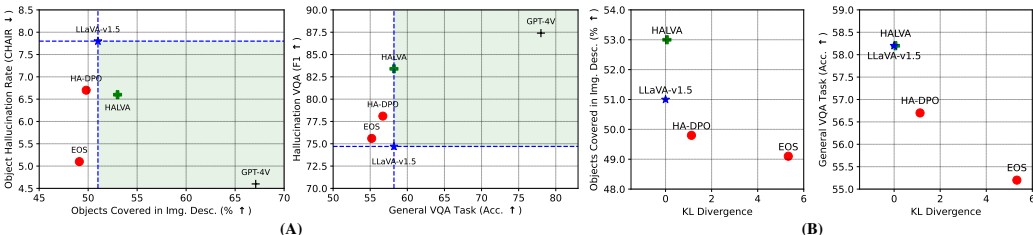

Figure 2: **(A)**: A high-level overview comparing the performance of HALVA (the finetuned model with DPA) with existing finetuning methods in mitigating object hallucination, and their ability on general vision-language tasks. **(B)**: Unlike HALVA, the existing finetuning approaches (e.g., HA-DPO and EOS) substantially diverge from their base model (LLaVA-v1.5$_{7B}$).

of object hallucinations in Figure 1, where on the left LLaVA-v1.5$_{13B}$ inaccurately describes a 'tooth-pick' in an image of utensils (knife, spoon, fork) as these items frequently appear together, while it missed identifying 'Legos' due to their rare occurrence with utensils. On the right, LLaVA-v1.5$_{13B}$ incorrectly confirms the presence of a 'tie' for the image of a 'wedding cake'. This is likely due to two reasons: first, the frequent co-occurrence of wedding attire such as 'ties' and 'wedding cakes', and second, MLLMs tend to answer 'Yes' for most instructions presented due to positive instruction bias in the training data (Liu et al., 2023b; Bai et al., 2024).

Prior research has attempted to address object hallucination in one of three key stages: inference (Deng et al., 2024a; Yin et al., 2023; Leng et al., 2023; Lee et al., 2023; Zhou et al., 2023; Biten et al., 2022), pretraining (Sun et al., 2023; Jiang et al., 2023; Liu et al., 2023b), and finetuning (Zhao et al., 2023b; Yue et al., 2024). Inference-based methods aim to mitigate hallucinations during text generation, either through specialized decoding (Leng et al., 2023; Deng et al., 2024a; Zhu et al., 2024) or through iterative corrections (Lee et al., 2023; Wu et al., 2024; Zhou et al., 2023), among others. One of the key limitations of such approaches is that they can substantially increase inference time and cost, and often require modifications to the serving infrastructure (Lee et al., 2023; Bai et al., 2024). Pretraining techniques, such as negative instruction tuning or contrastive learning, have also been used to mitigate object hallucination (Liu et al., 2023b; Jiang et al., 2023). The main limitation of such approaches is that they require massive training data (>500K samples) and can not be applied to off-the-shelf MLLMs. Finally, finetuning-based approaches attempt to mitigate object hallucination through preference optimization (Zhao et al., 2023b) or human feedback (Sun et al., 2023; Yu et al., 2023a), among others (Ben-Kish et al., 2023; Yue et al., 2024).

We note that hallucinations typically occur locally and can be pinpointed to specific words or phrases, such as 'tooth-pick' in Figure 1. This is in contrast to other alignment problems such as helpfulness, where it is difficult to identify if a particular word contributes to the overall helpfulness (or lack thereof) in a response. Existing alignment methods (e.g., DPO (Rafailov et al., 2023)) do not leverage this and instead attempt to mitigate hallucinations using a sequence-level loss. Such sequence level loss provides a coarse and noisy signal, making it less effective and causing the model to degenerate from its initial state, leading to a deterioration in general vision-language capabilities (see Figure 2).

Our goal is to achieve a fine-grained mechanism to mitigate hallucinations that allows to tackle hallucinations while not hurting the general capabilities of the model without adding to inference time or requiring substantial re-training.To this end, we first use generative data augmentation (Qin et al., 2022; Zheng et al., 2024) to construct a training set of 'hallucinated' and 'correct' response pairs, by selectively altering the ground-truth phrases in the correct responses, while keeping the overall structure intact. Next, to reduce the likelihood of hallucinations, we introduce a training objective called *Data-augmented Phrase-level Alignment (DPA)*, to finetune MLLMs using the constructed correct and hallucinated response pairs. Our proposed DPA loss consists of two terms: the first term computes the relative log-probability of the hallucinated tokens compared to the correct ones, and the second term calculates the token-wise KL divergence using a frozen reference model. Accordingly, the MLLM is trained to minimize the likelihood of hallucinated tokens while keeping the divergence minimal. As a result, while DPA is effective in mitigating hallucination it closely retains the general capabilities of the base MLLM. We refer to MLLMs trained with our proposed DPA loss as *Hallucination Attenuated Language and Vision Assistant (HALVA)*. We perform rigorous evaluations on hallucination benchmarks, showing the benefits of our method in mitigating hallucination in both generative and discriminative vision-language tasks. While the primary goal of this work is to mitigate object hallucinations, we take a further step to also evaluate on

general vision-language hallucination benchmarks. The results show that DPA also provides benefits toward other forms of vision-language hallucinations that may arise due to visual illusions among others. Finally, to ensure that the proposed DPA does not adversely affect the general capabilities of MLLMs, we evaluate HALVA on popular vision-language benchmarks. Our extensive studies confirm the effectiveness of the proposed method in mitigating object hallucinations while retaining or improving the performance in general vision-language tasks.

In summary, our main contribution is DPA, a novel method to finetune MLLMs for mitigating object hallucination in vision-language tasks. Unlike existing finetuning-based hallucination mitigation methods, DPA works at a phrase-level and penalizes the tokens where hallucination occurs and not across all the tokens. Such localized and fine-grained feedback reduces object hallucination while retaining the general performance of MLLMs. We open-source the code, checkpoints, and the generated hallucinated and correct response pairs used in training, at GitHub.

## 2    METHOD: DATA-AUGMENTED PHRASE-LEVEL ALIGNMENT (DPA)

Consider an MLLM, denoted as $\pi_\theta$, trained in an auto-regressive manner to predict an output $y$ for a given vision-language instruction $x = \{x_v, x_q\}$, where $x_v$ is an image and $x_q$ is the corresponding instruction. During inference, the generated sequence $s$ of length $T_s$ is represented as $\{t_1, t_2, \ldots, t_{T_s}\}$, where each $t_i$ represents a language token. The sequence $s$ is said to contain hallucinations if the occurrence of $t_i$ is not grounded in, or cannot be verified from, the input $x$. If the data used to train $\pi_\theta$ comprises frequent appearance of certain concepts (e.g., objects, object-attribute pairs), the MLLM may generate responses based on learned spurious correlations while ignoring the given inputs (Zhou et al., 2023; Bai et al., 2024; Rohrbach et al., 2018; Li et al., 2023d). Here, we present our strategy to mitigate object hallucinations that may occur due to such co-occurrences.

**Generative data augmentation.** We discuss our strategy to construct 'hallucinated' and 'correct' response pairs through generative data augmentation. Let $y^c$ and $y^h$ be a correct and hallucinated response, respectively, to a vision-language instruction $\{x_v, x_q\}$. We design a generative data-augmentation setup to generate $y^h$ by selectively altering the ground-truth concepts in $y^c$, thus introducing hallucinated concepts that are not present in the vision input $x_v$. Note that there is no overlap between the correct and the induced hallucinated concepts. Formally, we generate $y^h$, by replacing the ground-truth set $o$ containing the true concepts in $y^c$, with the hallucinated set $o'$, where $o' \in \mathbb{O}$ and $o' \notin x_v$. Here, $\mathbb{O}$ is a set containing hallucinated concepts. We define $\mathbb{O} = \{(o_i, c_i) \mid o_i \in U \text{ and } c_i \subseteq U\}$, where $o_i$ is a concept (e.g., object, attribute, or action), $c_i$ is a subset of concepts that co-occur with $o_i$, and $U$ represents the universal set of all possible concepts of objects and object-related attributes. See an example in Figure 3.

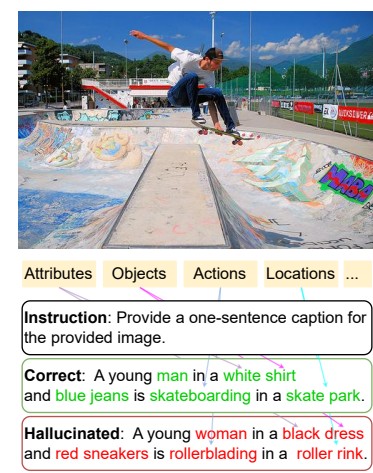

Figure 3:    An example of correct and hallucinated response pairs constructed through our generative data-augmentation. The hallucinated responses are generated by selectively altering the true concepts in the correct response. For instance, we alter 'objects': shirt → dress, & jeans → sneakers; 'attributes': white → black, & blue → red; 'actions': skateboarding → rollerblading; and other object-related information such as 'location': skate park → roller rink. Best viewed in color.

We approximate $\mathbb{O}$ for hallucinated concepts that are both closed set ($\mathbb{O}_{cc}$) and open-set ($\mathbb{O}_{oc}$). We prepare $\mathbb{O}_{cc}$ based on the co-occurring concepts in a large object-centric dataset. For $\mathbb{O}_{oc}$ we sample hallucinated concepts by directly prompting an LLM. In addition to generating descriptive responses, we also use a small set of Yes-or-No questions based on an existing visual question-answering dataset, for which we generate $y^h$ by simply inverting $y^c$. This yields the correct and hallucinated response pairs $\{y^c, y^h\}$, which we subsequently use in DPA. Additional details of generative data augmentation, including the templates for generating correct and hallucinated responses, as well as end-to-end examples of the entire augmentation process, are presented in Appendix D.3.

**Proposed phrase-level loss.** Given an off-the-shelf trained MLLM susceptible to hallucinations, our objective is to minimize the likelihood of generating hallucinated tokens using the correct and

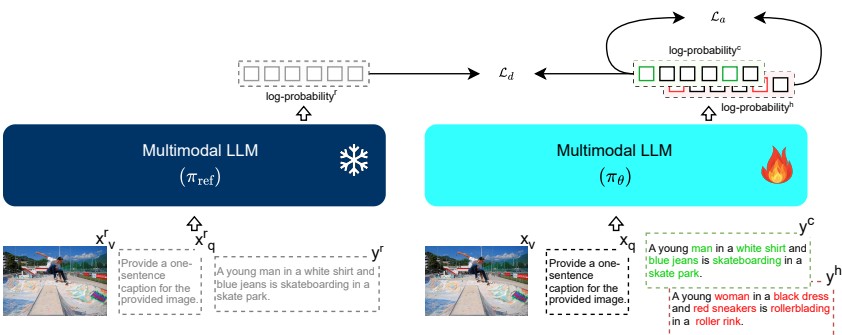

Figure 4: **Overview of our method:** Given a vision-language instruction and its correct and hallucinated response pair, the alignment objective ($\mathcal{L}_a$) reduces the log-likelihood of hallucinated tokens compared to the correct ones. Also, a token-wise KL divergence regularizer ($\mathcal{L}_d$) is employed using a reference model ($\pi_{\text{ref}}$), to restrict the divergence of the MLLM ($\pi_\theta$) during DPA training.

hallucinated response pairs $\{y^c, y^h\}$ obtained through generative data-augmentation. To this end, we define an alignment objective based on the relative probabilities of correct and hallucinated phrases.

Let's take an example with a correct response $y^c$ as 'A young man in a white shirt' and its corresponding hallucinated response $y^h$ as 'A young woman in a black dress'. Let $y_i^h$ denote the $i$-th hallucinated phrase in $y^h$ and $y_i^c$ be the corresponding correct phrase in $y^c$. In this example, the hallucinated phrases are 'woman' and 'black dress', while their corresponding correct phrases are 'man' and 'white shirt'. $y^h$ can be expressed as a sequence of tokens $T_h = \{t_1^h, t_2^h, \ldots, t_{|T_h|}^h\}$, according to which $y_i^h = T_h[s_i^h : e_i^h]$, where $s_i^h$ and $e_i^h$ are the start and end indices of $y_i^h$ with $1 \leq s_i^h \leq e_i^h \leq |T_h|$. Accordingly, we can compute the probability of hallucinated phrase $y_i^h$ as $\prod_{j=s_i^h}^{e_i^h} \pi_\theta(t_j^h | x, t_{<j}^h)$.

Similarly, the probability of the correct phrase $y_i^c$ can be expressed as: $\prod_{j=s_i^c}^{e_i^c} \pi_\theta(t_j^c | x, t_{<j}^c)$, where $s_i^c$ and $e_i^c$ are the start and end indices of $y_i^c$. Note that for every $y_i^h \in y^h$ there exists a corresponding $y_i^c \in y^c$. To reduce the relative likelihood of hallucinated phrases compared to the correct ones, we define the alignment loss $\mathcal{L}_a$ as:

$$\mathcal{L}_a = \frac{1}{N} \sum_{i=1}^{N} -\log \frac{\prod_{j=s_i^c}^{e_i^c} \pi_\theta(t_j^c | x, t_{<j}^c)}{\prod_{j=s_i^c}^{e_i^c} \pi_\theta(t_j^c | x, t_{<j}^c) + \prod_{j=s_i^h}^{e_i^h} \pi_\theta(t_j^h | x, t_{<j}^h)}, \tag{1}$$

where $N$ represents the total number of hallucinated phrases in $y^h$. Note that our loss is designed to penalize the model $\pi_\theta$ only for the hallucinated tokens rather than for all tokens in the sequence. This localized and fine-grained feedback is one of the key concepts that sets our method apart from existing preference optimization techniques, e.g., (Christiano et al., 2017; Rafailov et al., 2023).

Note that simply optimizing $\pi_\theta$ to minimize $\mathcal{L}_a$ may cause $\pi_\theta$ to substantially diverge from its initial state, which may hurt its ability in general vision-language tasks. To mitigate this effect, we train $\pi_\theta$ with a KL-divergence constraint using a frozen reference model $\pi_{\text{ref}}$. For a given reference sample $\{x^r, y^r\}$, $y^r$ can be expressed as a sequence of tokens $T_r = \{t_1^r, t_2^r, \ldots, t_{|T_r|}^r\}$. We formulate the token-wise KL-divergence regularization term $\mathcal{L}_d$ as:

$$\mathcal{L}_d = \sum_{j=1}^{|T_r|} \pi_{\text{ref}}(t_j^r | x^r, t_{<j}^r) \cdot \left( \log\left( \pi_{\text{ref}}(t_j^r | x^r, t_{<j}^r) \right) - \log\left( \pi_\theta(t_j^r | x^r, t_{<j}^r) \right) \right). \tag{2}$$

Our formulation of $\mathcal{L}_d$ serves as a token-level regularizer to restrict the model from diverging too far from its initial state, thus losing its general initial abilities. Note that $\{x^r, y^r\}$ represent any set of vision-language instructions and their correct responses, which may or may not include $\{x^c, y^c\}$. Moreover, note that $\pi_{\text{ref}}$ and $\pi_\theta$ are initialized from the same checkpoint, therefore $\mathcal{L}_d$ estimates the divergence of $\pi_\theta$ from its initial state during training. It should be noted that we adopt a forward KL-divergence approach in calculating $\mathcal{L}_d$ which is different from the reverse KL-divergence used in RLHF (Christiano et al., 2017). This choice is essential in our case, as we do not conduct rollouts of $\pi_\theta$ during training and rely solely on responses from $\pi_{\text{ref}}$, ensuring that $\pi_\theta$ focuses on high-probability tokens of the reference distribution. Finally, we train $\pi_\theta$ to minimize the final DPA objective:

$$\mathcal{L}_{dpa} = \mathcal{L}_a + \alpha \cdot \mathcal{L}_d, \tag{3}$$

where $\alpha$ is a coefficient to control the divergence of $\pi_\theta$ during training. The value of $\alpha$ is set based on ablation studies presented in Section 4.4. We present the pseudo code in Appendix A.

# 3 EXPERIMENT SETUP

**Training data.** We prepare vision-language instructions based on Visual Genome (VG) (Krishna et al., 2017), which is an object-centric image dataset consisting of a total of 108K images and their annotations. Accordingly, we prepare the correct responses with both descriptive (e.g., `Describe the image in detail.`) and non-descriptive (e.g., `<Question>, Please answer in one word, yes or no`) instructions. Descriptive instructions include one-sentence captions, short descriptions, and detailed descriptions of images. Moreover, the non-descriptive question-answers are directly taken from (Zhao et al., 2023b). We prepare the correct responses using Gemini Vision Pro (Team et al., 2023) and based on the original images and ground-truth annotations. Subsequently, we perform generative data augmentation to obtain hallucinated responses, as described in Section 2. Our final training set consists of a total of 21.5K vision-language instructions and their corresponding correct and hallucinated responses.

**Implementation details.** We use LLaVA-v1.5 (Liu et al., 2023c) and VILA-v1.5 (Lin et al., 2024) as our base models considering their superior performance in general vision-language tasks and the availability of their code and models. LLaVA-v1.5 uses Vicuna-v1.5 (Chiang et al., 2023; Touvron et al., 2023b) as the language encoder and CLIP ViT-$L_{14}$ (Radford et al., 2021) as the vision encoder. VILA-v1.5 uses Vicuna-v1.5 (Chiang et al., 2023; Touvron et al., 2023b) as the language encoder and SigLip-L-400M (Zhai et al., 2023) as the vision encoder. Note that while LLaVA-v1.5 uses images of resolution 336 pixels, VILA-v1.5 is trained with images of resolution 384 pixels. During training, we freeze the vision encoder and projection layers, and only train the LLM using LoRA (Hu et al., 2021). We refer to the resulting DPA trained checkpoints as HALVA, i.e., HALVA$_{7B}$ based on LLaVA-v1.5$_{13B}$, HALVA$_{13B}$ based on LLaVA-v1.5$_{13B}$, and HALVA$_{13B/384}$ based on VILA-v1.5$_{13B/384}$. All experiments are conducted on 4 A100-80GB GPUs. We utilize an effective batch size of 64 and train for 1 epoch or 342 steps. The training time ranges from 1.5 to 3 hours for 7B and 13B variants. The additional implementation details are presented in Appendix D.

**Evaluation setup.** First, we evaluate HALVA on four object hallucination benchmarks encompassing both generative and discriminative tasks, including **CHAIR** (Rohrbach et al., 2018), **MME-Hall** (Fu et al., 2023), **AMBER** (Wang et al., 2023b), and **MMHal-Bench** (Sun et al., 2023). Additionally, we perform a curiosity driven experiment to critically test the impact of our proposed DPA beyond object hallucination, using **HallusionBench** (Liu et al., 2023a). Furthermore, to ensure that DPA does not adversely affect the general language generation capabilities of MLLMs, we evaluate HALVA on five popular vision-language benchmarks: **VQA-v2** (Goyal et al., 2017), **MM-Vet** (Yu et al., 2023b), **TextVQA** (Singh et al., 2019), **MME** (Fu et al., 2023) and **LLaVA-Bench** (Liu et al., 2024). All evaluations are conducted thrice, and we report average scores. In the case of GPT-4-based evaluation, the performance slightly varies due to the randomness, where we also report the standard deviations.

# 4 RESULTS

Earlier in Figure 2, we present a high-level overview of HALVA vs. existing finetuning approaches (e.g., HA-DPO and EOS) in mitigating object hallucinations and their effect on the general vision-language capabilities. Note that both HA-DPO and EOS are based on the same LLaVA-v1.5$_{7B}$ as HALVA, ensuring a fair comparison. We consider LLaVA-v1.5$_{7B}$ as the lower bound and GPT-4V as strong reference point given its performance on the standard benchmarks.

**Image description task.** In Figure 2 (A) Left, we compare MLLMs on image description tasks in terms of both hallucination rate (AMBER CHAIR) and their detailedness, captured through the number of ground-truth objects covered (AMBER Cover). Our goal is to mitigate hallucinations while retaining or improving the richness of image descriptions compared to the base model. As shown, HALVA captures more ground-truth objects while hallucinating less than HA-DPO. Moreover, while EOS achieves a lower hallucination rate, it degrades the detailedness of image descriptions, performing worse than the base model. This is an undesired artifact in MLLMs, particularly for tasks that require detailedness such as medical imaging analysis (Wang et al., 2023c; Hu et al., 2023).

Table 1: Results on **CHAIR**. ‡ and † indicate that the reported values are from (Chen et al., 2023a) and (Yue et al., 2024). *Results are computed by us, using their official checkpoints. $C_i$ and $C_s$ refer to CHAIR at instance and sentence levels.

| Method | $C_i(\downarrow)$ | $C_s(\downarrow)$ | Len. |
|---|---|---|---|
| mPLUG-Owl‡7B (Ye et al., 2023a) | 30.2 | 76.8 | 98.5 |
| MultiModal-GPT‡7B (Gong et al., 2023) | 18.2 | 36.2 | 45.7 |
| MiniGPT-v2‡7B (Chen et al., 2023a) | 8.7 | 25.3 | 56.5 |
| InstructBlip7B (Dai et al., 2023) | 17.5 | 62.9 | 102.9 |
| LLaVA-v1.5†7B (Liu et al., 2023c) | 15.4 | 50.0 | 100.6 |
| EOS7B (Yue et al., 2024) | 12.3 | 40.2 | 79.7 |
| OPERA7B (Huang et al., 2023) | 12.8 | 44.6 | - |
| DoLA7B (Chuang et al., 2023) | 13.8 | 47.8 | - |
| HA-DPO*7B (Zhao et al., 2023b) | 11.0 | 38.2 | 91.0 |
| MEMVR7B (Zou et al., 2024) | 13.0 | 46.6 | 99.6 |
| AGLA7B (An et al., 2024) | 14.1 | 43.0 | 98.8 |
| **HALVA7B (Ours)** | 11.7$_{\downarrow 3.7}$ | 41.4$_{\downarrow 8.6}$ | 92.2 |
| MiniGPT-4†13B (Zhu et al., 2023) | 9.2 | 31.5 | 116.2 |
| InstructBlip13B (Dai et al., 2023) | 16.0 | 51.2 | 95.6 |
| LLaVA‡13B (Liu et al., 2024) | 18.8 | 62.7 | 90.7 |
| LLaVA-v1.5†13B (Liu et al., 2023c) | 13.0 | 47.2 | 100.9 |
| EOS13B (Yue et al., 2024) | 11.4 | 36.8 | 85.1 |
| **HALVA13B (Ours)** | 12.8$_{\downarrow 0.2}$ | 45.4$_{\downarrow 1.8}$ | 98.0 |
| VILA-v1.513B/384 (Lin et al., 2024) | 9.2 | 33.0 | 183.4 |
| **HALVA13B/384 (Ours)** | 8.4$_{\downarrow 0.8}$ | 30.0$_{\downarrow 3.0}$ | 182.6 |

Table 2: Results on **MME-Hall**. ‡ indicating reported values from (Bai et al., 2024). *Results are computed by us, using official checkpoints. Red: worse than base model.

| Method | MME-Hall ($\uparrow$) |
|---|---|
| Cheetor7B‡ (Li et al., 2023b) | 473.4 |
| LRV-Instruction7B‡ (Liu et al., 2023b) | 528.4 |
| Otter7B‡ (Li et al., 2023a) | 483.3 |
| mPLUG-Owl27B‡ (Ye et al., 2023b) | 578.3 |
| Lynx7B‡ (Zeng et al., 2023) | 606.7 |
| Qwen-VL-Chat7B‡ (Bai et al., 2023) | 606.6 |
| LLaMA-Adapter V27B‡ (Gao et al., 2023) | 493.3 |
| LLaVA-v1.57B (Liu et al., 2023c) | 648.3 |
| HA-DPO*7B (Zhao et al., 2023b) | 618.3 |
| EOS*7B (Yue et al., 2024) | 606.7 |
| VCD7B (Leng et al., 2023) | 604.7 |
| Woodpecker*7B (Yin et al., 2023) | 366.7 |
| MEMVR7B (Zou et al., 2024) | 648.3 |
| ARA7B (Qu et al., 2024) | 648.3 |
| AGLA7B (An et al., 2024) | 640.0 |
| **HALVA7B (Ours)** | 665.0$_{\uparrow 16.7}$ |
| BLIVA11B‡ (Hu et al., 2024) | 580.0 |
| MMICL12B‡ (Zhao et al., 2023a) | 568.4 |
| InstructBLIP13B‡ (Dai et al., 2023) | 548.3 |
| SPHINX13B‡ (Lin et al., 2023) | 668.3 |
| Muffin13B‡ (Lou et al., 2023) | 590.0 |
| RLHF-V13B (Yu et al., 2023a) | 585.0 |
| LLaVA-v1.513B (Liu et al., 2023c) | 643.3 |
| **HALVA13B (Ours)** | 675.0$_{\uparrow 31.7}$ |
| VILA-v1.513B/384 (Lin et al., 2024) | 688.3 |
| **HALVA13B/384 (Ours)** | 691.7$_{\uparrow 3.4}$ |

**Question answering task.** In Figure 2 (A) Right, we compare the performance of MLLMs on visual question-answering tasks using both object hallucination (AMBER) and general vision-language (TextVQA) benchmarks. As shown, both HA-DPO and EOS underperform HALVA in mitigating object hallucination and even deteriorate general vision-language abilities compared to the base model. These results show the shortcomings of existing approaches, which we address in this work.

To further understand the limitations of existing methods in greater detail, we measure divergence from the base model in Figure 2 (B). Here we observe that unlike HALVA, both HA-DPO and EOS substantially diverge from the base model, resulting in poor performance in general tasks.

### 4.1 EVALUATION ON OBJECT HALLUCINATION

**CHAIR.** MLLMs can be prone to hallucinations when generating detailed image descriptions (Bai et al., 2024; Rohrbach et al., 2018; Wang et al., 2023b). To assess the impact of DPA in such scenarios, we evaluate HALVA on CHAIR, which stands for Caption Hallucination Assessment with Image Relevance (Rohrbach et al., 2018). This metric calculates the number of objects that appear in the image caption but are not present in the image. Specifically, CHAIR measures hallucination at two levels: instance-level ($C_i$) and sentence-level ($C_s$). During this task, HALVA is prompted with 'Describe the image in detail', allowing for the generation of detailed image descriptions. The results in Table 1 demonstrate that HALVA substantially reduces hallucination in image descriptions compared to the base variants. For instance, compared to LLaVA-v1.57B, HALVA7B reduces $C_s$ from 50.0 to 41.4, similarly, compared to VILA-v1.513B/384, HALVA13B/384 reduces $C_s$ from 33.0 to 30.0. Furthermore, HALVA7B outperforms or matches the performance of other hallucination mitigation methods, such as OPERA (Huang et al., 2023), EOS (Yue et al., 2024), and HA-DPO (Zhao et al., 2023b). It should be noted that our proposed DPA does not negatively impact the language generation ability or expressiveness of MLLMs, unlike EOS (Yue et al., 2024), which substantially reduces the average generation length from 100 to 85 and 79 for the 13B and 7B variants, respectively. As discussed earlier in Section 4, such a degree of reduction can lead to missing key details in image descriptions and are undesirable for MLLMs. In contrast, HALVA maintains the same generation length as the base model, e.g., 98 vs. 100.9 or 182.6 vs. 183.4, while effectively reducing hallucination. However, a limitation of CHAIR (Rohrbach et al., 2018) is that it does not consider other key aspects of image descriptions, such as coverage of objects and detailedness of

descriptions, when evaluating hallucination. Therefore, we also evaluate on AMBER (Wang et al., 2023b), a more recent object hallucination benchmark, which we discuss later.

**MME-Hall.** We evaluate HALVA on discriminative tasks using MME (Fu et al., 2023). Specifically, we utilize the hallucination subset of MME, which consists of four object-related subtasks: existence, count, position, and color, referred to as MME-Hall. The full score of each category is 200, making the maximum total score 800. The results presented in Table 2 demonstrate that HALVA substantially improves performance compared to the base model. For instance, $\text{HALVA}_{13B}$ achieves a score of 675.0, resulting in a performance gain of 31.7 points with respect to the base model $\text{LLaVA-v1.5}_{13B}$. Moreover, as presented in Table 2, existing methods including finetuning (e.g., HA-DPO, EOS) and inference-based (e.g., VCD, Woodpecker) approaches are ineffective in mitigating hallucinations across such broad categories and worsen the performance compared to their base model. The detailed results of MME-Hall are presented in Appendix C.

**AMBER.** To evaluate performance on both generative and discriminative tasks, we use AMBER (Wang et al., 2023b), which measures hallucination using several metrics. For generative tasks, AMBER assesses the frequency of hallucinated objects in image descriptions, similar to (Rohrbach et al., 2018). Moreover, AMBER evaluates hallucination in three additional aspects of generative abilities: the number of ground-truth objects covered in the description, the hallucination rate, and the similarity of hallucinations in MLLMs to those observed in human cognition. Discriminative tasks are categorized into three broad groups: existence, attribute, and relation, each assessed using F1 scores. For additional details on these evaluation metrics, we refer the reader to (Wang et al., 2023b).

Table 3: Results on **AMBER**. Cover.: coverage of ground-truth objects; Hall.: Hallucination Rate; $\ddagger$ indicates that the reported values are from (Wang et al., 2023b). *Results are computed by us, using their checkpoint. Red: worse than base model.

| Method | Generative | | | Discriminative |
|---|---|---|---|---|
| | CHAIR ($\downarrow$) | Cover. ($\uparrow$) | Hall. ($\downarrow$) | (F1$\uparrow$) |
| mPLUG-Owl$\ddagger_{7B}$ (Ye et al., 2023a) | 21.6 | 50.1 | 76.1 | 18.9 |
| LLaVA$\ddagger_{7B}$ (Liu et al., 2024) | 11.5 | 51.0 | 48.8 | 32.7 |
| MiniGPT-4$\ddagger_{7B}$ (Zhu et al., 2023) | 13.6 | 63.0 | 65.3 | 64.7 |
| mPLUG-Owl2$\ddagger_{7B}$ (Ye et al., 2023b) | 10.6 | 52.0 | 39.9 | 78.5 |
| InstructBLIP$\ddagger_{7B}$ (Dai et al., 2023) | 8.8 | 52.2 | 38.2 | 81.7 |
| LLaVA-v1.5$\ddagger_{7B}$ | 7.8 | 51.0 | 36.4 | 74.7 |
| HA-DPO$^*_{7B}$ (Zhao et al., 2023b) | 6.7 | 49.8 | 30.9 | 78.1 |
| EOS$^*_{7B}$ (Yue et al., 2024) | 5.1 | 49.1 | 22.7 | 75.6 |
| Woodpecker$^*_{7B}$ (Yin et al., 2023) | 6.9 | 48.9 | 30.4 | 67.0 |
| **HALVA$_{7B}$ (Ours)** | $6.6_{\downarrow 1.2}$ | $53.0_{\uparrow 2.0}$ | $32.2_{\downarrow 4.2}$ | $83.4_{\uparrow 8.7}$ |
| RLHF-V$_{13B/448}$ Yu et al. (2023a) | 6.8 | 46.1 | 27.4 | 87.1 |
| LLaVA-v1.5$_{13B}$ (Liu et al., 2023c) | 6.6 | 51.9 | 30.5 | 73.1 |
| **HALVA$_{13B}$ (Ours)** | $6.4_{\downarrow 0.2}$ | $52.6_{\uparrow 0.7}$ | $30.4_{\downarrow 0.1}$ | $86.5_{\uparrow 13.4}$ |
| VILA-v1.5$_{13B/384}$ (Lin et al., 2024) | 9.9 | 63.3 | 56.1 | 82.2 |
| **HALVA$_{13B/384}$ (Ours)** | $9.1_{\downarrow 0.8}$ | $63.9_{\uparrow 0.6}$ | $54.2_{\downarrow 1.9}$ | $87.9_{\uparrow 5.7}$ |
| GPT-4V$\ddagger$ (Achiam et al., 2023) | 4.6 | 67.1 | 30.7 | 87.4 |

The results presented in Table 3 demonstrate that HALVA outperforms the base model by a large margin, in both generative and discriminative tasks. For instance, $\text{HALVA}_{7B}$ reduces hallucination in caption generation from 7.8 to 6.6, while increasing the coverage of ground-truth objects in the descriptions from 51% to 53%. This confirms that our method reduces hallucination without compromising the descriptive power of MLLMs. On the other hand, while HA-DPO and EOS report slightly lower hallucination rates, the number of ground-truth objects covered is reduced to 49.8% and 49.1%, respectively. This indicates a degradation in the overall performance of these MLLMs on general tasks. Similar shortcomings are also noticed when using inference-based correction methods such as Woodpecker (Yin et al., 2023), where the object coverage is reduced by 2.1% compared to the base model. Woodpecker also performs poorly on discriminative tasks as it fails to capture key concepts from short responses of LLaVA-v1.5 which it aims to correct. Moreover, our proposed DPA substantially enhances performance on discriminative tasks, for both 7B and 13B variants. For instance, $\text{HALVA}_{7B}$ improves the F1-score on the attribute category from 64.6% to 80.0%. Additionally, $\text{HALVA}_{13B}$ improves the F1 score on relation-based tasks from 45.0% to 73.5%. Overall, $\text{HALVA}_{7B}$ outperforms both HA-DPO and EOS on discriminative tasks by a large margin, achieving a 5.3% and 7.8% higher F1 score respectively. Furthermore, $\text{HALVA}_{13B}$ and $\text{HALVA}_{13B/384}$ perform better or on par with GPT-4V on discriminative tasks, i.e., F1-score of 86.5 by $\text{HALVA}_{13B}$, 87.9 by $\text{HALVA}_{13B/384}$, and 87.4 by GPT-4V. The detailed results are in Appendix C.

**MMHal-Bench.** We also conduct LLM-assisted hallucination evaluation to rigorously test for potential hallucinations in generated responses that might not be captured when validated against a limited ground-truth information, as done in (Rohrbach et al., 2018). We utilize MMHal-Bench (Sun et al., 2023), which evaluates hallucination across 12 object-topics, including object attributes, presence of adversarial objects, and spatial relations, among others. Following (Sun et al., 2023), we use GPT-4 (Achiam et al., 2023) as the judge to rate the responses on a scale of 0 to 6, with respect to standard human-generated answers and other ground-truth information of the images. The results

Table 4: Results on **MMHal-Bench**. [†], [‡], and [**] indicate that the reported values are from (Sun et al., 2023), (Jiang et al., 2023), and (Yu et al., 2024). [*]Results are computed by us, using their official checkpoint. Red: worse than base model.

| Method | Overall Score ($\uparrow$) | Hall. Rate ($\downarrow$) |
|---|---|---|
| Kosmos-2[‡] (Peng et al., 2023) | 1.69 | 0.68 |
| IDEFIC[‡]$_{9B}$ (Laurençon et al., 2024) | 1.89 | 0.64 |
| InstructBLIP[‡]$_{7B}$ (Dai et al., 2023) | 2.10 | 0.58 |
| LLaVA[‡]$_{7B}$ (Liu et al., 2024) | 1.55 | 0.76 |
| VCD[**]$_{7B}$ (Leng et al., 2023) | 2.12 | 0.54 |
| OPERA$_{7B}$ (Huang et al., 2023) | 2.33 | 0.50 |
| LURE$_{7B}$ (Zhou et al., 2023) | 1.64 | 0.60 |
| LLaVA-SFT$_{7B}$ (Sun et al., 2023) | 1.76 | 0.67 |
| LLaVA-RLHF$_{7B}$ (Sun et al., 2023) | 2.05 | 0.68 |
| LLaVA-v1.5$_{7B}$ (Liu et al., 2023c) | $2.11^{\pm0.05}$ | $0.54^{\pm0.01}$ |
| HACL$_{7B}$ (Jiang et al., 2023) | 2.13 | 0.50 |
| HA-DPO[*]$_{7B}$ (Zhao et al., 2023b) | 1.97 | 0.60 |
| EOS[*]$_{7B}$ (Yue et al., 2024) | 2.03 | 0.59 |
| **HALVA$_{7B}$ (Ours)** | $2.25^{\pm0.09}_{\uparrow0.14}$ | $0.54^{\pm0.01}_{\downarrow0.00}$ |
| LLaVA[†]$_{13B}$ (Liu et al., 2024) | 1.11 | 0.84 |
| InstructBLIP[‡]$_{13B}$ (Dai et al., 2023) | 2.14 | 0.58 |
| RLHF-V$_{13B/448}$ (Yu et al., 2023a) | - | 0.52 |
| LLaVA-SFT$_{13B}$ (Sun et al., 2023) | 2.43 | 0.55 |
| LLaVA-RLHF$_{13B}$ (Sun et al., 2023) | 2.53 | 0.57 |
| LLaVA-v1.5$_{13B}$ (Liu et al., 2023c) | $2.37^{\pm0.02}$ | $0.50^{\pm0.00}$ |
| CODE$_{13B}$ (Kim et al., 2024) | 2.49 | 0.51 |
| **HALVA$_{13B}$ (Ours)** | $2.58^{\pm0.07}_{\uparrow0.21}$ | $0.45^{\pm0.02}_{\downarrow0.05}$ |
| VILA-v1.5$_{13B/384}$ (Lin et al., 2024) | $2.58^{\pm0.02}$ | $0.46^{\pm0.01}$ |
| **HALVA$_{13B/384}$ (Ours)** | $2.58^{\pm0.06}$ | $0.45^{\pm0.01}_{\downarrow0.01}$ |
| GPT4V (Achiam et al., 2023) | 3.49 | 0.28 |

Table 5: Results on **HallusionBench**. [†] indicates that the reported values are from (Liu et al., 2023a). [*]Results are computed by us, using their official checkpoint.

| Method | Yes/No Bias ($\sim$0) | Overall Acc. ($\uparrow$) |
|---|---|---|
| mPLUG_Owl-v1[†]$_{7.2B}$ (Ye et al., 2023a) | 0.32 | 43.93 |
| MiniGPT5[†]$_{7B}$ (Zheng et al., 2023) | 0.25 | 40.30 |
| MiniGPT4[†]$_{7B}$ (Zhu et al., 2023) | 0.19 | 35.78 |
| InstructBLIP[†]$_{7B}$ (Dai et al., 2023) | -0.13 | 45.26 |
| BLIP2[†]$_{7B}$ (Li et al., 2023c) | 0.18 | 40.48 |
| mPLUG_Owl-v2[†]$_{7B}$ (Ye et al., 2023b) | 0.25 | 47.30 |
| LRV-Instruction[†]$_{7B}$ (Liu et al., 2023b) | 0.26 | 42.78 |
| LLaVA-1.5[*]$_{7B}$ (Liu et al., 2023c) | 0.31 | $47.09^{\pm0.14}$ |
| LLaVA-RLHF[*]$_{7B}$ Sun et al. (2023) | 0.24 | 42.96 |
| HA-DPO[*]$_{7B}$ (Zhao et al., 2023b) | 0.26 | 48.36 |
| EOS[*]$_{7B}$ (Yue et al., 2024) | 0.29 | 48.72 |
| **HALVA$_{7B}$ (Ours)** | $0.17_{\downarrow0.14}$ | $48.95^{\pm0.13}_{\uparrow1.86}$ |
| Qwen-VL[†]$_{9.6B}$ (Bai et al., 2023) | 0.12 | 39.15 |
| Open-Flamingo[†]$_{9B}$ (Awadalla et al., 2023) | 0.33 | 38.44 |
| BLIP2-T5[†]$_{12B}$ (Li et al., 2023c) | 0.08 | 48.09 |
| RLHF-V[*]$_{13B/448}$ (Yu et al., 2023a) | 0.13 | 47.47 |
| LLaVA-1.5[†]$_{13B}$ (Liu et al., 2023c) | 0.26 | 46.94 |
| LLaVA-1.5[*]$_{13B}$ (Liu et al., 2023c) | 0.38 | $46.50^{\pm0.09}$ |
| LLaVA-RLHF[*]$_{13B}$ (Sun et al., 2023) | 0.17 | 46.41 |
| **HALVA$_{13B}$ (Ours)** | $0.20_{\downarrow0.18}$ | $49.10^{\pm0.05}_{\uparrow2.60}$ |
| VILA-v1.5[*]$_{13B/384}$ (Lin et al., 2024) | 0.19 | $55.39^{\pm0.05}$ |
| **HALVA$_{13B/384}$ (Ours)** | $0.02_{\downarrow0.17}$ | $56.60^{\pm0.18}_{\uparrow1.21}$ |
| GPT4V[†] (Achiam et al., 2023) | 0.06 | 65.28 |
| Gemini Pro Vision[†] (Team et al., 2023) | -0.02 | 36.85 |

presented in Table 4 demonstrate that HALVA considerably improves performance with respect to LLaVA-v1.5. Furthermore, we observe that our approach is more effective in mitigating hallucination than existing RLHF, SFT, or DPO-based methods. For example, HALVA$_{7B}$ achieves a score of 2.25 surpassing the 7B variants of RLHF, DPO, and SFT -based methods, which report scores of 2.05, 1.97, and 1.76, respectively. Moreover, HALVA$_{13B}$ reduces the hallucination rate to 0.45, compared to 0.57 for LLaVA-RLHF. Note that as LLaVA-RLHF and LLaVA-SFT use the same language and vision encoders as HALVA (Vicuna-V1.5 and ViT-L/14), ensuring a fair direct comparison. The detailed results for the individual categories are presented in Appendix C.

## 4.2 EVALUATION ON HALLUCINATION BENCHMARKS BEYOND OBJECT HALLUCINATION

To further stress-test DPA on other forms of vision-language hallucinations that are not restricted to objects and may occur due to visual illusions, we evaluate performance on HallusionBench (Liu et al., 2023a). The results presented in Table 5 demonstrate that our proposed method directly benefits other forms of vision-language hallucinations as well. HALVA$_{7B}$, HALVA$_{13B}$, and HALVA$_{13B/384}$ improve the overall accuracy by 1.86%, 2.16%, and 1.21%, respectively, compared to their base models. Moreover, DPA mitigates Yes/No bias in MLLM responses. Specifically, HALVA$_{13B/384}$ reduces Yes/No bias from 0.19 to 0.02. Detailed results on HallusionBench are in Appendix C.

## 4.3 EVALUATION ON NON-HALLUCINATION BENCHMARKS

We further assess HALVA on general vision-language tasks using four popular benchmarks: VQA-v2 (Goyal et al., 2017), MM-Vet (Yu et al., 2023b), TextVQA (Singh et al., 2019), MME (Fu et al., 2023), and LLaVA-Bench-in-the-Wild (LLaVA-BW) (Liu et al., 2024). We follow the evaluation protocol mentioned in LLaVA-v1.5 (Liu et al., 2023c). The results presented in Table 6 show that HALVA maintains or improves performance with respect to the base models. For example, HALVA$_{7B}$ improves on MME, MM-Vet, and LLaVA-BW by 16.3, 1%, and 1.8% respectively, while retaining the same performance on VQA-v2. A similar trend is noticed in the case of HALVA$_{13B}$ and HALVA$_{13B/384}$. Unlike HALVA$_{7B}$, existing finetuning methods such as HA-DPO$_{7B}$ and EOS$_{7B}$, based on LLaVA-v1.5$_{7B}$, exhibit statistically significant performance drops in general tasks when tuned for hallucination mitigation. We present the details of our statistical analysis in Appendix C.11.

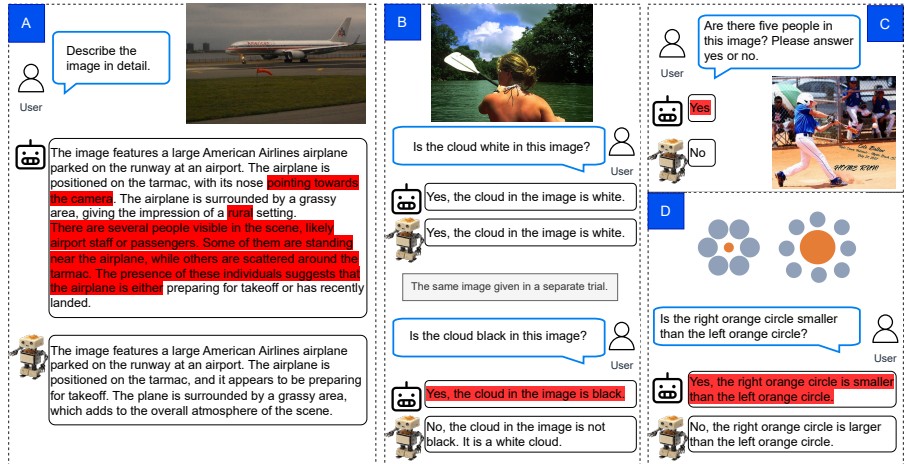

Figure 6: Qualitative comparisons between HALVA [🤖] and LLaVA-v1.5 [🖼]. Our proposed DPA effectively mitigates hallucination under different setups: (A) detail image description, (B) visual question-answering, (C) Yes-or-No answer, (D) visual illusion. Hallucinations are highlighted in red. More examples, comparing with LLaVA-v1.5 and VILA-v1.5, are in Appendix E.

Table 6: Results on **general vision-language tasks**. *Results are computed by us, using their official checkpoint. Red underline indicates that the performance drop is statistically significant.

| Method | VQA$^{v2}_{\uparrow}$ | MM-Vet$_{\uparrow}$ | TextVQA$_{\uparrow}$ | MME$_{\uparrow}$ | LLaVA-BW$_{\uparrow}$ |
|---|---|---|---|---|---|
| LLaVA-v1.5$_{7B}$ | 78.5 | 31.1 | 58.3 | 1510.7 | 65.4 |
| HA-DPO$_{7B}$ | 77.6*$_{\downarrow0.9}$ | 30.7*$_{\downarrow0.4}$ | 56.7*$_{\downarrow1.6}$ | 1502.6*$_{\downarrow8.1}$ | 66.2$_{\uparrow0.8}$ |
| EOS$_{7B}$ | 77.6*$_{\downarrow0.9}$ | 31.4*$_{\uparrow0.3}$ | 55.2*$_{\downarrow3.1}$ | 1424.4*$_{\downarrow102.6}$ | 65.8$_{\uparrow0.4}$ |
| **HALVA$_{7B}$** | 78.5 | 32.1$_{\uparrow1.0}$ | 58.2$_{\downarrow0.04}$ | 1527.0$_{\uparrow16.3}$ | 67.2$_{\uparrow1.8}$ |
| LLaVA-v1.5$_{13B}$ | 80.0 | 36.1 | 61.2 | 1530.1 | 72.5 |
| **HALVA$_{13B}$** | 80.0 | 37.8$_{\uparrow1.7}$ | 61.2 | 1544.0$_{\uparrow13.9}$ | 72.7$_{\uparrow0.2}$ |
| VILA-v1.5$_{13B}$ | 82.8 | 44.3 | 65.0 | 1569.6 | 80.8 |
| **HALVA$_{13B/384}$** | 82.8 | 44.3 | 64.8$_{\downarrow0.2}$ | 1575.7$_{\uparrow6.1}$ | 82.4$_{\uparrow1.6}$ |

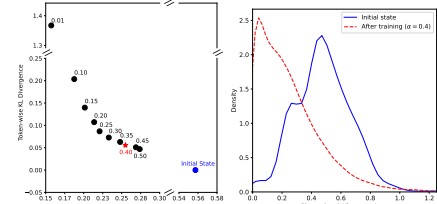

Figure 5: **Left**: Changes in the model state due to DPA training with varying $\alpha$. **Right**: Changes is alignment loss before and after training across all training samples. Default $\alpha$ is 0.4 for HALVA$_{7B}$.

## 4.4 ABLATION STUDY

Recalling the final DPA objective, which combines the alignment loss ($\mathcal{L}_a$) and KL divergence ($\mathcal{L}_d$), defined as $\mathcal{L}_{dpa} = \mathcal{L}_a + \alpha \cdot \mathcal{L}_d$, we examine the change in model state with varying $\alpha$, as depicted in Figure 5 (Left). The $y$ axis represents the extent to which the model diverges from its initial state during DPA training, while the $x$ axis shows the change in the relative log-probability of the hallucinated tokens. Each data point in this figure represents the calculated alignment loss and divergence after training for different values of $\alpha$. The figure illustrates that with a very low $\alpha$, e.g. 0.01, the model substantially diverges from its initial state. As $\alpha$ increases, the model tends to retain a state similar to the base model. We empirically find that $\alpha = 0.4$ works optimally for HALVA$_{7B}$. The change in $\mathcal{L}_a$ before and after DPA training computed over the entire training samples is presented in Figure 5 (Right). In-depth ablation studies on the proposed loss and generative data-augmentation are presented in Appendix C.

## 4.5 QUALITATIVE ANALYSIS

A qualitative comparison of HALVA to the base model is shown in Figure 6, with additional examples in Appendix E. HALVA consistently provides more accurate image descriptions than LLaVA-v1.5. For example, in Figure 6 (A), LLaVA-v1.5 hallucinates 'people', 'airport staff', 'passengers' in an image of a parked airplane. In contrast, HALVA accurately describes the image with necessary details. Additionally, our method does not exhibit LLaVA-v1.5's tendency to answer 'Yes' to most questions, which can contribute to hallucinations. This is shown in Figure 6 (B), where HALVA correctly answers 'Yes' when asked 'Is the cloud white in the image?' and responds with 'No' when asked 'Is the cloud black in this image?', whereas LLaVA-v1.5 answers 'Yes' to both cases. In another example, shown in Figure 6 (C), unlike LLaVA-v1.5, HALVA provides the correct answer to the number of people present in the image. Lastly, we present an example of hallucination caused by visual illusion in Figure 6 (D). While HALVA is not explicitly trained for such vision-language hallucinations, our approach shows some ability to mitigate it.

## 5 RELATED WORK

**Multimodal LLM (MLLM).** Vision-language models (VLMs) often align image and text features in a shared embedding space, as pioneered by CLIP (Radford et al., 2021) and ALIGN (Jia et al., 2021), and others (Yu et al., 2022; Chen et al., 2022; Li et al., 2022; Wang et al., 2022). This alignment is achieved through contrastive learning on large image-text datasets. VLMs show strong generalization across various tasks. Leveraging LLMs and vision encoders from VLMs like CLIP, recent MLLMs (Liu et al., 2024; Zhu et al., 2023; Team et al., 2023; Achiam et al., 2023; Dai et al., 2023; Li et al., 2023c; Peng et al., 2023; Hu et al., 2024; Dai et al., 2023; Bai et al., 2023; Chen et al., 2023b) further enhance visual perception, understanding, and reasoning. While some MLLMs are open-source, others are only accessible through APIs (Achiam et al., 2023; Team et al., 2023; Bai et al., 2023). Among the publicly available MLLMs, LLaVA (Liu et al., 2024; 2023c) and VILA (Lin et al., 2024) are widely used due to their simplicity and the availability of code, models, and training data. This makes them suitable base models for demonstrating applicability of DPA on off-the-shelf MLLMs.

**Alignment.** Reinforcement learning from human feedback (RLHF) (Christiano et al., 2017) aligns models by training a new model via a KL-regularized RL problem using an outcome reward. DPO (Rafailov et al., 2023) and many follow-ups (Azar et al., 2024; Pal et al., 2024; Tang et al., 2024; Amini et al., 2024) emerged as a simple alternative to RLHF that sidesteps reward modeling. Note that all these methods operate at the sequence level, which provides noisy feedback in alignment in all intermediate steps. On the other hand, recent work has focused on token-level alignment methods (Mudgal et al., 2023; Zeng et al., 2024; Rafailov et al., 2024; Chakraborty et al., 2024) with a process reward ($q$-function). In contrast, our proposed DPA is an offline alignment method designed to overcome two key limitations of existing methods: providing fine-grained feedback through phrase-level alignment and restricting divergence by applying a strong token-wise forward KL regularizer. Notably, the forward KL regularizer helps avoid the mode-seeking behavior of reverse KL-based RL fine-tuning, which may lead to low diversity in generations (Wang et al., 2023a).

**Hallucination mitigation.** Multimodal hallucination generally refers to the misrepresentation of verifiable information in relation to the given input. This phenomenon has been primarily studied in the context of object hallucination (Rohrbach et al., 2018; Bai et al., 2024; Zhou et al., 2023; Sun et al., 2023; Biten et al., 2022). Prior work to mitigate this issue can be categorized into three phases: pretraining, where techniques include using balanced instruction-tuning data with equal positive and negative examples (Liu et al., 2023b) or generating and correcting image-instruction pairs on-the-fly (Wang et al., 2024); inference, with methods involving specialized decoding strategies (Leng et al., 2023; Deng et al., 2024a; Zhu et al., 2024) or iterative corrections using offline models to detect and correct hallucinations at inference time (Zhou et al., 2023; Yin et al., 2023); and finetuning, with approaches relying on human feedback (Sun et al., 2023; Yu et al., 2023a) to train reward models or employing preference optimization techniques (Zhao et al., 2023b; Yu et al., 2023a; 2024; Pi et al., 2024; Zhou et al., 2024; Deng et al., 2024b). While finetuning methods are a more efficient direction as they do not require training from scratch (unlike pretraining-based methods) nor changes in the serving infrastructure (unlike inference-based methods), existing finetuning approaches may deteriorate the performance of the base model on general vision-language tasks (Figure 2). To address this, we introduce DPA, which is effective in mitigating object hallucination on a broad set of vision-language tasks while retaining or improving the general abilities of the base model. In contrast to (Gunjal et al., 2024) that explores training a reward model to provide sub-sequence level feedback for preference optimization training, we introduce a fine-grained objective function that can be directly used to finetune multimodal LLMs for hallucination mitigation.

## 6 CONCLUDING REMARKS

We introduce data-augmented phrase-level alignment to mitigate object hallucination in MLLMs. Our approach uses generative data augmentation to create pairs of hallucinated and correct responses by selectively altering ground-truth phrases in the correct responses. These pairs are then used to train MLLMs with our proposed DPA loss, which reduces the relative log-likelihood of hallucinated tokens compared to correct ones. Our extensive study demonstrates the effectiveness of DPA in mitigating various forms of object hallucinations, including those related to existence and attributes, as well as hallucinations arising from visual illusions or complex charts. Additionally, unlike existing fine-tuning-based solutions, DPA effectively mitigates hallucination across diverse vision-language tasks while maintaining or even enhancing performance on general vision-language tasks.

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

# Appendix

The organization of the appendix is as follows:

## A DPA PSEUDO CODE

Our proposed DPA is fairly straightforward to implement. Below, we provide a PyTorch-based pseudo code. Please note that this is a minimal implementation to present the key steps of our algorithm. Some of the intermediary and rudimentary steps (e.g., ignoring padded inputs during loss calculation) are intentionally omitted for brevity. The code will be made publicly available.

```python
import torch
import torch.nn.functional as F

def forward(self, **inputs):
    """x: vision-language input
    y_pos: correct response of x
    y_neg: hallucinated response of x constructed through gen. data aug.
    x_ref, y_ref: reference input-output pair to calculate divergence
    """

    batch_size = x.shape[0]

    # forward pass with correct and hallucinated responses
    pos_logits = self.model(x, y_pos)
    neg_logits = self.model(x, y_neg)

    # calculate log-probabilities
    pos_logps, pos_labels = self.log_softmax(pos_logits, y_pos)
    neg_logps, neg_labels = self.log_softmax(neg_logits, y_neg)

    # accumulate log-probabilities of
    # correct and hallucinated tokens at phrase level
    pos_logps = self.accumulate_logps(pos_logps)
    neg_logps = self.accumulate_logps(neg_logps)

    # phrase-level alignment loss
    alignment_loss = torch.log(1 + torch.exp(neg_logps - pos_logps))
    alignment_loss = alignment_loss.mean()

    # forward pass with the reference samples
    logits = self.model(x_ref, y_ref)
    with torch.no_grad():
        reference_logits = self.reference_model(x_ref, y_ref)

    # calculate probability
    proba = F.softmax(logits, dim=-1)
    reference_proba = F.softmax(reference_logits, dim=-1)

    # token-wise KL divergence
    divergence = (reference_proba*(reference_proba.log()-proba.log()))
    divergence = divergence.sum()/batch_size

    # final loss
    loss = alignment_loss + self.alpha*divergence
```

```
return loss
```

# B DISTINCTION BETWEEN OURS DPA AND DPO-BASED HALLUCINATION MITIGATION METHODS

Several existing and concurrent works, such as HA-DPO (Zhao et al., 2023b), RLHF-V (Yu et al., 2023a), and RLAIF (Yu et al., 2024), have introduced hallucination mitigation techniques for MLLMs, that are derived from DPO (Rafailov et al., 2023). Following, we discuss the differences between our proposed DPA and DPO.

We write both DPA (ours) and the DPO (Rafailov et al., 2023) objectives using the same notations, which are as follows: $\pi_\theta$ as the model being trained; $\pi_{\text{ref}}$ as the frozen reference model; $x$ as the input; $y^c$ and $y^h$ as correct and hallucinated responses; $\mathcal{D}$ as training samples. We express $y^h$ as a sequence of tokens $T_h = \{t_1^h, t_2^h, \ldots, t_{|T_h|}^h\}$ and denote the $i$-th hallucinated phrase $y_i^h = T_h[s_i^h : e_i^h]$, where $s_i^h$ and $e_i^h$ are the start and end indices of $y_i^h$ with $1 \leq s_i^h \leq e_i^h \leq |T_h|$. Similarly, $y^c$ is expressed as a sequence of tokens $T_c = \{t_1^c, t_2^c, \ldots, t_{|T_c|}^c\}$, and we denote the $i$-th correct phrase $y_i^c = T_c[s_i^c : e_i^c]$, where $s_i^c$ and $e_i^c$ are the start and end indices of $y_i^c$ with $1 \leq s_i^c \leq e_i^c \leq |T_c|$. $N$ is the total number of hallucinated phrases in $y^h$; $\alpha$ and $\beta$ are loss coefficients to control the influence of the reference model in training. For the sake of simplicity, we assume that $\{x_c, y_c\}$ are reused as reference sample in DPA. Therefore, as discussed in Section 2, the final DPA loss can be expressed as:

$$
\mathcal{L}_{dpa}(\pi_\theta; \pi_{\text{ref}}) = - \mathbb{E}_{(x,y^c,y^h)\sim\mathcal{D}} \left[ \frac{1}{N} \sum_{i=1}^{N} \underbrace{- \log \frac{\prod_{j=s_i^c}^{e_i^c} \pi_\theta(t_j^c|x, t_{<j}^c)}{\prod_{j=s_i^c}^{e_i^c} \pi_\theta(t_j^c|x, t_{<j}^c) + \prod_{j=s_i^h}^{e_i^h} \pi_\theta(t_j^h|x, t_{<j}^h)}}_{\text{phrase-level alignment loss}} \right.
$$

$$
\left. + \alpha \cdot \underbrace{\sum_{j=1}^{|T_c|} \pi_{\text{ref}}(t_j^c|x, t_{<j}^c) \cdot \left( \log\left(\pi_{\text{ref}}(t_j^c|x, t_{<j}^c)\right) - \log\left(\pi_\theta(t_j^c|x, t_{<j}^c)\right) \right)}_{\text{token-wise KL divergence}} \right]
$$

On the other hand, the training objective of DPO is:

$$
\mathcal{L}_{dpo}(\pi_\theta; \pi_{\text{ref}}) = - \mathbb{E}_{(x,y^c,y^h)\sim\mathcal{D}} \left[ \log \sigma\left(\beta \log \frac{\pi_\theta(y^c|x)}{\pi_{\text{ref}}(y^c|x)} - \beta \log \frac{\pi_\theta(y^h|x)}{\pi_{\text{ref}}(y^h|x)}\right) \right]
$$

Note that in our proposed DPA ($\mathcal{L}_{dpa}$), given $\{x, y^c, y^h\}$, we calculate the phrase-level alignment loss based on the log-probabilities of the tokens in the hallucinated phrases and not on all the tokens of a sequence. Additionally, the KL-regularizer is applied at the token-level to closely retain the vision-language capabilities of the base model. In DPO ($\mathcal{L}_{\text{DPO}}$), however, given $x, y^c, y^h$, the reward margin between the correct and hallucinated responses is maximized to increase the log-likelihood of the correct response while reducing that of the hallucinated response. Despite the fact that the loss formulation of DPO is different from ours DPA, one fundamental difference is that their loss is calculated at a sequence level, i.e., penalizing all the tokens of a hallucinated response. Intuitively, the training objective of DPA provides more localized and fine-grained feedback unlike DPO (Rafailov et al., 2023) and other existing alignment techniques (Christiano et al., 2017; Schulman et al., 2017). This makes DPA unique and effective compared to existing and concurrent works.

Accordingly, the nature of the correct and hallucinated responses used in DPO-based methods and our DPA also differ. To illustrate this we present one side-by-side comparison using a training sample from HA-DPO (Zhao et al., 2023b) and ours in Figure S1, which shows that while HA-DPO make changes at the sequence level, we apply changes at the word or phrase-level to construct the negative responses. In particular, unlike, HA-DPO, we selectively alter the ground-truth information in the correct description, while keeping the rest of the response intact.

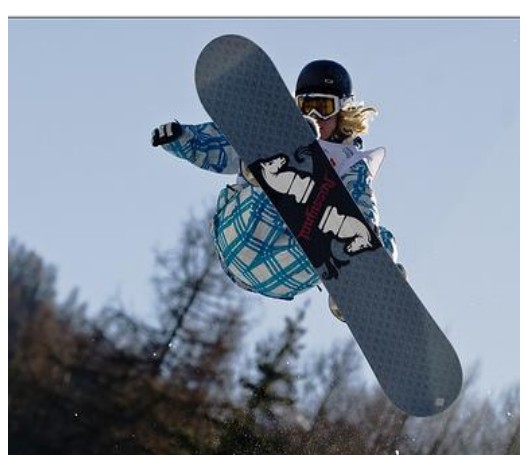 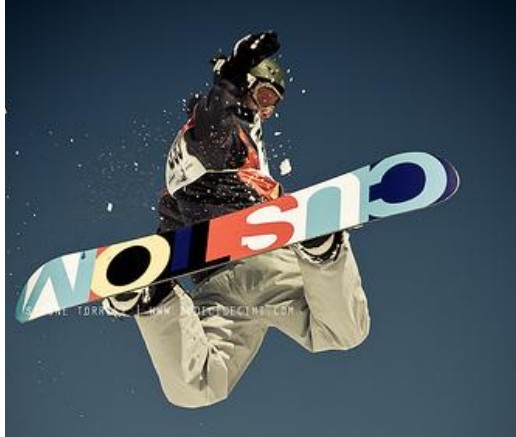

**Chosen:** The photo depicts an exciting moment of a snowboarder executing a mid-air jump, with the snowboard prominently visible underneath. The snowboarder is wearing protective gear, including a helmet and goggles, to ensure safety while experiencing the exhilarating activity. The snowy landscape with trees in the backdrop sets the scene, and the snowboarder takes center stage, exhibiting impressive skill and athleticism as they soar through the air.

**Rejected:** The picture depicts an electrifying moment of a snowboarder executing a mid-air jump, with the snowboard clearly visible underneath. The snowboarder, wearing a helmet and goggles, ensures safety while relishing in the exhilarating activity. The snowy landscape, adorned with trees, serves as the backdrop for this scene, where the snowboarder takes center stage, showcasing their skill and athleticism as they soar through the air.

**Correct:** A snowboarder is jumping in the air. The snowboarder is surrounded by snow and has a blue sky in the background. He has a patch of clear blue sky behind him. The snowboarder is doing a trick and has his legs bent in the air with his arms extended downward. He has a black and white glove on his right hand. The snowboarder is wearing a white vest with a black number on the back.

**Hallucinated:** A skier is jumping in the air. The skier is surrounded by snow and has a blue water in the background. He has a patch of clear blue water behind him. The skier is doing a trick and has his legs bent in the air with his arms extended downward. He has a black and white hat on his right hand. The skier is wearing a white vest with a black number on the back.

Figure S1: We present training samples from the DPO-based method on the left (from HA-DPO) and ours on the right, highlighting differences in the *nature of the negative samples*. While HA-DPO makes changes (highlighted in blue) at a sequence level, we apply one-to-one changes (highlighted in green and red) at the word or phrase-level to construct the negatives. The positives are referred to as 'Chosen' in HA-DPO, while we refer to them as 'Correct'; and the negatives are referred to as 'Reject' in HA-DPO, while we refer to them an 'Hallucinated'. Since there are no overlapping samples of descriptive responses between HA-DPO and our data, we use a sample that closely resemble each other.

# C ADDITIONAL EXPERIMENTS AND RESULTS

## C.1 ABLATION ON LOSS

Recall our final objective function, which is comprised of both alignment loss ($\mathcal{L}_a$), and token-wise KL divergence ($\mathcal{L}_d$) between the $\pi_\theta$ (the model being trained) and $\pi_{\text{ref}}$ (the reference model that is kept frozen), defined as: $\mathcal{L}_{dpa} = \mathcal{L}_a + \alpha \cdot \mathcal{L}_d$. First, we study the behavior of HALVA with varying $\alpha$. Simply put, a lower $\alpha$ allows $\pi_\theta$ to diverge more from $\pi_{\text{ref}}$, whereas a higher $\alpha$ aligns $\pi_\theta$ more closely with $\pi_{\text{ref}}$. By default, we initialize both $\pi_\theta$ and $\pi_{\text{ref}}$ from the same base model. Therefore, a higher $\alpha$ would result in $\pi_\theta$ to perform the same as the base model. Following, we analyze the impact of varying $\alpha$ on HALVA$_{7B}$ and HALVA$_{13B}$, while tracking their performance on the MME-Hall dataset. The results are presented in Figures S2 and S3. We observe that for HALVA$_{7B}$, an $\alpha$ of between 0.3 and 0.4 yields a better outcome, whereas the model behaves similar to the base model when $\alpha > 0.4$. For HALVA$_{13B}$ on the other hand, an $\alpha$ in the range of 0.4 to 0.6 shows the highest performance. We present qualitative examples in Figure S4, showing the adverse effect of using a very low $\alpha$. By default, we use $\alpha = 0.4$ for HALVA$_{7B}$, $\alpha = 0.5$ for HALVA$_{13B}$, and $\alpha = 0.2$ for HALVA$_{13B/384}$.

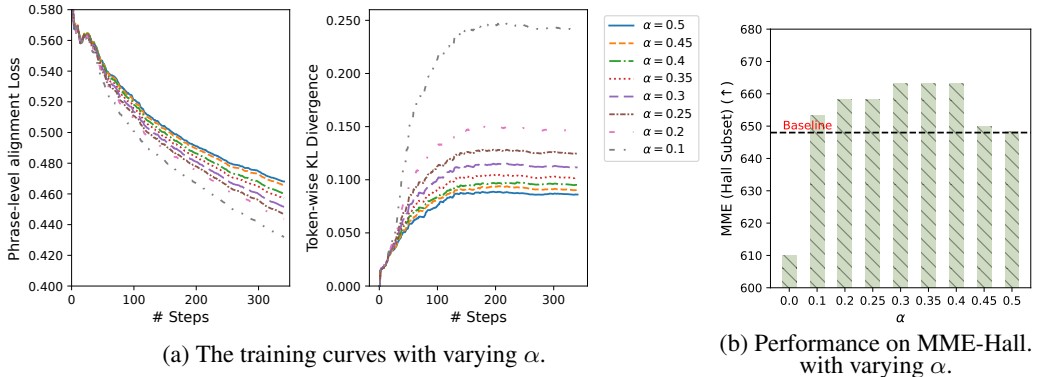

(a) The training curves with varying $\alpha$.

(b) Performance on MME-Hall. with varying $\alpha$.

Figure S2: The training curves with varying $\alpha$ (a) and their performance on object hallucination (b) are presented. $\alpha$ in the range of 0.3 to 0.4 achieves optimal performance on the 7B variant.

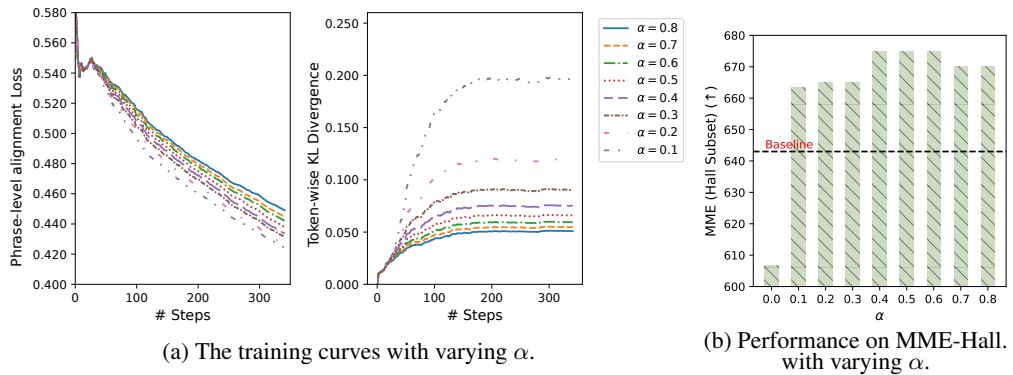

(a) The training curves with varying $\alpha$.

(b) Performance on MME-Hall. with varying $\alpha$.

Figure S3: The training curves with varying $\alpha$ (a) and their performance on object hallucination (b) are presented. $\alpha$ in the range of 0.4 to 0.6 achieves optimal performance on the 13B variant.

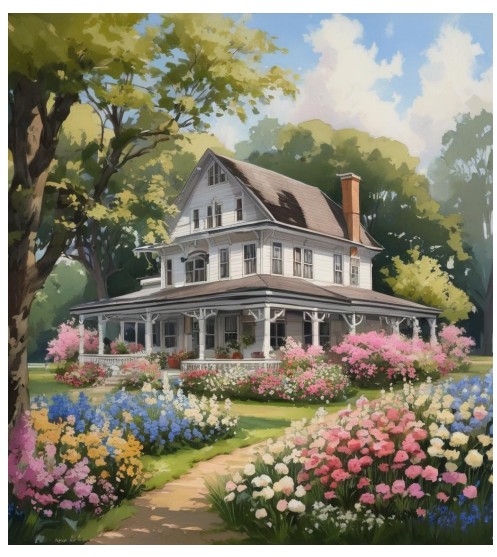 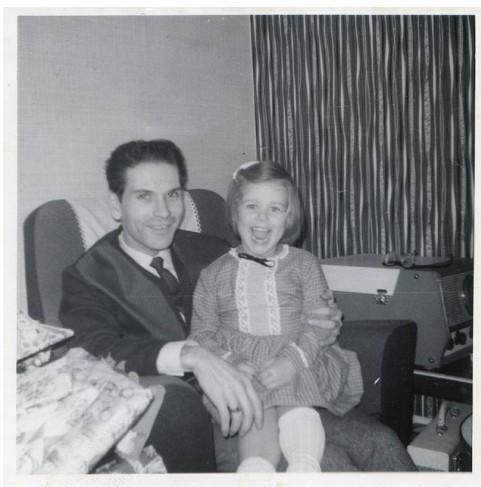

Question: Provide a one-sentence caption for the provided image.

$\alpha = 0.01$: A painting of a house with a white house with a pink house with a large white house with a pink house with a large white house with a pink house with a large white house with a pink house with a large white house with a pink house with a large white house with a pink house with a large

$\alpha = 0.4$ (default): A painting of a house with a white porch and pink flowers.

Question: Describe the image in detail.

$\alpha = 0.01$: The image is a black and white photo of a man and a little girl sitting together on a couch. The man is holding the little girl, and they are both smiling.

$\alpha = 0.4$ (default): The image is a black and white photo of a man and a young girl sitting together on a couch. The man is wearing a tie, and the little girl is wearing a dress. They are both smiling and appear to be enjoying their time together.

Figure S4: Qualitative examples demonstrating the impact of DPA training with a very low $\alpha$. As presented here, training with a very low $\alpha$ of 0.01 may occasionally hurt the language generation ability of an MLLM. The example on the left side shows an instance of degeneration, while the example on the right side shows a lack of descriptive power, failing to mention key details in the description, such as 'the man is wearing a tie' or 'the girl is wearing a dress'. The 7B variant is used in this study.

## C.2 Ablation on generative data augmentation

We perform an ablation study to explore the effect of different sampling strategies which have been used in generative data augmentation. As mentioned in Section 2, we generate hallucinated responses in three setups: closed-set co-occurrences (9K), open-set co-occurrences (11K), and Yes-or-No questions (1.5K). We generate a total of 21.5K samples that contains 28K unique pairs of correct and hallucinated phrases based on 5K unique hallucinated objects. We study the impact of these categories along with their varying number of samples. We perform this study on HALVA$_{7B}$ and use the same training hyperparameters as those obtained by tuning on the entire data. From the results presented in Table S1, three key observations are made. First, open-set hallucinated descriptions show benefits in reducing hallucinations in generative tasks, as evidenced by the superior performance on CHAIR. Second, mixing the Yes-or-No hallucinated responses reduces hallucination in discriminative tasks, leading to an F1 boost on the AMBER dataset. Finally, combining all the splits results in overall improvements or competitive performances across a broader range of tasks. We present the key statistics of all the splits in Table S2. In Figure S5, we present the training curves for different generative data augmentations, demonstrating stability during training across various data splits.

Table S1: Ablation study on sampling strategy used in generative data augmentation. $C_i$ and $C_s$ refer to CHAIR at instance and sentence-level; F1 refers to the F1-scores of all the discriminative tasks and HR refers to hallucination rate on generative tasks.

| Data Split | CHAIR | | AMBER | | MME-Hall |
|---|---|---|---|---|---|
| | $C_i \downarrow$ | $C_s \downarrow$ | F1$\uparrow$ | HR$\downarrow$ | Score $\uparrow$ |
| Closed set | 12.6 | 45.0 | 73.9 | 34.7 | 643.3 |
| Open-set | **11.2** | **39.6** | 73.1 | 33.3 | 643.3 |
| Closed set + Open-set (50%) | 11.7 | 41.8 | 79.8 | **32.0** | 643.3 |
| Closed set + Open-set | 12.6 | 43.6 | 74.1 | 34.0 | 648.3 |
| Closed set + Open-set + Y-or-N (50%) | 11.8 | 43.2 | 82.4 | 32.2 | 641.0 |
| **Closed set + Open-set + Y-or-N** | 11.7 | 41.4 | **83.4** | 32.2 | **665.0** |

Table S2: Key statistics of training samples used in DPA training.

| Data Split | # Samples | # Avg. hallucinated instances per sample | Length (in words) Avg./Min./Max. |
|---|---|---|---|
| One-sentence caption | 528 | 2.7 | 15/6/53 |
| Short description | 11573 | 6.9 | 42/12/128 |
| Detailed description | 8268 | 11.3 | 71/32/246 |
| Yes-or-No (one word answer) | 1510 | 1 | 1/1/1 |
| Full | 21874 | 8.1 | 49/1/246 |

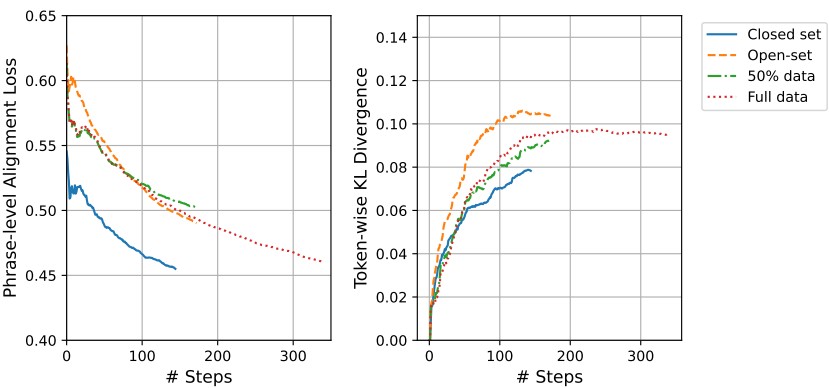

Figure S5: Training curves for different generative data augmentations using $\alpha = 0.4$.

Table S3: Ablation study on divergence measure using HALVA$_{7B}$. **(a)** We find that using *seen* samples as the reference data for divergence measure achieve overall better performance. **(b)** Our study shows that initializing the reference model and the model being trained from the same checkpoint, achieves optimal performance. $C_i$ and $C_s$ refer to CHAIR at instance and sentence-level; F1 refers to the F1-scores of all the discriminative tasks and HR refers to hallucination rate in the image descriptions.

(a) Ablation study on reference data.

| Ref. Data | CHAIR | | AMBER | | MME-Hall |
|---|---|---|---|---|---|
| | $C_i \downarrow$ | $C_s \downarrow$ | F1↑ | HR↓ | Score ↑ |
| Unseen data | 12.7 | 47.4 | 81.7 | 34.7 | **668.3** |
| **Seen data** | **11.7** | **41.4** | **83.4** | **32.2** | 665.0 |

(b) Ablation study on reference model.

| Ref. Model | CHAIR | | AMBER | | MME-Hall |
|---|---|---|---|---|---|
| | $C_i \downarrow$ | $C_s \downarrow$ | F1↑ | HR↓ | Score ↑ |
| **7B** | **11.7** | **41.4** | **83.4** | **32.2** | **665.0** |
| 13B | 12.4 | 45.2 | 80.1 | 34.7 | 640.0 |

## C.3 ABLATION ON DIVERGENCE MEASURE

**Reference data.** We experiment with the reference data that has been used to measure KL divergence with respect to the reference model. We briefly experiment in two setups:

- Unseen data: we directly use the vision-language instructions and *correct* responses as the reference samples.
- Seen data: we take a fraction of the instruction tuning dataset the base model is originally trained on, and use them as reference samples.

We perform this experiment on HALVA$_{7B}$ and the results are presented in Table S3 (a). The results demonstrate that using seen samples to measure divergence gives a better estimate of model state during training, and accordingly the tuned model overall performs better, across various benchmarks.

**Reference model.** By default, we initialize the reference model (the model kept frozen) and the online model (the model being trained) from the same checkpoint. Additionally, we experiment with initializing the reference model different than the model being trained. In particular, we experiment with training LLaVA$_{7B}$ while using LLaVA$_{13B}$ as the reference model. We find this interesting to explore as both LLaVA$_{7B}$ and LLaVA$_{13B}$ are originally trained in a similar setup, and LLaVA$_{13B}$ performs relatively better compared to the LLaVA$_{7B}$, on most of the benchmarks (Liu et al., 2023c). The results presented in Table S3 (b) show that initializing the reference model and the online model from the same checkpoint, achieve optimal performance. We believe this is likely since the reference model initialized from an identical state of the model being trained, gives a true estimate of divergence and accordingly optimized model performs better across a variety of benchmarks.

## C.4 DETAILED RESULTS OF MME-HALL

In Table S4, we present the detailed results of the MME-Hall (Fu et al., 2023) benchmark across its four sub-categories: existence, count, position, and color. Our results indicate that DPA mitigates (or retains the same performance as the base model) object hallucination across different aspects, unlike prior finetuning methods such as HA-DPO (Zhao et al., 2023b) and EOS (Yue et al., 2024), or inference-based methods such as VCD (Leng et al., 2023) and Woodpecker (Yin et al., 2023), which either degrade overall performance or show improvement in one category but suffer in others.

Table S4: Detailed results on **MME-Hall**.

| Method | Object (↑) | | Attribute (↑) | | Total (↑) |
|---|---|---|---|---|---|
| | Existence | Count | Position | Color | |
| LLaVA-v1.5$_{7B}$ | 190.0 | 155.0 | 133.3 | 170.0 | 648.3 |
| HA-DPO$_{7B}$ | 190.0 | 133.3 | 136.7 | 158.3 | 618.3 |
| EOS$_{7B}$ | 190.0 | 138.3 | 118.3 | 160.0 | 606.7 |
| VCD$_{7B}$ | 184.7 | 138.3 | 128.7 | 153.0 | 604.7 |
| Woodpecker$_{7B}$ | 165.0 | 98.3 | 56.7 | 46.7 | 366.7 |
| **HALVA$_{7B}$ (Ours)** | 190.0 | 165.0 | 135.0 | 175.0 | **665.0** |
| LLaVA-v1.5$_{13B}$ | 185.0 | 155.0 | 133.3 | 170.0 | 643.3 |
| **HALVA$_{13B}$ (Ours)** | 190.0 | 163.3 | 141.7 | 180.0 | **675.0** |
| VILA-v1.5$_{13B/384}$ | 185.0 | 170.0 | 148.3 | 185.0 | 688.3 |
| **HALVA$_{13B/384}$ (Ours)** | 185.0 | 173.3 | 148.3 | 185.0 | **691.7** |

## C.5 DETAILED RESULTS OF AMBER

In Table S5, we present the detailed results of AMBER (Wang et al., 2023b). As shown, HALVA reduces hallucination (e.g., from 7.8 to 6.6) while improving object coverage (from 51% to 53%) in image description tasks, outperforming HA-DPO, EOS, and Woodpecker. HALVA significantly improves performance on discriminative tasks, achieving F1 score improvements of up to 13.4%.

Table S5: Detailed results on **AMBER**.

| Method | Generative Task | | | | Discriminative Task (F1↑) | | | |
|---|---|---|---|---|---|---|---|---|
| | CHAIR (↓) | Coverage (↑) | Hall. Rate (↓) | Cognition (↓) | Existence | Attribute | Relation | Overall |
| LLaVA-v1.5$^{\ddagger}_{7B}$ | 7.8 | 51.0 | 36.4 | 4.2 | 83.3 | 64.6 | 65.6 | 74.7 |
| HA-DPO$_{7B}$ | 6.7 | 49.8 | 30.9 | 3.3 | 88.1 | 66.1 | 68.8 | 78.1 |
| EOS$_{7B}$ | 5.1 | 49.1 | 22.7 | 2.0 | 82.8 | 67.4 | 69.2 | 75.6 |
| Woodpecker$_{7B}$ | 6.9 | 48.9 | 30.4 | 3.6 | 81.7 | 53.5 | 41.5 | 67.0 |
| **HALVA$_{7B}$ (Ours)** | 6.6 | 53.0 | 32.2 | 3.4 | 93.3 | 77.1 | 63.1 | **83.4** |
| LLaVA-v1.5$_{13B}$ | 6.6 | 51.9 | 30.5 | 3.3 | 78.5 | 70.2 | 45.0 | 73.1 |
| **HALVA$_{13B}$ (Ours)** | 6.4 | 52.6 | 30.4 | 3.2 | 92.6 | 81.4 | 73.5 | **86.5** |
| VILA-v1.5$_{13B/384}$ | 9.9 | 63.3 | 56.1 | 4.8 | 87.5 | 77.8 | 66.7 | 82.2 |
| **HALVA$_{13B/384}$ (Ours)** | 9.1 | 63.9 | 54.2 | 4.0 | 93.9 | 82.6 | 75.9 | **87.9** |

## C.6 DETAILED RESULTS OF MMHAL-BENCH

In Table S6, we present the detailed results of MMHal-Bench (Sun et al., 2023) across its eight sub-categories. Our proposed DPA demonstrates consistent effectiveness in mitigating object hallucinations in the following types: adversarial, comparison, relation, and holistic on both HALVA$_{7B}$ and HALVA$_{13B}$. Additionally, DPA improves performance in 6 out of 8 subcategories for both the 13B variants. Moreover, recent hallucination mitigation methods such as HA-DPO and EOS prove ineffective in addressing such broad categories of hallucinations, even resulting in worsened baseline performance.

Table S6: Detailed results on **MMHal-Bench**.

| Method | Overall Score (↑) | Hall. Rate (↓) | Score in Each Question Type (↑) | | | | | | | |
|---|---|---|---|---|---|---|---|---|---|---|
| | | | Attribute | Adversarial | Comparison | Counting | Relation | Environment | Holistic | Other |
| LLaVA-v1.5$_{7B}$ | $2.11_{\pm0.06}$ | $0.56_{\pm0.01}$ | $3.06_{\pm0.27}$ | $1.00_{\pm0.00}$ | $1.61_{\pm0.05}$ | $1.97_{\pm0.09}$ | $2.36_{\pm0.05}$ | $3.20_{\pm0.05}$ | $2.14_{\pm0.30}$ | $1.53_{\pm0.25}$ |
| HA-DPO$_{7B}$ | $1.97_{\pm0.04}$ | $0.59_{\pm0.01}$ | $3.56_{\pm0.17}$ | $1.08_{\pm0.09}$ | $1.14_{\pm0.13}$ | $1.89_{\pm0.21}$ | $2.22_{\pm0.33}$ | $3.31_{\pm0.10}$ | $1.42_{\pm0.14}$ | $1.17_{\pm0.00}$ |
| EOS$_{7B}$ | $2.03_{\pm0.02}$ | $0.59_{\pm0.02}$ | $2.69_{\pm0.13}$ | $1.78_{\pm0.09}$ | $1.89_{\pm0.13}$ | $1.53_{\pm0.18}$ | $2.09_{\pm0.14}$ | $3.08_{\pm0.30}$ | $1.67_{\pm0.29}$ | $1.53_{\pm0.09}$ |
| **HALVA$_{7B}$ (Ours)** | $2.25_{\pm0.10}$ | $0.54_{\pm0.01}$ | $2.78_{\pm0.09}$ | $1.47_{\pm0.18}$ | $1.97_{\pm0.13}$ | $1.89_{\pm0.05}$ | $3.03_{\pm0.21}$ | $3.20_{\pm0.05}$ | $2.42_{\pm0.43}$ | $1.22_{\pm0.27}$ |
| LLaVA-v1.5$_{13B}$ | $2.38_{\pm0.02}$ | $0.50_{\pm0.01}$ | $3.20_{\pm0.05}$ | $2.53_{\pm0.18}$ | $2.55_{\pm0.05}$ | $2.20_{\pm0.05}$ | $1.97_{\pm0.05}$ | $3.33_{\pm0.14}$ | $1.50_{\pm0.22}$ | $1.72_{\pm0.13}$ |
| **HALVA$_{13B}$ (Ours)** | $2.58_{\pm0.08}$ | $0.46_{\pm0.02}$ | $3.03_{\pm0.09}$ | $2.58_{\pm0.09}$ | $2.66_{\pm0.14}$ | $2.08_{\pm0.14}$ | $2.45_{\pm0.05}$ | $3.36_{\pm0.17}$ | $2.44_{\pm0.39}$ | $2.00_{\pm0.08}$ |
| VILA-v1.5$_{13B/384}$ | $2.58_{\pm0.02}$ | $0.46_{\pm0.01}$ | $3.36_{\pm0.13}$ | $1.08_{\pm0.09}$ | $3.39_{\pm0.13}$ | $2.05_{\pm0.05}$ | $2.97_{\pm0.21}$ | $3.11_{\pm0.05}$ | $2.19_{\pm0.13}$ | $2.47_{\pm0.05}$ |
| **HALVA$_{13B/384}$ (Ours)** | $2.58_{\pm0.06}$ | $0.45_{\pm0.01}$ | $3.11_{\pm0.05}$ | $1.47_{\pm0.05}$ | $3.47_{\pm0.05}$ | $2.08_{\pm0.00}$ | $3.11_{\pm0.13}$ | $3.19_{\pm0.13}$ | $1.64_{\pm0.24}$ | $2.58_{\pm0.09}$ |

## C.7 DETAILED RESULTS OF HALLUSIONBENCH

In Table S7, we present the detailed results of HallusionBench (Liu et al., 2023a), which evaluates MLLMs beyond object hallucination, including those may cause by visual illusions and quantitative analysis form charts or graphs, among others. In addition to improving the overall performance, the results demonstrate the effectiveness of DPA on all the sub-categories (i.e., easy set, hard set) of HallusionBench as well. For example, we find that HALVA$_{7B}$ and HALVA$_{13B}$ substantially improve performance (4.34%-6.90%) on the *Hard Set* of HallusionBench, which consists of human-edited image-question pairs specially crafted to elicit hallucinations in MLLMs. We note that, in addition to hallucination mitigation, DPA helps MLLMs in reducing Yes/No bias. As discussed earlier, LLaVA-v1.5 is prone to answering 'Yes', in most cases. Our proposed DPA effectively reduces Yes/No bias from 0.31 to 0.17 and from 0.38 to 0.20 on HALVA$_{7B}$ and HALVA$_{13B}$, respectively. Moreover, in the case of HALVA$_{13B/384}$, the Yes/No bias is reduced from 0.19 to 0.02, with 0 being ideal.

## C.8 A CRITICAL ANALYSIS OF OUR PROPOSED DPA

Here, we critically assess whether the performance enhancement observed in our proposed DPA is attributable to generative data augmentation, the proposed training objective, or their combination. To investigate this, we apply our generative data augmentation directly to another finetuning-based hallucination mitigation approach, HA-DPO (Zhao et al., 2023b). In HA-DPO, correct and hallucinated pairs are employed to finetune MLLMs,

Table S7: Detailed results on **HallusionBench**.

| Method | Yes/No Bias | | Question Pair Acc. | Fig. Acc. | Easy Acc. | Hard Acc. | All Acc. |
|---|---|---|---|---|---|---|---|
| | Pct. Diff ($\sim 0$) | FP Ratio ($\sim 0.5$) | ($qAcc$) ↑ | ($fAcc$) ↑ | (Easy $aAcc$) ↑ | (Hard $aAcc$) ↑ | ($aAcc$) ↑ |
| LLaVA-v1.5$_{7B}$ | $0.31_{\pm 0.00}$ | $0.79_{\pm 0.00}$ | $10.70_{\pm 0.13}$ | $19.65_{\pm 0.00}$ | $42.34_{\pm 0.13}$ | $41.47_{\pm 0.13}$ | $47.09_{\pm 0.14}$ |
| HA-DPO$_{7B}$ | 0.26 | 0.76 | 11.21 | 19.08 | 42.86 | 44.19 | 48.36 |
| EOS$_{7B}$ | 0.29 | 0.78 | 11.21 | 18.50 | 43.96 | 42.09 | 48.72 |
| **HALVA$_{7B}$ (Ours)** | $\mathbf{0.17}_{\pm 0.00}$ | $\mathbf{0.67}_{\pm 0.00}$ | $13.85_{\pm 0.00}$ | $21.48_{\pm 0.17}$ | $42.71_{\pm 0.13}$ | $45.81_{\pm 0.00}$ | $\mathbf{48.95}_{\pm 0.14}$ |
| LLaVA-v1.5$_{13B}$ | $0.38_{\pm 0.00}$ | $0.85_{\pm 0.00}$ | $8.79_{\pm 0.22}$ | $15.22_{\pm 0.17}$ | $44.25_{\pm 0.13}$ | $35.97_{\pm 0.13}$ | $46.50_{\pm 0.09}$ |
| **HALVA$_{13B}$ (Ours)** | $\mathbf{0.20}_{\pm 0.00}$ | $\mathbf{0.70}_{\pm 0.00}$ | $13.85_{\pm 0.22}$ | $20.13_{\pm 0.17}$ | $44.47_{\pm 0.13}$ | $42.87_{\pm 0.13}$ | $\mathbf{49.10}_{\pm 0.05}$ |
| VILA-v1.5$_{13B/384}$ | $0.19_{\pm 0.00}$ | $0.71_{\pm 0.00}$ | $18.90_{\pm 0.00}$ | $24.86_{\pm 0.29}$ | $52.38_{\pm 0.13}$ | $46.20_{\pm 0.27}$ | $55.39_{\pm 0.05}$ |
| **HALVA$_{13B/384}$ (Ours)** | $\mathbf{0.02}_{\pm 0.00}$ | $\mathbf{0.53}_{\pm 0.00}$ | $22.71_{\pm 0.46}$ | $27.65_{\pm 0.17}$ | $52.89_{\pm 0.34}$ | $46.96_{\pm 0.23}$ | $\mathbf{56.60}_{\pm 0.18}$ |

aiming to maximize the reward margin between the correct responses and the hallucinated ones. Accordingly, we train HA-DPO by replacing their data with the output of our generative data augmentation module. We utilize the official code released by (Zhao et al., 2023b) and conduct hyper-parameter tuning (mainly with varying $\beta$ and learning rate) ensure effective training. Subsequently, we evaluate the performance of the newly trained HA-DPO on both hallucination (CHAIR, AMBER, MME-Hall) and non-hallucination (MME) benchmarks. The results presented in Table S8 indicate that applying our proposed generative data augmentation to HA-DPO does not yield the same level of performance boost as HALVA. This confirms that the performance boost of our proposed method stems from a combination of the KL-regularized phrase-level alignment objective and the data augmentation setup. Note that since our proposed method necessitates a pair of aligned correct and hallucinated phrases, and the descriptive responses utilized in HA-DPO do not meet this requirement, we are unable to apply DPA directly to their data.

Table S8: Effect of generative data augmentation on HA-DPO. Here, CHAIR, AMBER, and MME-Hall are hallucination benchmarks, and MME is a general vision-language benchmark.

| | CHAIR ($C_i$) ↓ | AMBER F1 ↑ | MME-Hall ↑ | MME ↑ |
|---|---|---|---|---|
| HA-DPO$_{7B}$ | **11.0** | 78.1 | 618.3 | 1502.6 |
| HA-DPO$_{7B}$ w/ Generative Data Aug. | 14.6 | 77.7 | 631.7 | 1508.9 |
| HALVA$_{7B}$ | 11.7 | **83.4** | **665.0** | **1527.0** |

## C.9 RESULTS ON POPE

In addition to the hallucination benchmarks in the main paper, we also evaluate HALVA using POPE (Li et al., 2023d). While POPE is used in prior works, we note a few key limitations and find it to be a not well suited benchmark for evaluating MLLMs, as listed below. Please note that the similar concerns are also echoed in recent works (Wang et al., 2023b; Bai et al., 2024).

First, POPE employs a Yes-or-No protocol to check for existence of an object, but lacks coverage of other types of object hallucinations, such as object attributes (e.g., color, count) and object relations (e.g., position, environment). Second, the questions are formulated based on only 500 images and include a total of 79 unique objects, which fails to capture object hallucinations across diverse visual concepts. Third, POPE does not evaluate hallucinations in descriptive tasks (e.g., image description), where MLLMs tend to hallucinate more. These limitations led to introduction of more comprehensive benchmarks such as AMBER and MME among others, which we are used as the primary evaluation benchmarks in this work.

As shown in Table S9, we observe that while models such as GPT-4o and InternVL2 perform considerably better than others on MME and HallusionBench, they are not well-represented by POPE. Despite these shortcomings, we were able to obtain 87.1 and 87.9 for HALVA$_{7B}$ and HALVA$_{13B}$ using a different $\alpha = 0.005$.

## C.10 RESULTS ON LINGUISTIC QUALITY

To analyse whether DPA training have an adverse affect on the linguistic quality of the responses generated by MLLMs, we evaluate the responses on four aspects: grammatical correctness, fluency, detailedness, and choice of words. Since there is no standard or commonly used benchmark for these tasks, we use randomly selected 100 detailed image descriptions (a subset from the AMBER (Wang et al., 2023b) image description task) generated

Table S9: The results on POPE are presented. * Results are obtained using a different $\alpha$ than our default. † Added here for reference only, and should not be directly compared with 7B and 13 models, due to the large discrepancy in their model sizes.

| Method | POPE (F1 ↑) | AMBER (F1 ↑) | HallusionBench (Acc. ↑) | MME-Hall (Score ↑) | MME (Score ↑) |
|---|---|---|---|---|---|
| LLaVA-v1.5$_{7B}$ | 85.9 | 74.7 | 47.1 | 684.3 | 1510.7 |
| LLaVA-RLHF$_{7B}$ | 81.5 | 76.3 | 43.0 | 493.3 | 1190.0 |
| HA-DPO$_{7B}$ | 86.9 | 78.1 | 48.4 | 618.3 | 1502.6 |
| EOS$_{7B}$ | 86.0 | 75.6 | 48.7 | 606.7 | 1424.4 |
| **HALVA$_{7B}$ (Ours)** | 84.8/87.1* | **83.4** | **49.0** | **665.0** | **1527.0** |
| LLaVA-v1.5$_{13B}$ | 85.9 | 73.1 | 46.5 | 643.3 | 1530.1 |
| LLaVA-RLHF$_{13B}$ | 81.9 | 83.7 | 46.4 | 585.0 | 1367.7 |
| **HALVA$_{13B}$ (Ours)** | 84.9/87.9* | **86.5** | **49.1** | **675.0** | **1544.0** |
| VILA-v1.5$_{13B}$ | 86.3 | 82.2 | 55.4 | 688.3 | 1569.6 |
| **HALVA$_{13B/384}$ (Ours)** | 86.1 | **87.9** | **56.6** | **691.7** | **1575.7** |
| GPT-4o† (v.0513, detail-high) | 85.6 | - | 55.0 | - | 2310.3 |
| InternVL2$_{40B}$† (Chen et al., 2024) | 81.9 | - | 56.5 | - | 2293.1 |

by LLaVA 1.5$_{7B}$ and HALVA$_{7B}$, with GPT-4o-mini as the judge to rate them on a scale of 0 to 10. The template used in evaluation is presented in Figure S6. As shown in Table S10, HALVA$_{7B}$ exhibits the same performance as LLaVA 1.5$_{7B}$.

Table S10: Results on linguistic qualities of the responses.

| Model | Grammatical Correctness | Fluency | Detailedness | Choice of Words |
|---|---|---|---|---|
| LLaVA 1.5$_{7B}$ | $9.90 \pm 0.30$ | $9.64 \pm 0.52$ | $8.37 \pm 0.48$ | $8.93 \pm 0.26$ |
| **HALVA$_{7B}$ (Ours)** | $9.99 \pm 0.10$ | $9.51 \pm 0.50$ | $8.35 \pm 0.48$ | $8.99 \pm 0.23$ |

```
Following is a detailed image description.
Your task is to assess the response on the following criteria:
1. Grammatical Correctness: Analyze the response for grammar,
punctuation, and syntax accuracy.
2. Fluency: Evaluate whether the response flows smoothly,
reads naturally, and maintains coherence throughout.
3. Detailedness: Check if the response provides sufficient and
relevant detail to address the topic comprehensively, without
redundancy or unnecessary information.
4. Choice of Words: Assess if the words used are appropriate,
varied, and effectively convey the intended message.

Rate each criterion on a scale from 0 to 10, where 0 indicates
poor quality and 10 signifies an excellent response.

Here is the image description to evaluate:

{description}

Your response should be in this format:

Grammatical Correctness: SCORE
Fluency: SCORE
Detailedness: SCORE
Choice of Words: SCORE
```

Figure S6: The template for evaluating the linguistic quality of the responses.

Table S11: Analysing if the base models are better than the hallucination mitigation methods on general tasks. Here Model 1 is a base model and Model 2 is a hallucination mitigation method. Red: performance drop with statistical significance.

| Evaluation benchmark | Model 1 | Model 2 | $\Delta$ | Adjusted $\Delta$ | Standard Error | # samples |
|---|---|---|---|---|---|---|
| **LLaVA-v1.5$_{7B}$ vs. HA-DPO$_{7B}$** | | | | | | |
| VQAv2 | 0.7850 | 0.7760 | 0.0090 | 0.0065 | 0.0013 | 107394 |
| MMVet | 0.3110 | 0.3070 | 0.0040 | -0.0574 | 0.0314 | 218 |
| TextVQA | 0.5820 | 0.5670 | 0.0150 | 0.0013 | 0.0070 | 5000 |
| MME | 0.7554 | 0.7513 | 0.0041 | -0.0143 | 0.0093 | 2114 |
| LLaVA-Bench-in-the-wild | 0.6540 | 0.6620 | -0.008 | -0.1284 | 0.0614 | 60 |
| **LLaVA-v1.5$_{7B}$ vs. EOS$_{7B}$** | | | | | | |
| VQAv2 | 0.7850 | 0.7760 | 0.0090 | 0.0065 | 0.0013 | 107394 |
| MMVet | 0.3110 | 0.3140 | -0.0030 | -0.0644 | 0.0314 | 218 |
| TextVQA | 0.5820 | 0.5520 | 0.0300 | 0.0163 | 0.0070 | 5000 |
| MME | 0.7554 | 0.7122 | 0.0432 | 0.0248 | 0.0093 | 2114 |
| LLaVA-Bench-in-the-wild | 0.6540 | 0.6580 | -0.0040 | -0.1244 | 0.0614 | 60 |
| **LLaVA-v1.5$_{7B}$ vs. HALVA$_{7B}$ (Ours)** | | | | | | |
| VQAv2 | 0.7850 | 0.7850 | 0.0000 | -0.0025 | 0.0013 | 107394 |
| MMVet | 0.3110 | 0.3210 | -0.0100 | -0.0714 | 0.0314 | 218 |
| TextVQA | 0.5820 | 0.5820 | 0.0000 | -0.0137 | 0.0070 | 5000 |
| MME | 0.7554 | 0.7635 | -0.0081 | -0.0265 | 0.0093 | 2114 |
| LLaVA-Bench-in-the-wild | 0.6540 | 0.6720 | -0.0180 | -0.1384 | 0.0614 | 60 |
| **LLaVA-v1.5$_{13B}$ vs. HALVA$_{13B}$ (Ours)** | | | | | | |
| VQAv2 | 0.8000 | 0.8000 | 0.0000 | -0.0024 | 0.0012 | 107394 |
| MMVet | 0.3610 | 0.3780 | -0.0170 | -0.0808 | 0.0325 | 218 |
| TextVQA | 0.6120 | 0.6120 | 0.0000 | -0.0135 | 0.0069 | 5000 |
| MME | 0.7651 | 0.7720 | -0.0070 | -0.0250 | 0.0092 | 2114 |
| LLaVA-Bench-in-the-wild | 0.7250 | 0.7270 | -0.0020 | -0.1150 | 0.0576 | 60 |
| **VILA-v1.5$_{13B/384}$ vs. HALVA$_{13B/384}$ (Ours)** | | | | | | |
| VQAv2 | 0.8280 | 0.8280 | 0.0000 | -0.0023 | 0.0012 | 107394 |
| MMVet | 0.4430 | 0.4430 | 0.0000 | -0.0659 | 0.0336 | 218 |
| TextVQA | 0.6500 | 0.6480 | 0.0020 | -0.0112 | 0.0067 | 5000 |
| MME | 0.7848 | 0.7879 | -0.0031 | -0.0206 | 0.0089 | 2114 |
| LLaVA-Bench-in-the-wild | 0.8080 | 0.8240 | -0.0160 | -0.1157 | 0.0508 | 60 |

## C.11 STATISTICAL ANALYSIS

We accept the performance improvement or drop in Model 1 compared to Model 2 on a given task to be statistically significant if the Adjusted $\Delta$ is greater than 0, where $\Delta$ is the performance difference between Model 1 and Model 2. We consider the Standard Error (SE) with statistical significance at 95% confidence. Note that the original results are scaled between 0 to 1, where 0 represents the worst and 1 represents the best. The mathematical expressions are given below

$$SE = \sqrt{\frac{\text{Model 1} \times (1 - \text{Model 1})}{\text{number of samples}}},$$

$$\Delta = \text{Model 1} - \text{Model 2},$$

$$\text{Adjusted } \Delta = \Delta - 1.96 * \text{SE}.$$

The results are presented in Tables S11 to S14 show that the existing finetuning-based hallucination mitigation methods such as HA-DPO and EOS show statistically significant performance drops on general tasks. In contrast, our proposed DPA does not exhibit such deterioration. Moreover, we observe that HALVA$_{7B}$ shows statistically significant improvements in CHAIR, AMBER generative, and AMBER discriminative tasks. The same holds true for HA-DPO$_{7B}$ and EOS$_{7B}$. Both of our 13B variants (HALVA$_{13B}$ and HALVA$_{13B/384}$) show improvements across all setups, with the improvements on AMBER discriminative tasks being statistically significant. Unlike DPA, existing methods, such as HA-DPO and EOS, exhibit performance deterioration in 2 out of 6 hallucination tasks compared to the base model, where the performance drop for EOS$_{7B}$ on MME-Hall is statistically significant.

Table S12: Analysing if hallucination mitigation methods are better than the base models on general tasks. Here Model 1 is a hallucination mitigation method and Model 2 is a base model.

| Evaluation benchmark | Model 1 | Model 2 | Δ | Adjusted Δ | Standard Error | # samples |
|---|---|---|---|---|---|---|
| **HA-DPO$_{7B}$ vs. LLaVA-v1.5$_{7B}$** | | | | | | |
| VQAv2 | 0.7760 | 0.7850 | -0.0090 | -0.0115 | 0.0013 | 107394 |
| MMVet | 0.3070 | 0.3110 | -0.0040 | -0.0652 | 0.0312 | 218 |
| TextVQA | 0.5670 | 0.5820 | -0.0150 | -0.0287 | 0.0070 | 5000 |
| MME | 0.7513 | 0.7554 | -0.0041 | -0.0225 | 0.0094 | 2114 |
| LLaVA-Bench-in-the-wild | 0.6620 | 0.6540 | 0.008 | -0.1117 | 0.0611 | 60 |
| **EOS$_{7B}$ vs. LLaVA-v1.5$_{7B}$** | | | | | | |
| VQAv2 | 0.7760 | 0.7850 | -0.0090 | -0.0115 | 0.0013 | 107394 |
| MMVet | 0.3140 | 0.3110 | 0.0030 | -0.0586 | 0.0314 | 218 |
| TextVQA | 0.5520 | 0.5820 | -0.0300 | -0.0438 | 0.0070 | 5000 |
| MME | 0.7122 | 0.7554 | -0.0432 | -0.0624 | 0.0098 | 2114 |
| LLaVA-Bench-in-the-wild | 0.6580 | 0.6540 | 0.0040 | -0.1160 | 0.0612 | 60 |
| **HALVA$_{7B}$ (Ours) vs. LLaVA-v1.5$_{7B}$** | | | | | | |
| VQAv2 | 0.7850 | 0.7850 | 0.0000 | -0.0025 | 0.0013 | 107394 |
| MMVet | 0.3210 | 0.3110 | 0.0100 | -0.0520 | 0.0316 | 218 |
| TextVQA | 0.5820 | 0.5820 | 0.0000 | -0.0137 | 0.007 | 5000 |
| MME | 0.7635 | 0.7554 | 0.0081 | -0.0100 | 0.0092 | 2114 |
| LLaVA-Bench-in-the-wild | 0.6720 | 0.6540 | 0.0180 | -0.1008 | 0.0606 | 60 |
| **HALVA$_{13B}$ (Ours) vs. LLaVA-v1.5$_{13B}$** | | | | | | |
| VQAv2 | 0.8000 | 0.8000 | 0.0000 | -0.0024 | 0.0012 | 107394 |
| MMVet | 0.3780 | 0.3610 | 0.0170 | -0.0474 | 0.0328 | 218 |
| TextVQA | 0.6120 | 0.6120 | 0.0000 | -0.0135 | 0.0069 | 5000 |
| MME | 0.7720 | 0.7651 | 0.0070 | -0.0109 | 0.0091 | 2114 |
| LLaVA-Bench-in-the-wild | 0.7270 | 0.7250 | 0.002 | -0.1107 | 0.0575 | 60 |
| **HALVA$_{13B/384}$ (Ours) vs. VILA-v1.5$_{13B/384}$** | | | | | | |
| VQAv2 | 0.8280 | 0.8280 | 0.0000 | -0.0023 | 0.0012 | 107394 |
| MMVet | 0.4430 | 0.4430 | 0.0000 | -0.0659 | 0.0336 | 218 |
| TextVQA | 0.6480 | 0.6500 | -0.0020 | -0.0152 | 0.0068 | 5000 |
| MME | 0.7879 | 0.7848 | 0.0031 | -0.0144 | 0.0089 | 2114 |
| LLaVA-Bench-in-the-wild | 0.8240 | 0.8080 | 0.0160 | -0.0804 | 0.0492 | 60 |

Table S13: Analysing if hallucination mitigation methods are better than the base models on hallucination tasks. Here Model 1 is a hallucination mitigation method and Model 2 is a base model. Green: performance improvement with statistical significance.

| Evaluation benchmark | Model 1 | Model 2 | Δ | Adjusted Δ | Standard Error | # samples |
|---|---|---|---|---|---|---|
| **HA-DPO$_{7B}$ vs. LLaVA-v1.5$_{7B}$** | | | | | | |
| CHAIR | 0.6180 | 0.5000 | 0.1180 | 0.0754 | 0.0217 | 500 |
| MME-Hall | 0.7729 | 0.8104 | -0.0375 | -0.0905 | 0.027 | 240 |
| AMBER-Generative (Hall. Rate) | 0.6910 | 0.6360 | 0.0550 | 0.0264 | 0.0146 | 1004 |
| AMBER-Discriminative | 0.7810 | 0.7470 | 0.0340 | 0.0272 | 0.0035 | 14216 |
| MMHal-Bench (Hall. Rate) | 0.4000 | 0.4600 | -0.0600 | -0.1580 | 0.0500 | 96 |
| Hallusion-Bench | 0.4836 | 0.4709 | 0.0127 | -0.0165 | 0.0149 | 1129 |
| **EOS$_{7B}$ vs. LLaVA-v1.5$_{7B}$** | | | | | | |
| CHAIR | 0.5980 | 0.5000 | 0.0980 | 0.0550 | 0.0219 | 500 |
| MME-Hall | 0.7584 | 0.8104 | -0.0520 | -0.1062 | 0.0276 | 240 |
| AMBER-Generative (Hall. Rate) | 0.7730 | 0.6360 | 0.1370 | 0.1111 | 0.0132 | 1004 |
| AMBER-Discriminative | 0.7560 | 0.7470 | 0.0090 | 0.0019 | 0.0036 | 14216 |
| MMHal-Bench (Hall. Rate) | 0.4100 | 0.4600 | -0.0500 | -0.1484 | 0.0502 | 96 |
| Hallusion-Bench | 0.4872 | 0.4709 | 0.0163 | -0.0129 | 0.0149 | 1129 |
| **HALVA$_{7B}$ (Ours) vs. LLaVA-v1.5$_{7B}$** | | | | | | |
| CHAIR | 0.5860 | 0.5000 | 0.0860 | 0.0428 | 0.0220 | 500 |
| MME-Hall | 0.8313 | 0.8104 | 0.0209 | -0.0265 | 0.0242 | 240 |
| AMBER-Generative (Hall. Rate) | 0.6780 | 0.6360 | 0.0420 | 0.0131 | 0.0147 | 1004 |
| AMBER-Discriminative | 0.8340 | 0.7470 | 0.087 | 0.0809 | 0.0031 | 14216 |
| MMHal-Bench (Hall. Rate) | 0.4600 | 0.4600 | 0.0000 | -0.0997 | 0.0509 | 96 |
| Hallusion-Bench | 0.4895 | 0.4709 | 0.0186 | -0.0106 | 0.0149 | 1129 |
| **HALVA$_{13B}$ (Ours) vs. LLaVA-v1.5$_{13B}$** | | | | | | |
| CHAIR | 0.5460 | 0.5280 | 0.0180 | -0.0256 | 0.0223 | 500 |
| MME-Hall | 0.8438 | 0.8041 | 0.0396 | -0.0063 | 0.0234 | 240 |
| AMBER-Generative (Hall. Rate) | 0.6960 | 0.695 | 0.0010 | -0.0275 | 0.0145 | 1004 |
| AMBER-Discriminative | 0.8650 | 0.7310 | 0.1340 | 0.1284 | 0.0029 | 14216 |
| MMHal-Bench (Hall. Rate) | 0.5500 | 0.5000 | 0.0500 | -0.0495 | 0.0508 | 96 |
| Hallusion-Bench | 0.4910 | 0.4650 | 0.0260 | -0.0032 | 0.0149 | 1129 |
| **HALVA$_{13B/384}$ (Ours) vs. VILA-v1.5$_{13B/384}$** | | | | | | |
| CHAIR | 0.7000 | 0.6700 | 0.0300 | -0.0102 | 0.0205 | 500 |
| MME-Hall | 0.8646 | 0.8604 | 0.0043 | -0.0390 | 0.0221 | 240 |
| AMBER-Generative (Hall. Rate) | 0.4580 | 0.4390 | 0.0190 | -0.0118 | 0.0157 | 1004 |
| AMBER-Discriminative | 0.8790 | 0.8220 | 0.0570 | 0.0516 | 0.0027 | 14216 |
| MMHal-Bench (Hall. Rate) | 0.5500 | 0.5400 | 0.0100 | -0.0895 | 0.0508 | 96 |
| Hallusion-Bench | 0.5660 | 0.5539 | 0.0121 | -0.0168 | 0.0148 | 1129 |

Table S14: Analysing if base models are better than the hallucination mitigation methods on hallucination tasks. Here Model 1 is a base model and Model 2 is a hallucination mitigation method. Red: performance drop with statistical significance.

| Evaluation benchmark | Model 1 | Model 2 | Δ | Adjusted Δ | Standard Error | # samples |
|---|---|---|---|---|---|---|
| **LLaVA-v1.5$_{7B}$ vs. HA-DPO$_{7B}$** | | | | | | |
| CHAIR | 0.5000 | 0.6180 | -0.1180 | -0.1618 | 0.0224 | 500 |
| MME-Hall | 0.8104 | 0.7729 | 0.0375 | -0.0121 | 0.0253 | 240 |
| AMBER-Generative (Hall. Rate) | 0.6360 | 0.6910 | -0.0550 | -0.0848 | 0.0152 | 1004 |
| AMBER-Discriminative | 0.7470 | 0.7810 | -0.0340 | -0.0411 | 0.0036 | 14216 |
| MMHal-Bench (Hall. Rate) | 0.4600 | 0.400 | 0.0600 | -0.0397 | 0.0509 | 96 |
| Hallusion-Bench | 0.4709 | 0.4836 | -0.0127 | -0.0418 | 0.0149 | 1129 |
| **LLaVA-v1.5$_{7B}$ vs. EOS$_{7B}$** | | | | | | |
| CHAIR | 0.5000 | 0.5980 | -0.0980 | -0.1418 | 0.0224 | 500 |
| MME-Hall | 0.8104 | 0.7584 | 0.0520 | 0.0024 | 0.0253 | 240 |
| AMBER-Generative (Hall. Rate) | 0.6360 | 0.7730 | -0.1370 | -0.1668 | 0.0152 | 1004 |
| AMBER-Discriminative | 0.7470 | 0.7560 | -0.0090 | -0.0161 | 0.0036 | 14216 |
| MMHal-Bench (Hall. Rate) | 0.4600 | 0.4100 | 0.0500 | -0.0497 | 0.0509 | 96 |
| Hallusion-Bench | 0.4709 | 0.4872 | -0.0163 | -0.0454 | 0.0149 | 1129 |
| **LLaVA-v1.5$_{7B}$ vs. HALVA$_{7B}$ (Ours)** | | | | | | |
| CHAIR | 0.5000 | 0.5860 | -0.0860 | -0.1298 | 0.0224 | 500 |
| MME-Hall | 0.8104 | 0.8313 | -0.0209 | -0.0705 | 0.0253 | 240 |
| AMBER-Generative (Hall. Rate) | 0.6360 | 0.6780 | -0.0420 | -0.0718 | 0.0152 | 1004 |
| AMBER-Discriminative | 0.7470 | 0.8340 | -0.0870 | -0.0941 | 0.0036 | 14216 |
| MMHal-Bench (Hall. Rate) | 0.4600 | 0.4600 | 0.0000 | -0.0997 | 0.0509 | 96 |
| Hallusion-Bench | 0.4709 | 0.4895 | -0.0186 | -0.0477 | 0.0149 | 1129 |
| **LLaVA-v1.5$_{13B}$ vs. HALVA$_{13B}$ (Ours)** | | | | | | |
| CHAIR | 0.5280 | 0.5460 | -0.0180 | -0.0618 | 0.0223 | 500 |
| MME-Hall | 0.8041 | 0.8438 | -0.0396 | -0.0898 | 0.0256 | 240 |
| AMBER-Generative (Hall. Rate) | 0.6950 | 0.6960 | -0.0010 | -0.0295 | 0.0145 | 1004 |
| AMBER-Discriminative | 0.7310 | 0.8650 | -0.1340 | -0.1413 | 0.0037 | 14216 |
| MMHal-Bench (Hall. Rate) | 0.5000 | 0.5500 | -0.0500 | -0.1500 | 0.0510 | 96 |
| Hallusion-Bench | 0.4650 | 0.4910 | -0.0260 | -0.0551 | 0.0148 | 1129 |
| **VILA-v1.5$_{13B/384}$ vs. HALVA$_{13B/384}$ (Ours)** | | | | | | |
| CHAIR | 0.6700 | 0.7000 | -0.0300 | -0.0712 | 0.0210 | 500 |
| MME-Hall | 0.8604 | 0.8646 | -0.0043 | -0.0481 | 0.0224 | 240 |
| AMBER-Generative (Hall. Rate) | 0.4390 | 0.4580 | -0.0190 | -0.0497 | 0.0157 | 1004 |
| AMBER-Discriminative | 0.8220 | 0.8790 | -0.0570 | -0.0633 | 0.0032 | 14216 |
| MMHal-Bench (Hall. Rate) | 0.5400 | 0.5500 | -0.0100 | -0.1097 | 0.0509 | 96 |
| Hallusion-Bench | 0.5539 | 0.5660 | -0.0121 | -0.0411 | 0.0148 | 1129 |

# D    IMPLEMENTATION DETAILS

## D.1    TRAINING HYPERPARAMETERS

The details of training hyperparameters used in DPA training is presented in Table S15.

Table S15: Details of training hyperparameters used in DPA training.

|  | HALVA$_{7B}$ | HALVA$_{13B}$ | HALVA$_{13B/384}$ |
|---|---|---|---|
| Base model | LLaVA-v1.5$_{7B}$ | LLaVA-v1.5$_{13B}$ | VILA-v1.5$_{13B}$ |
| LLM | Vicuna-v1.5$_{7B}$ | Vicuna-v1.5$_{13B}$ | |
| Vision encoder | CLIP ViT-L$_{336/14}$ | | SigLIP-L-400M |
| Trainable module | LoRA in LLM and everything else is kept frozen | | |
| LoRA setup (Hu et al., 2021) | rank=128, alpha=256 | | |
| Learning rate | 5e-6 | | 2.5e-5 |
| Learning rate scheduler | Cosine | | |
| Optimizer | AdamW (Loshchilov & Hutter, 2017) | | |
| Weight decay | 0. | | |
| Warmup ratio | 0.03 | | |
| Epoch | 1 (342 steps) | | |
| Batch size per GPU | 16 | | |
| Batch size (total) | 64 | | |
| $\alpha$ (loss coefficient) | 0.4 | 0.5 | 0.2 |
| Memory optimization | Zero stage 3 (Ren et al., 2021; Rajbhandari et al., 2021) | | |
| Training time | 1.5 hrs | 3 hrs. | 3 hrs. |

## D.2    LICENSES OF EXISTING ASSETS USED

For images, we use publicly-available Visual Genome dataset (Krishna et al., 2017). This dataset can be downloaded from `https://homes.cs.washington.edu/˜ranjay/visualgenome/api.html` and is licensed under a Creative Commons Attribution 4.0 International License.

For the base MLLM, we use LLaVA-v1.5 (Liu et al., 2023c) and VILA-v1.5 (Lin et al., 2024). LLaVA-v1.5 is publicly available and its Apache license 2.0 can be found at `https://github.com/haotian-liu/LLaVA/blob/main/LICENSE`. VILA-v1.5 is publicly available and its Apache license 2.0 can be found at `https://github.com/NVlabs/VILA/blob/main/LICENSE`. The weights used in this work are available as follows:

- LLaVA-v1.5$_{7B}$: `https://huggingface.co/liuhaotian/llava-v1.5-7b`
- LLaVA-v1.5$_{13B}$: `https://huggingface.co/liuhaotian/llava-v1.5-13b`
- VILA-v1.5$_{13B}$: `https://huggingface.co/Efficient-Large-Model/VILA1.5-13b`

## D.3    GENERATIVE DATA AUGMENTATION SETUP

We present the prompt templates that are used to prepare correct and hallucinated descriptions in Figures S7, S8, and S9. The full list of instructions used in generating image descriptions is presented in Figure S10. We leverage Gemini Vision Pro (`gemini-1.0-pro-vision`) in preparing the responses. Complete examples depicting the pipeline of generating correct descriptions, closed-set hallucinated descriptions, and open-set descriptions are presented in Figures S11, S12, and S13. We present additional examples of training samples for one sentence image caption, short image description, detailed image description, and Yes-or-No questions in Figures S14, S15, S16, and S17, respectively.

```
# Input

## Image

<Image>

## Text

Here are the region descriptions of the given image.

<Region description 1>
<Region description 2>
<Region description 3>
...

The descriptions are the ground truth information for the image.

Based on the given region descriptions,
write a response for the following question.

Question:

<Instruction>

The response must be correct and has strong readability.

Do NOT add any new information or additional details.

# Output

<Correct description>
```

Figure S7: The **template** for generating the **correct** image descriptions.

```
# Input

## Text

The given text is a description of an image.
<Correct description>

Please rewrite the given text by replacing the mentioned words
with those from the given options.

Please choose the replacement that sounds the most appropriate.

Replace the word: <ground-truth object 1> - with a word from the
        given options: <list of hallucinated objects 1>
Replace the word: <ground-truth object 2> - with a word from the
        given options: <list of hallucinated objects 2>

...

The description should logically make sense, the style of the
new text should be the same as the original text, and
has strong readability.

Please make sure to NOT include the following words in the
description: <list of ground-truth objects>.

Your response should only include the new description
and nothing else.

# Output

<Hallucinated description>
```

Figure S8: The **template** for generating the **closed-set** hallucinated descriptions.

# Input

## Text

```
The given text is a description of an image.
<Correct description>

Please rewrite the given text by replacing the mentioned object
with another object of similar types or categories.
For example, an animal can be replaced with another
animal or one type of vehicle can be replaced by another type
of vehicle and so on.

The description should logically makes sense, the style of
the new text should be the same as the original text,
and has strong readability.

Your response should only include the new description and
nothing else.

The following objects need to be replaced:
<list of ground-truth objects>.
```

# Output

```
<Hallucinated description>
```

Figure S9: The **template** for generating the **open-set** hallucinated descriptions.

**# Instructions for one sentence caption:**

```
Provide a one-sentence caption for the provided image.
```

**# Instructions for short description:**

```
Describe the image concisely.
Provide a brief description of the given image.
Offer a succinct explanation of the picture presented.
Summarize the visual content of the image.
Give a short and clear explanation of the subsequent image.
Share a concise interpretation of the image provided.
Present a compact description of the photo's key features.
Relay a brief, clear account of the picture shown.
Render a clear and concise summary of the photo.
Write a terse but informative summary of the picture.
Create a compact narrative representing the image presented.
Please provide a short description of this image.
```

**# Instructions for detailed description:**

```
Provide a detailed description of the given image.
Give an elaborate explanation of the image you see.
Share a comprehensive rundown of the presented image.
Offer a thorough analysis of the image.
Explain the various aspects of the image before you.
Clarify the contents of the displayed image with great detail.
Characterize the image using a well-detailed description.
Break down the elements of the image in a detailed manner.
Walk through the important details of the image.
Portray the image with a rich, descriptive narrative.
Narrate the contents of the image with precision.
Analyze the image in a comprehensive and detailed manner.
Illustrate the image through a descriptive explanation.
Examine the image closely and share its details.
Write an exhaustive depiction of the given image.
Write a detailed description of the given image.
```

Figure S10: **Instructions** for different types of image descriptions. These instructions are directly taken from (Liu et al., 2024), we list them here for the sake of completeness.

# Input

## Image

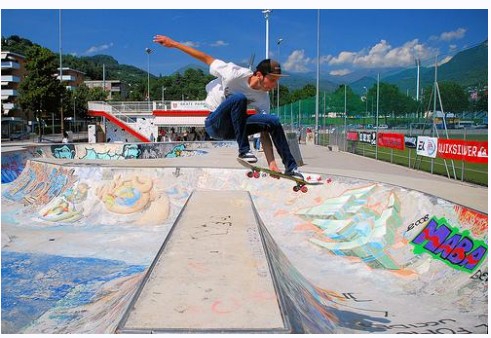

## Text

Here are the region descriptions of the given image.

a man jumping on a skateboard
a skate park with colorful graffiti
white clouds over mountains
man in white shirt and blue jeans
skating ramp with graffiti markings
Skateboarder making a jump
... 45 more

The descriptions are the ground truth information for the image.

Based on the given region descriptions,
write a response for the following question.

Question: Walk through the important details of the image.

The response must be correct and has strong readability.

Do NOT add any new information or additional details.

# Output

A young man in a white shirt and blue jeans is skateboarding in a
skate park. The skate park has colorful graffiti on the walls
and ramps. There are stairs and a fence behind the skate park.
There are also some trees and buildings in the background.
The sky is blue with some clouds.

Figure S11: A **complete example** of generating the **correct** image descriptions.

# Input

## Image

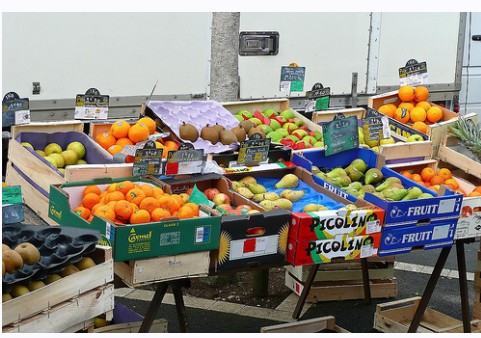

## Text

The given text is a description of an image.

Correct description:

The image shows a variety of fruits on wooden tables at a farmer's market. There are pears, apples, oranges, and pineapples. The fruits are arranged in boxes and crates. There is a price sign on some of the boxes.

Please rewrite the given text by replacing the mentioned words with those from the given options.

Please choose the replacement that sounds the most appropriate.

Replace the word: fruit – with a word from the given options:
                   plate, leaf, food, basket, vegetable

Replace the word: apple – with a word from the given options:
                   table, banana, root, bowl, shirt

Replace the word: pear – with a word from the given options:
                   tomato, gummed label, container, writing, hair

The description should logically make sense, the style of the new text should be the same as the original text, and has strong readability.

Please make sure to NOT include the following words in the description: apple, box, crate, fruit, ground, letter, line, pear, tree trunk, wall, word.

Your response should only include the new description and nothing else.

# Output

The image displays an assortment of vegetables on wooden tables at a farmer's market. There are tomatoes, bananas, oranges, and pineapples. The vegetables are organized in containers and signs. There is a price tag on some of the containers.

Figure S12: A **complete example** of generating **closed set** hallucinated descriptions is provided. The image is not fed to Gemini in generating hallucinated response, shown here for reference.

# Input

## Image

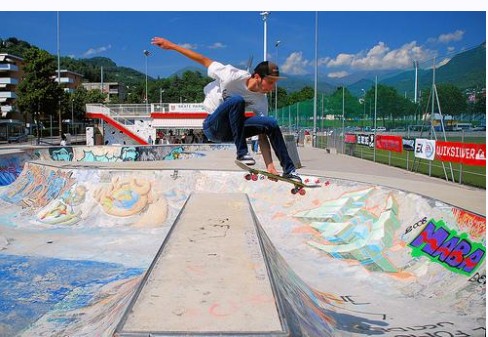

## Text

The given text is a description of an image.

Correct description:

A young man in a white shirt and blue jeans is skateboarding
in a skate park. The skate park has colorful graffiti on
the walls and ramps. There are stairs and a fence behind
the skate park. There are also some trees and buildings in
the background. The sky is blue with some clouds.

Please rewrite the given text by replacing the mentioned object
with another object of similar types or categories.
For example, an animal can be replaced with another
animal or one type of vehicle can be replaced by another type
of vehicle and so on.

The description should logically makes sense, the style of
the new text should be the same as the original text,
and has strong readability.

Your response should only include the new description and
nothing else.

The following objects need to be replaced:
advertisement, arm, baseball cap, building, cloud, design,
distance, fence, head, jean, leaf, man, mountain, park, pole,
ramp, shadow, sign, skate park, skateboard, skating, sky, stair,
t-shirt, tree, trick, wheel.

# Output

A young woman in a black dress and red sneakers is rollerblading
in a roller rink. The roller rink has vibrant murals on the walls
and floors. There are benches and a railing behind the roller rink.
There are also some plants and shops in the background. The ceiling
is white with some lights.

Figure S13: A **complete example** of generating **open-set** hallucinated descriptions is provided. The image is not fed to Gemini in generating hallucinated response, shown here for reference.

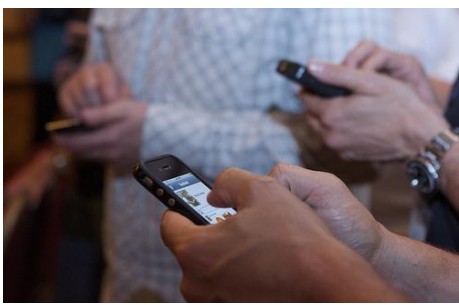

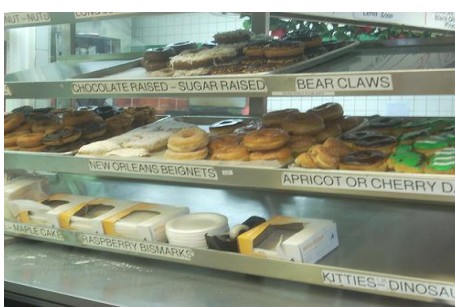

Question: Provide a one-sentence caption for the provided image.

Correct: There are three people holding and using their black smartphones.

Hallucinated: There are three people holding and using their black tablets.

Question: Provide a one-sentence caption for the provided image.

Correct: The image shows a variety of donuts on metal shelves in a donut shop.

Hallucinated: The image depicts an assortment of cupcakes on wooden shelves in a cupcake shop.

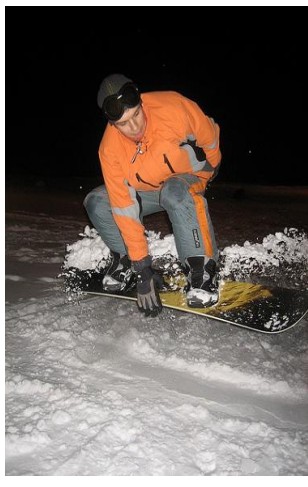

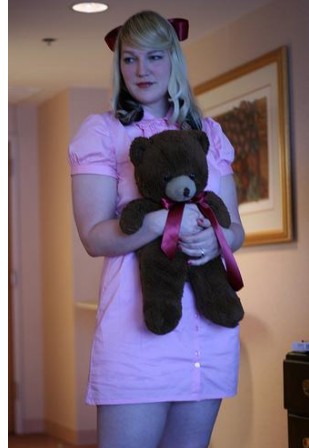

Question: Please provide a short description of this image.

Correct: A man is snowboarding down a snowy slope at night.

Hallucinated: A person is snowboarding down a snowy hill at night.

Question: Provide a one-sentence caption for the provided image.

Correct: The image shows a blonde woman wearing a pink dress with a red bow in her hair.

Hallucinated: The image displays a blonde woman wearing a pink gown with a red hat on her head.

Figure S14: Examples of **one sentence image captions** used in DPA training.

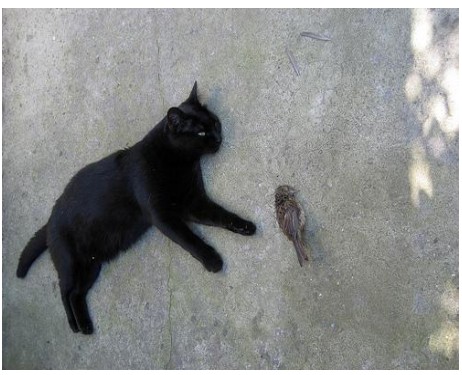

Question: Provide a brief description of the given image.

Correct: A black cat is lying on the ground next to a small brown bird. The cat has its eyes open and is looking at the bird.

Hallucinated: A black dog is lying on the grass next to a small brown leaf. The dog has its eyes open and is looking at the leaf.

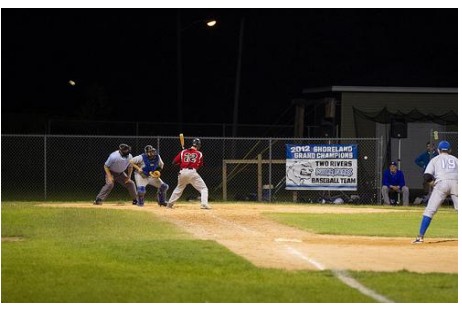

Question: Write a terse but informative summary of the picture.

Correct: The image is a night view of a baseball game. There are two baseball players, one is the batter and the other is the catcher.

Hallucinated: The image is a night view of a hockey game. There are two hockey players, one is the shooter and the other is the goalie.

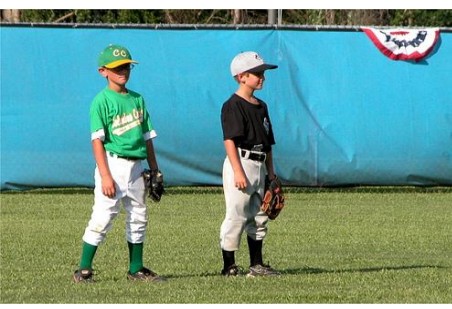

Question: Provide a brief description of the given image.

Correct: Two boys are standing in a baseball field. They are wearing baseball uniforms and holding baseball mitts. The boy on the left is wearing a green and white uniform and the boy on the right is wearing a black and white uniform.

Hallucinated: Two children are standing in a soccer field. They are wearing soccer uniforms and holding soccer balls. The child on the left is wearing a blue and white uniform and the child on the right is wearing a red and black uniform.

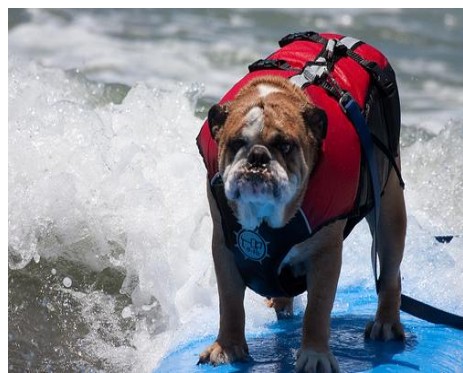

Question: Summarize the visual content of the image.

Correct: A brown and white bulldog is standing on a blue surfboard in the ocean. The bulldog is looking at the camera with an overbite. There is a big splash of water in front of the surfboard.

Hallucinated: A gray and white cat is standing on a yellow skateboard in the snow. The cat is looking at the camera with a snaggletooth. There is a big pile of snow in front of the skateboard.

Figure S15: Examples of **short image descriptions** used in DPA training.

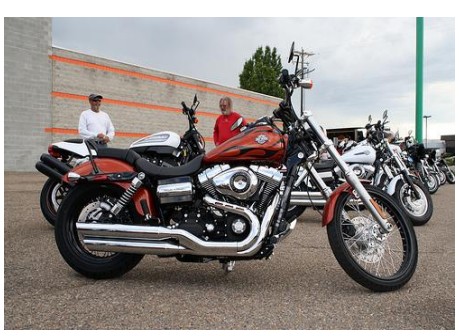 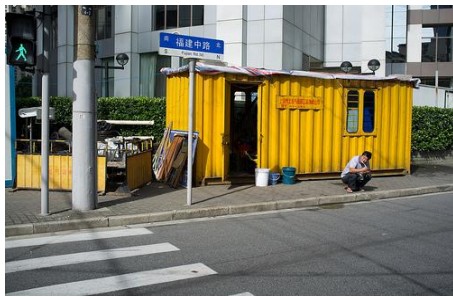

**Question**: Illustrate the image through a descriptive explanation.

**Correct**: There are a few motorcycles parked in a parking lot. There is a man standing behind one of the motorcycles. He is looking at the motorcycle. The motorcycle is orange and black. It has a chrome exhaust pipe. There are some trees and buildings in the background.

**Hallucinated**: There are a few trucks parked in a parking lot. There is a person standing behind one of the trucks. He is looking at the truck. The truck is orange and black. It has a chrome license plate. There are some plants and houses in the background.

**Question**: Clarify the contents of the displayed image with great detail.

**Correct**: A yellow container house is placed on the sidewalk. The house has a red and yellow sign on the front. There are some buckets in front of the house. A man is squatting on the sidewalk next to the house. There are green bushes and a brick sidewalk.

**Hallucinated**: A yellow trailer home is placed on the grass. The home has a blue and yellow flag on the front. There are some barrels in front of the home. A woman is kneeling on the grass next to the home. There are red flowers and a stone path.

Figure S16: Examples of **detailed image descriptions** used in DPA training.

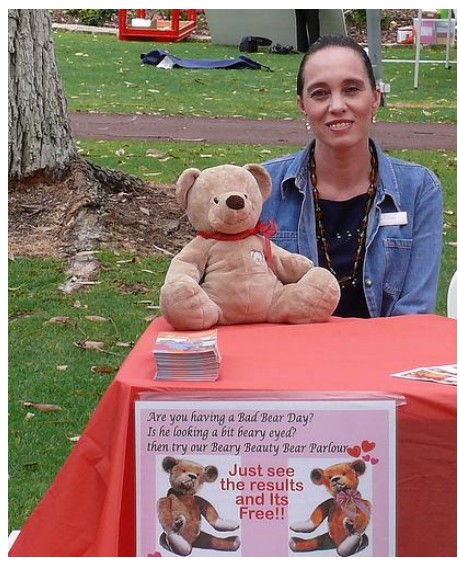

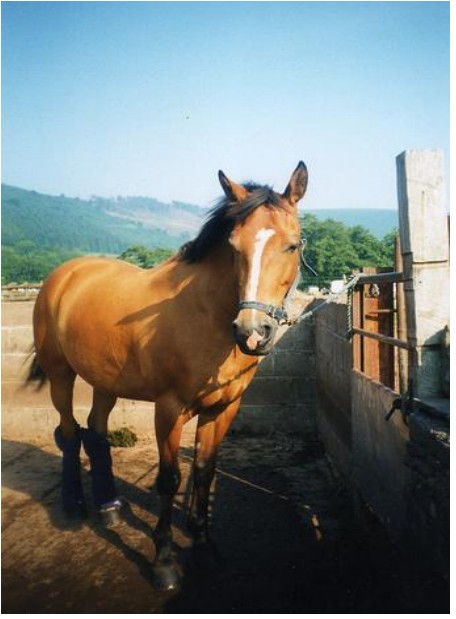

Question: Describe the image in detail.

Correct: A woman is sitting behind a table in a park. There is a sign on the table that says "Just see the results and its free". The woman is wearing a blue jean jacket and a beaded necklace. There is a stack of pamphlets on the table. The table is covered with a red tablecloth. The ground is covered with brown leaves. There is a large tree in the background.

Hallucinated: A man is sitting behind a chair in a garden. There is a poster on the chair that says "Just see the outcome and its free". The man is wearing a black leather coat and a golden chain. There is a pile of leaflets on the chair. The chair is covered with a blue sheet. The floor is covered with green grass. There is a tall building in the background.

Question: Explain the various aspects of the image before you.

Correct: This image shows a brown horse standing in a stall. The horse has a white blaze on its forehead and white socks on its back legs. The stall is made of cinder blocks and has a metal gate. There is a pile of manure in the stall. The horse is standing on dirt. There are green hills in the background.

Hallucinated: This image depicts a black cow standing in a pen. The cow has a black spot on its forehead and black socks on its front legs. The pen is made of wooden planks and has a wooden gate. There is a pile of hay in the pen. The cow is standing on straw. There are brown hills in the background.

Figure S16 (Continued): Examples of **detailed image descriptions** used in DPA training.

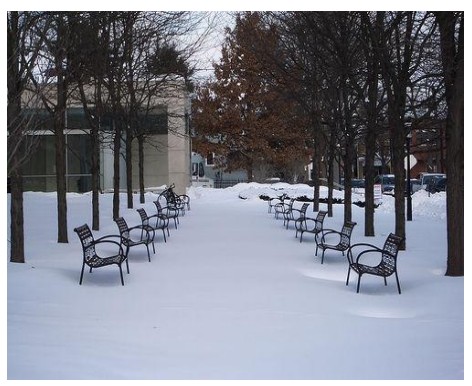

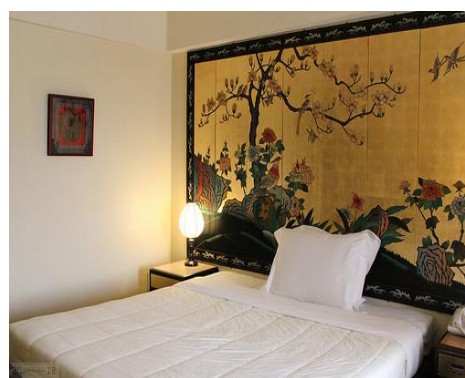

Question: Are there a total of
nine benches visible in the scene?
Please answer in one word Yes or No.

Correct: No

Hallucinated: Yes

Question: Is there a nightstand on
the left side of the bed? Please
answer in one word Yes or No.

Correct: Yes

Hallucinated: No

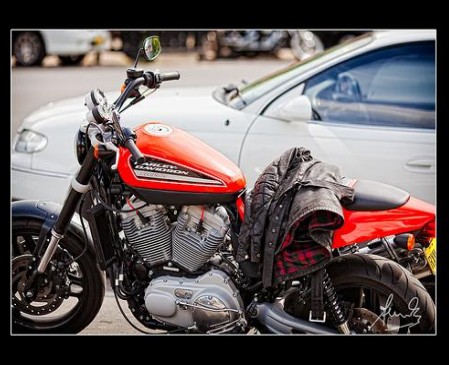

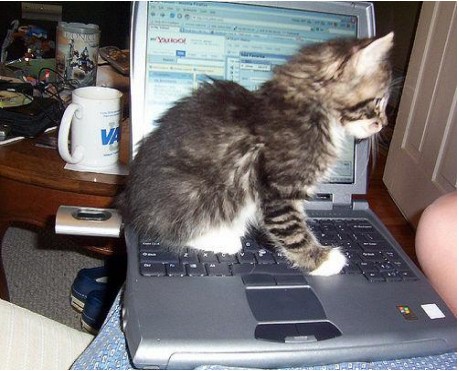

Question: Is there a person located
on the left side of the image?
Please answer in one word Yes or No.

Correct: No

Hallucinated: Yes

Question: Is the cup located on
the left side of the table? Please
answer in one word Yes or No.

Correct: Yes

Hallucinated: No

Figure S17: Examples of **Yes-or-No** questions used in DPA training.

# E  QUALITATIVE RESULTS

## E.1  QUALITATIVE COMPARISON BETWEEN HALVA AND LLAVA-V1.5

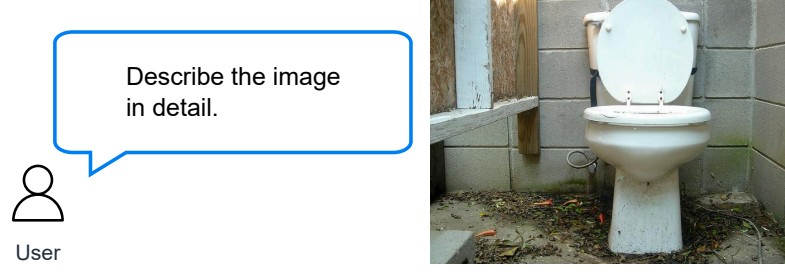

Describe the image in detail.

User

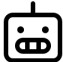

The image features a dirty, old toilet sitting in a dirt-covered area, possibly an outhouse or an abandoned building. The toilet is positioned in the center of the scene, surrounded by dirt and debris. There are several carrots scattered around the area, with some close to the toilet and others further away. The overall scene appears to be quite unclean and neglected.

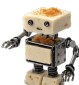

The image features a dirty, old toilet sitting in a dirt-covered room. The toilet is positioned in the middle of the room, surrounded by a pile of dirt and debris. The walls of the room are made of cinder blocks, giving the space a somewhat unfinished appearance.

Figure S18: Comparing detailed image descriptions between HALVA [🤖] and its base model LLaVA-v1.5 [🖼]. Unlike HALVA, LLaVA-v1.5 hallucinates 'carrot' in the image description. Hallucinations are highlighted in red.

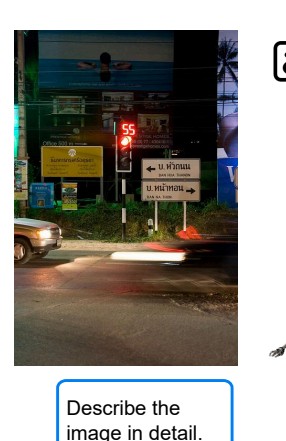

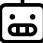

Describe the image in detail.

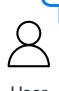

User

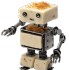

The image captures a busy street at night with a car driving down the road. The car is positioned in the middle of the scene, and its headlights are on, illuminating the street. There are several traffic lights visible in the image, with one located near the center and another towards the right side of the scene.

In addition to the car, there are two other people in the scene, one standing closer to the left side and the other near the center. A handbag can be seen on the ground, possibly belonging to one of the pedestrians. The overall atmosphere of the image is bustling and lively, with the car and people navigating the busy street at night.

The image captures a busy street at night with a car driving down the road. The street is filled with various signs, including a traffic light and a street sign, which are both lit up. The traffic light is positioned near the center of the scene, while the street sign is located on the right side.

There are also two other cars visible in the scene, one behind the main car and another further down the road. The street appears to be in a foreign country, as indicated by the presence of foreign signs.

Figure S19: Comparing detailed image descriptions between HALVA [🦸] and its base model LLaVA-v1.5 [🖼]. LLaVA-v1.5 wrongfully describes the position of the 'cars' and 'traffic lights'. Moreover, it hallucinates the presence of 'people', 'pedestrian', and 'handbag'. In contrast, HALVA correctly identifies the relative positions of the 'traffic lights' and 'cars', while accurately describing the image. Hallucinations are highlighted in red.

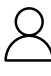

Describe the image in detail.

User

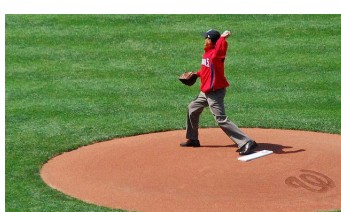

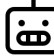

The image captures a baseball game in progress, with a pitcher standing on the pitcher's mound, preparing to throw the ball. The pitcher is wearing a red shirt and is in the process of winding up for the pitch.

In the scene, there are several other people present, including a batter, a catcher, and an umpire. The batter is holding a baseball bat, ready to swing at the incoming pitch. The catcher is positioned behind the batter, wearing a baseball glove, and the umpire is standing nearby, observing the game closely.

The baseball is visible in the air, close to the pitcher, as the pitcher prepares to release it towards the batter

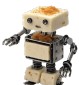

The image captures a baseball pitcher in the middle of a pitch on a baseball field. The pitcher is wearing a red shirt and is in the process of throwing the ball. He is standing on the pitcher's mound, which is a raised area in the center of the field.

The baseball can be seen in the air, close to the pitcher's hand, as he releases it. The pitcher's baseball glove is also visible, positioned near his other hand. The scene is set against a backdrop of a lush green field, showcasing the baseball diamond and the pitcher's skillful performance.

Figure S20: Comparing detailed image descriptions between HALVA [🤖] and its base model LLaVA-v1.5 [🖼]. LLaVA-v1.5 exhibits object hallucination that may happen due to object co-occurrence in the training data, such as the occurrence of 'pitcher' with 'batter', 'catcher', and 'umpire'. HALVA effectively mitigates such hallucination and accurately describes the image with necessary details. Hallucinations are highlighted in red.

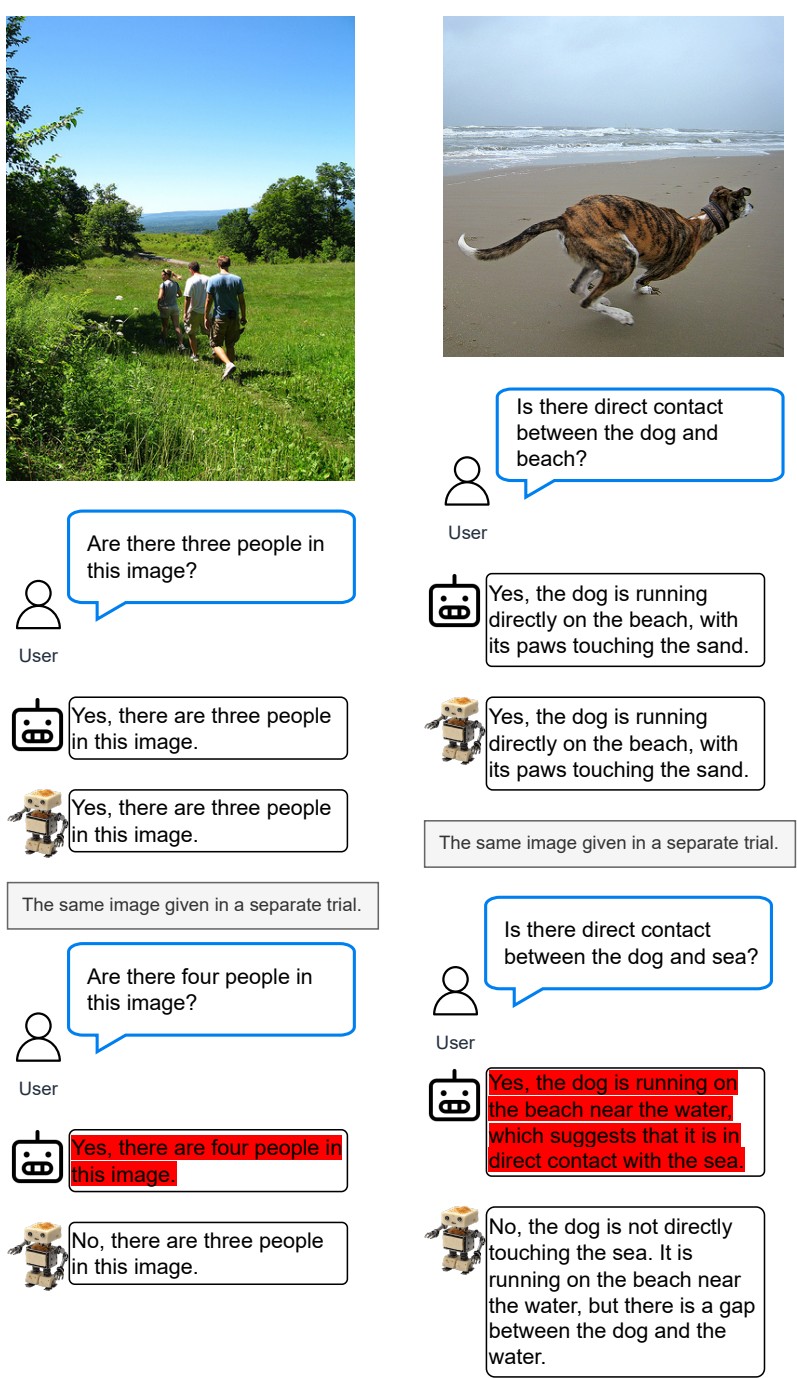

Figure S21: HALVA [🤖] does not exhibit a tendency to answer 'Yes' for any questions asked, like LLaVA-v1.5 [🖼]. Moreover, HALVA can accurately lay out the details of the image to further support the answer. As presented in the right side example, HALVA answers with "the dog is not directly touching the sea. It is running on the beach near the water, but there is a gap between the dog and the water.". Hallucinations are highlighted in red.

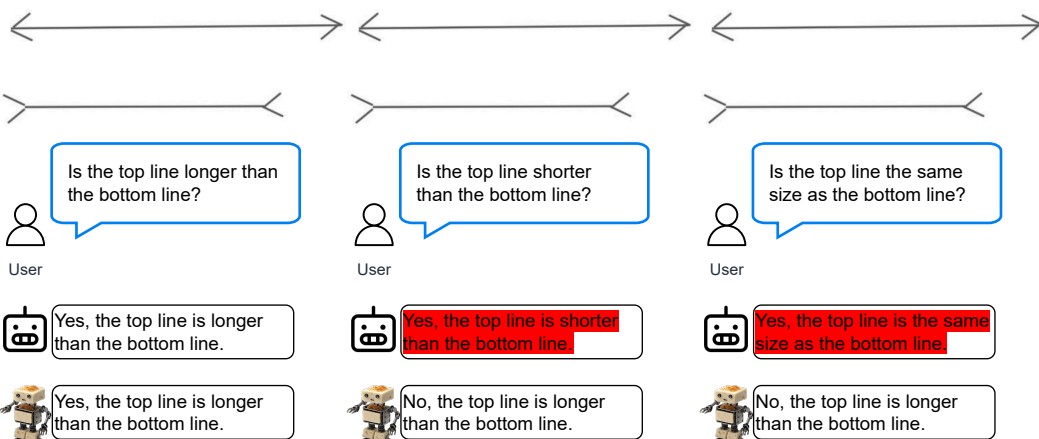

Figure S22: Examples illustrating the effectiveness of our proposed DPA in mitigating hallucinations that arise due to visual illusions. HALVA [🤖] does not exhibit a tendency to answer 'Yes' to any questions asked, like LLaVA-v1.5 [🖥]. Moreover, HALVA exhibits consistency in its responses unlike LLaVA-v1.5. Hallucinations are highlighted in red.

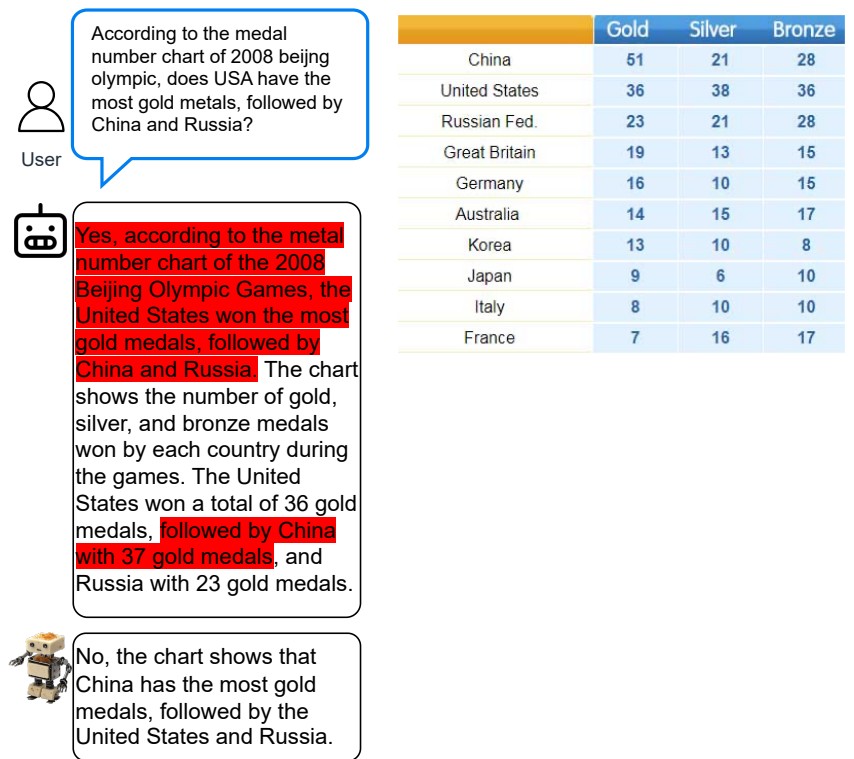

Figure S23: Examples illustrating the effectiveness of our proposed DPA in mitigating hallucinations that are beyond object centric. HALVA [🤖] accurately answers to this chart-based question unlike LLaVA-v1.5 [🖥]. Hallucinations are highlighted in red.

## E.2 QUALITATIVE COMPARISON BETWEEN HALVA AND VILA-V1.5

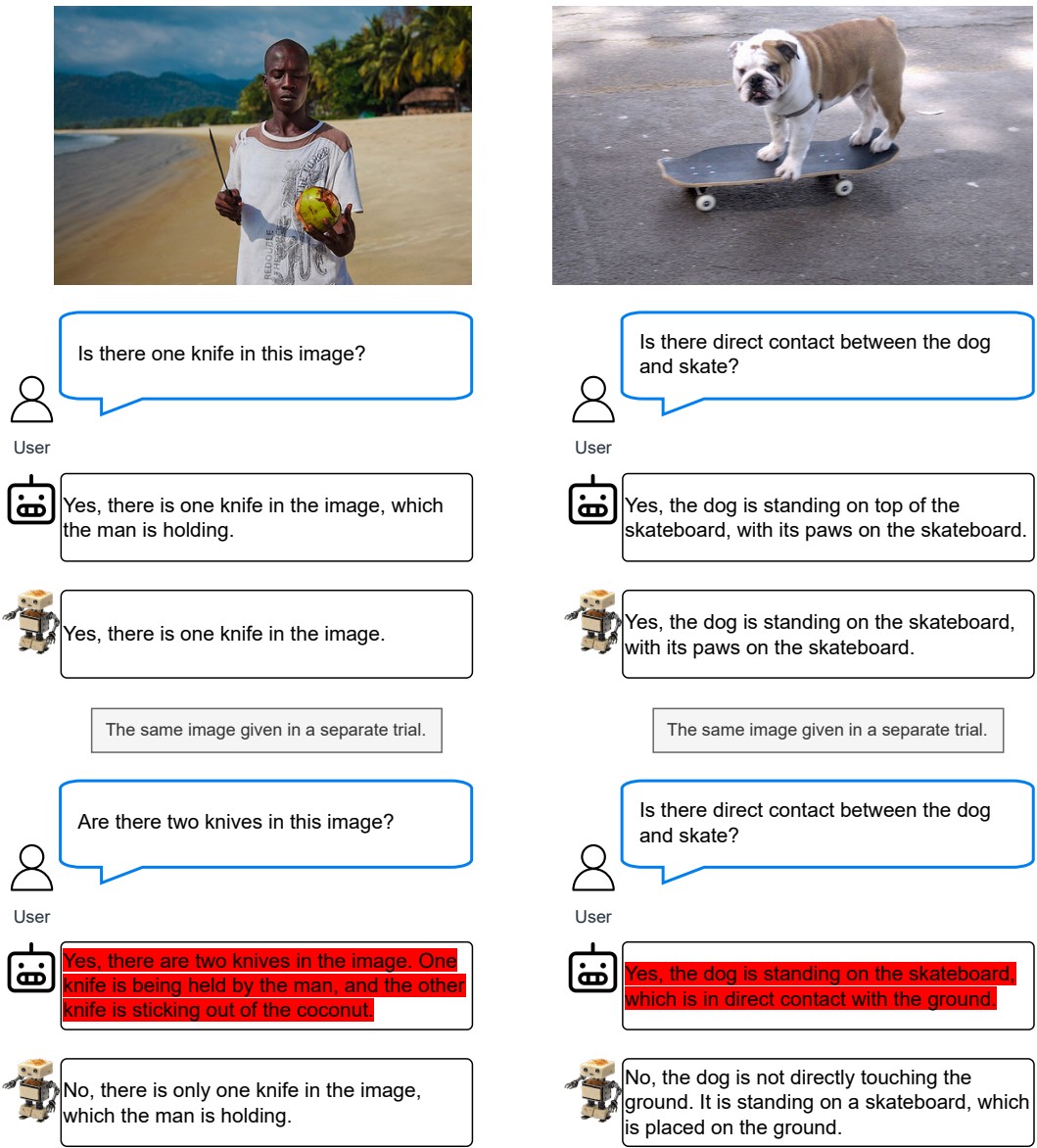

Figure S24: HALVA [🦮] does not exhibit a tendency to answer 'Yes' for any questions asked, like VILA-v1.5 [🤖]. Moreover, HALVA can accurately lay out the details of the image to further support the answer. As presented in the right side example, HALVA answers with "the dog is not directly touching the ground. It is standing on a skateboard, which is placed on the ground.". Hallucinations are highlighted in red.

## F LIMITATIONS

In this work, we focused on mitigating *object hallucinations* in MLLMs. However, MLLMs also suffer from other forms of hallucinations that may occur due to modality misalignment or over-reliance on language while ignoring other input modalities, among others. While we showed some promising results on generalization to other forms of hallucination, a rigorous exploration of those directions is left for future work. Finally, we believe our method may have applications in other areas as well. For example, it might be adapted to mitigate bias and harmful language generation, among others. We leave this exploration for future research.

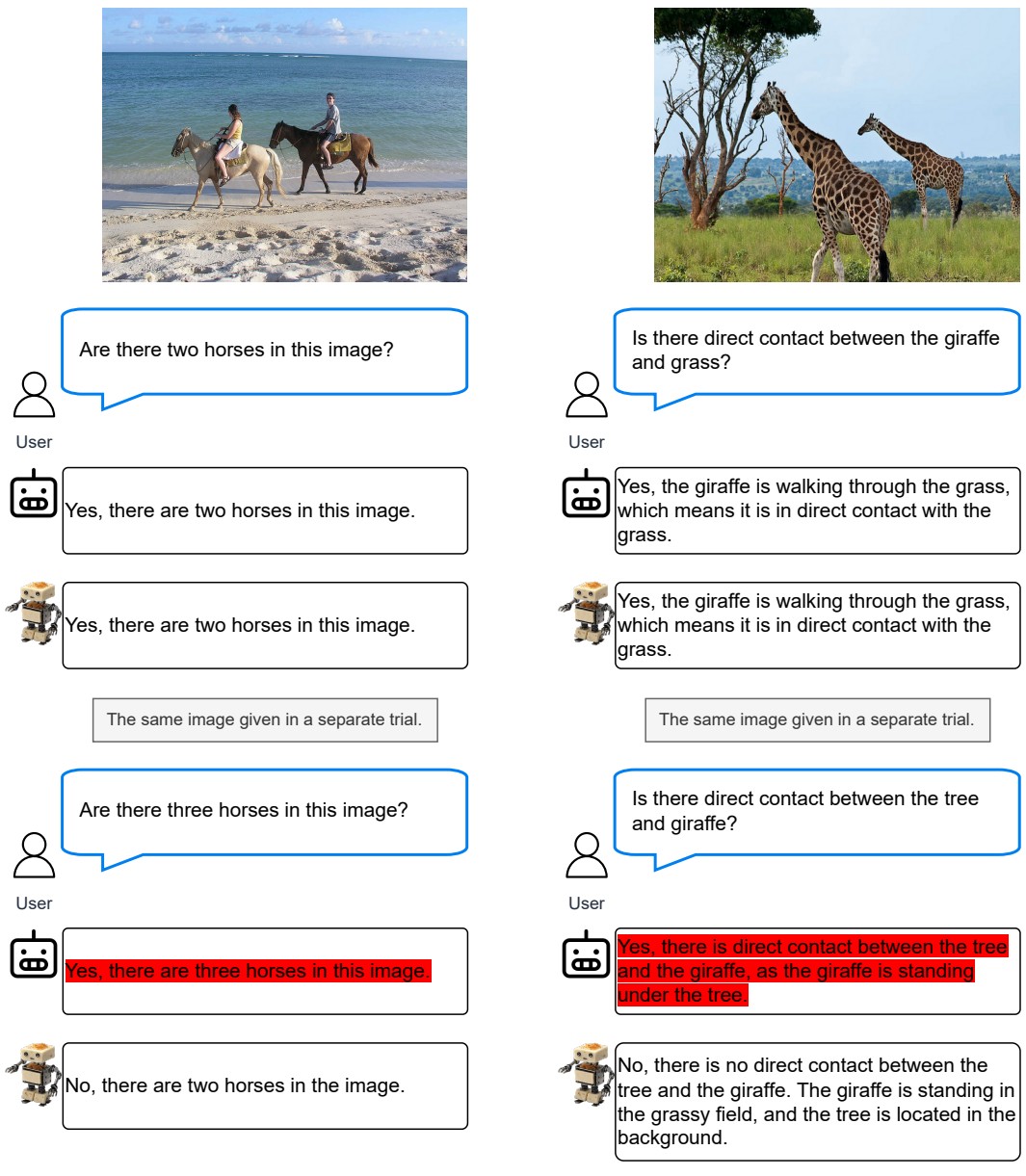

Figure S25: HALVA [🤖] does not exhibit a tendency to answer 'Yes' for any questions asked, like base model VILA-v1.5 [🤖]. Moreover, HALVA exhibit consistency in its response unlike VILA-v1.5, as shown in the left example, HALVA confirms the presence of two horses in both the time. Hallucinations are highlighted in red.

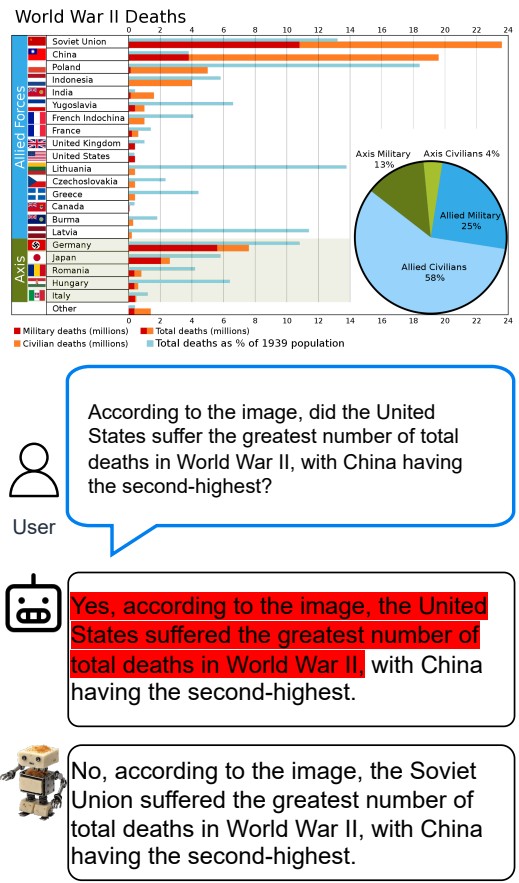

Figure S26: Examples illustrating the effectiveness of our proposed DPA in mitigating hallucinations that are beyond object centric. HALVA [🤖] accurately answers to this chart-based question unlike VILA-v1.5 [🖬]. Hallucinations are highlighted in red.

