# OpenReview forum: "Mitigating Object Hallucination in MLLMs via Data-augmented Phrase-level Alignment"
_ICLR.cc/2025/Conference — ICLR 2025 Poster_

### Official Review · Reviewer_WxnC · 2024-11-01

**Soundness:** 3
**Presentation:** 2
**Contribution:** 2
**Rating:** 6
**Confidence:** 4

**Summary:**

In this paper, the authors propose data-augmented phrase-level alignment to mitigate object hallucination in MLLMs. The method mainly involves the generation of negative hallucinated responses and phrase-level fine-tuning. The hallucination issue of models fine-tuned with the proposed method is alleviated and the general performance is maintained.

**Strengths:**

1. It is critical to study the issue of hallucination in MLLMs as well as the trade-off phenomenon in the mitigation of hallucination.
2. The proposed method is simple yet effective.
3. Extensive experiments verify the effectiveness of the method.

**Weaknesses:**

1. The motivation to mitigate hallucination at phrase level is not clearly addressed. There is no illustration on why the phrase-level loss can retain the original performance on general multimodal tasks. It seems straightforward that the constraint on the KL divergence can prevent the fine-tuned model from diverging too far. If it is the only reason, the contribution of the method is greatly weakened.

2. The explanation of Figure 2 is likely to be overclaimed. On line 260, it says "_EOS achieves a slightly lower hallucination rate_", but the figure shows that the hallucination rate of EOS is around 5.0 and that of HALVA is around 6.5. This difference is noticeable enough to me as the gap of this metric between HALVA and LLaVA-1.5 is similar. Meanwhile, Figure A right demonstrates that HALVA has a much higher F1 score on AMBER, which is natural because neither EOS nor HA-DPO is trained with Yes/No questions.

3. The presentation of results in experiments is inconsistent. The model lists are different in different tables. For instance, EOS-13B is only shown in Table 1, which makes the verification of the effectiveness less convincing.

4. There are other work mitigating hallucinaton with sub-sequence level training [1]. It is recommended to discuss the difference.

[1] Gunjal, Anish et al. “Detecting and Preventing Hallucinations in Large Vision Language Models.” AAAI Conference on Artificial Intelligence (2023).

**Questions:**

Minors:
1. The construction of dataset is not introduced with details. Details in appendix should be at least provided via cross-reference.
2. Figure 5 right is not illustrated in the main text. I find it not easy to understand the information it's trying to convey.

---

> ### Author Response · Authors · 2024-11-18
> **Rebuttal response part 1**
>
> W: Weakness, Q: Question
>
> > **W1. Motivation behind phrase-level alignment in mitigating hallucination**
>
> The motivation behind phrase-level alignment is attributed to the observation that hallucinations typically occur locally and can be pinpointed at a subsequence level, such as words or phrases. As shown in Figure 1,  “tooth-pick” is the hallucinated word in the entire response generated by LLaVA 1.5: *In the image, there are four different types of utensils:  a fork,a knife, a spoon, and a tooth-pick.* This is in contrast to other alignment problems e.g., helpfulness, where it is difficult to pinpoint if a particular word or phrase would be responsible for the overall helpfulness (or lack thereof) in a response. However, existing methods based on preference optimization techniques (e.g., RLHF, DPO) do not leverage this and instead attempt to mitigate hallucinations using a sequence-level loss. Such sequence level loss provides a coarse and noisy signal that attributes hallucinations to the entire response, making it less effective in mitigating hallucinations across diverse vision-language tasks and can also lead to a decline in general vision-language capabilities (see Figure 2).
>
> To address this we design a loss function that penalizes only the hallucinated tokens, rather than all tokens in a sequence that contains hallucinations. Our DPA loss formulation provides a fine-grained, localized signal and tends to diverge less from the model's initial state, unlike existing methods. Additionally, to further control model divergence during training, we apply a KL regularizer using a frozen reference model. Our formulation of the KL regularizer differs from that used in RLHF in two key aspects: 1\) we adopt a forward KL-divergence approach instead of the reverse KL-divergence used in RLHF, and 2\) we calculate token-wise KL divergence to better preserve the original qualities of the base model. These combined effects help in mitigating hallucinations while retaining the general capabilities of the base model.
>
> > **W2. Figure 2 and usage of Yes/No questions**
>
> **Figure 2:** Thank you for pointing this out. We removed the word “slightly” for a more accurate description. The modified line 260 now reads: “Moreover, while EOS achieves a lower hallucination rate, it degrades the detailedness of image descriptions, performing worse than the base model.”
>
> **Usage of Yes/No questions in HA-DPO and EOS:** Both HA-DPO and EOS also use Yes/No questions similar to ours. In fact, the Yes/No questions used in our work are a subset of those used in HA-DPO, as mentioned on line 220\. Below, we briefly summarize the number of Yes/No samples used in training different hallucination mitigation methods and their F1 scores on hallucination VQA tasks. As shown, despite being trained with fewer Yes/No samples (e.g., 1,510 vs. 15,893), HALVA achieves significantly better performance (e.g., 83.4 vs. 75.6) compared to the others. This demonstrates that the performance of our model is not simply attributed to using  Yes/No questions, but rather to the proposed DPA loss, especially considering the fact that HALVA uses a subset of Yes/No samples that HA-DPO is trained on.
>
> | Model | \# Yes/No samples | AMBER (F1 score $\\uparrow$) |
> | :---- | :---- | :---- |
> | LLaVA 1.5 7B (Baseline) | - | 74.7 |
> | HA-DPO 7B | 2673 | 78.1 (+3.4) |
> | EOS 7B | 15893 | 75.6 (+0.9) |
> | **HALVA 7B (Ours)** | 1510 | **83.4** (+8.7) |
>
> > **W3. EOS 13B results.**
>
> The authors of the EOS paper have only released the weights (the link to their official GitHub release: [https://github.com/yuezih/less-is-more?tab=readme-ov-file\#checkpoint](https://github.com/yuezih/less-is-more?tab=readme-ov-file#checkpoint)) of the 7B models and not the 13B one. Neither have they evaluated on such a diverse set of evaluation benchmarks as ours. For this reason, we used the EOS 7B version, which we evaluated on all benchmarks (see Tab. 1 through Tab. 6). We were able to compare our method against EOS 13B only on CHAIR in Table 1 as they have reported those numbers in their paper.
>
> > **W4. Suggested reference.**
>
> Thank you for suggesting the reference, which is indeed relevant and we have now included it in the Related work discussion in Section 5. This suggested paper shares a similar motivation as ours, i.e., fine-grained feedback in aligning multimodal LLMs. The key difference is that they focus on training a reward model to provide sub-sequence level feedback for DPO-based training to tackle hallucinations, whereas we introduce a fine-grained objective function that can be directly used to finetune multimodal LLMs for hallucination mitigation. The key differences between DPO-based training and ours are presented in Appendix B. Notably, the referenced work does not evaluate on standard hallucination benchmarks or release code or model weights, limiting our ability to conduct a direct performance comparison.

---

> ### Author Response · Authors · 2024-11-18
> **Rebuttal response part 2**
>
> > **Q1. Details of dataset construction.**
>
> Additional details on generating the training samples are provided in Appendix D.3. Below is a brief summary:
>
> - Key statistics of the training samples are presented in Table S2.
> - The prompt templates used to prepare the correct and hallucinated descriptions are shown in Figures S7-S9.
> - The full list of instructions for generating correct image descriptions is presented in Figure S10.
> - The entire set of data has been released and can be accessed at: https://anonymous.4open.science/r/HALVA/data/data.json.
> - Several examples containing both correct and hallucinated descriptions can be found in Figures S11-S17.
>
> A brief description on this can be found in page 3 line 150-153 which cross-references the supplementary material detailing the data construction process. If there is any additional information that we may have overlooked, please let us know, and we will gladly provide it.
>
> > **Q2. Figure 5 (Right).**
>
> Thank you for pointing this out. The change in alignment loss ($L\_a$) before and after training, computed over the entire training samples is presented in Figure 5 (Right). This plot further confirms that the relative likelihoods of the hallucinated concepts are indeed reduced due to DPA training. We have now added this description on line 454\.

---

> ### Author Response · Authors · 2024-11-23
> **Request for feedback**
>
> Dear Reviewer WxnC,
>
> Thank you for your thoughtful feedback on our paper. We kindly request that you confirm whether our responses have adequately addressed your questions. If there are any additional questions remaining, please do not hesitate to let us know\!
>
> Additionally, if our rebuttal has addressed your comments, we would be most grateful if you could consider updating your scores to reflect that.
>
> We sincerely value your time and consideration and look forward to your feedback.
>
> Thank you once again\!
> Authors

---

> > ### Comment · Reviewer_WxnC · 2024-11-24
> > **Response to Authors**
> >
> > Thanks for your detailed response. Additional results are provided and my major concern about the motivation of the proposed method is addressed. Therefore, I will raise my rating. Make sure you will revise the manuscript accordingly. Good luck.

---

> > > ### Author Response · Authors · 2024-11-24
> > > **Thank you for increasing the score**
> > >
> > > We greatly thank the reviewer for confirming that our responses have resolved their concerns and for increasing their score.
> > >
> > > We would like to confirm that the updated manuscript incorporates your suggested changes as follows:
> > >
> > > * The motivation behind our phrase alignment loss is added to Section 1, lines 078-087.
> > > * The description of figure 5 is added to Section 4, lines 453-455.
> > > * The suggested reference is added to Section 5, lines 519-521.
> > >
> > > We will ensure these changes are reflected in the final version as well.
> > >
> > > Thank you once again.
> > >
> > > Authors

---

### Official Review · Reviewer_mbRw · 2024-11-02

**Soundness:** 3
**Presentation:** 4
**Contribution:** 4
**Rating:** 8
**Confidence:** 4

**Summary:**

This paper introduces a novel method called Data-augmented Phrase-level Alignment (DPA) to mitigate object hallucinations in multimodal large language models (MLLMs) for vision-language tasks. DPA generates pairs of “hallucinated” and “correct” responses to fine-tune the model, reducing the generation of hallucinated phrases. A KL divergence regularization term is added to retain the model’s general capabilities. Experimental results demonstrate that models trained with DPA exhibit significant improvements in hallucination mitigation and maintain strong performance on general tasks across multiple benchmarks.

**Strengths:**

1. The writing of this paper is excellent and very detailed. The experiments are comprehensive, covering most of the popular benchmarks.

2. I really like this paper. Many works that use DPO-like methods to reduce hallucinations experience a decrease in VQA capabilities. The authors identified this issue and proposed a specialized loss to maintain the model’s performance while penalizing hallucinated phrases. Additionally, the authors validated their method separately on non-hallucination benchmarks, such as VQA-v2 and TextVQA, demonstrating its effectiveness. I believe this work makes a significant contribution to reducing hallucinations in MLLMs.

**Weaknesses:**

1. DPA relies on the quality of the generated “hallucinated-correct” response pairs. If these generated data lack accuracy or diversity, it may affect the model’s training effectiveness and generalization capability.
2. Although the experimental results demonstrate the effectiveness of DPA, the paper lacks a fine-grained analysis of hallucination types (such as objects, attributes, actions). Such analysis could provide a deeper understanding of the method’s performance across different types of hallucinations.

**Questions:**

1. VILA is also based on LLaVA-1.5 SFT. Is DPA equally effective on other architectures, such as Qwen?

---

> ### Author Response · Authors · 2024-11-18
> **Rebuttal response**
>
> W: Weakness, Q: Question
>
> > **W1. Diversity of generated response pairs**
>
> The reviewer has correctly noted that the performance of DPA depends on the quality of the generated response pairs. To obtain rich and diverse responses, we sample the hallucination concepts based on object co-occurrences in Visual Genome dataset which consists of 3.8M object instances, 34K unique object categories, and an average of 35 objects per image. Furthermore, to enhance the data diversity, we use a powerful multimodal LLM, i.e. Gemini-Vision-Pro, to generate an additional set of hallucination concepts. In total our training set consists of 28K unique pairs of correct-hallucinated phrases based on 5k unique hallucinated objects.
>
> As demonstrated through strong performance across various benchmarks, DPA is effective in addressing various forms of object hallucinations such as object existence, attributes, and relations, in both generative and discriminative tasks (Tables 1-4). Moreover, even though not explicitly trained for it, DPA can generalize to other types of unseen hallucinations that may occur due to visual illusions and complex charts among others (Table 5). These results exhibit the effectiveness and generalization capability of our method.
>
> > **W2. Fine-grained results on different object hallucinations**
>
> The fine-grained results for different forms of object hallucinations (e.g., object existence, attributes) for each benchmark are presented in the paper. Please see Table 3 (Discriminative tasks), Table S4, and Table S5 for the fine-grained categories of AMBER, MME-Hall, and MMHal-Bench, respectively. Overall, we notice all-round improvements in different fine-grained categories of object hallucinations. While both the 13B variants exhibit more effectiveness in object existence and relation, HALVA 7B achieves performance boosts on object existence and attributes.
>
> > **Q1. DPA on Qwen-based architecture**
>
> Thank you for this question. We expect our proposed approach to be effective on other LLM architectures such as Qwen as well, as none of the design choices in our method is LLM architecture specific. Due to the limited time during the rebuttal phase, we could not conduct experiments on Qwen-based multimodal LLMs, we plan to explore this in future.

---

> ### Author Response · Authors · 2024-11-23
> **Request for feedback**
>
> Dear Reviewer mbRw,
>
> Thank you for your thoughtful feedback on our paper. We kindly request that you confirm whether our responses have adequately addressed your questions. If there are any additional questions remaining, please do not hesitate to let us know\!
>
> Additionally, if our rebuttal has addressed your comments, we would be most grateful if you could consider updating your scores to reflect that.
>
> We sincerely value your time and consideration and look forward to your feedback.
>
> Thank you once again\!
> Authors

---

> ### Author Response · Authors · 2024-11-26
>
> Dear Reviewer mbRw,
>
> Thank you for reviewing our paper and providing insightful feedback. If there are any additional questions remaining, please do not hesitate to let us know\!
>
> Thank you again for your time and effort in reviewing our work.
>
> Best regards,
> Authors

---

> > ### Comment · Reviewer_mbRw · 2024-12-02
> > **Official Comment by Reviewer mbRw**
> >
> > I greatly appreciate the author’s response, which has addressed some of my questions very well. I still believe this is an excellent paper, and I will maintain my score.

---

### Official Review · Reviewer_EoXH · 2024-11-02

**Soundness:** 2
**Presentation:** 3
**Contribution:** 2
**Rating:** 6
**Confidence:** 5

**Summary:**

This paper presents a Data-Augmented Phrase-Level Alignment (DPA) approach aimed at reducing object hallucinations in Multimodal Large Language Models (MLLMs). The method centers on generating paired “correct” and “hallucinated” responses using data augmentation, which are then used to train a phrase-level alignment loss that reduces the probability of hallucinated tokens. The authors strive to maintain the model’s overall vision-language capabilities while minimizing hallucinations. Experimental results across multiple benchmarks indicate that DPA effectively mitigates hallucinations and may even improve detailed object coverage in generated descriptions.

**Strengths:**

1. This paper proposes an effective alignment method that successfully mitigates object hallucinations, showing improved scores across hallucination benchmarks in both discriminative and generative tasks.
2. DPA reduces object hallucinations without impacting the overall performance on VQA tasks, achieving comparable or even higher scores on VQA benchmarks.
3. The paper provides a variety of quantitative results across multiple benchmarks, including both generative and discriminative tasks. This breadth of evaluation provides some evidence of the DPA’s effectiveness in certain contexts.

**Weaknesses:**

1. The DPA approach offers limited novelty beyond existing finetuning and data augmentation techniques. While phrase-level alignment is applied in a new way here, it basicly builds on existing concepts and does not significantly advance the field of hallucination mitigation research.
2. Although the results include some competitive baselines, such as HA-DPO and EOS, several relevant and recent methods are omitted. A more comprehensive comparison would strengthen the evaluation. Additionally, some detailed results for LLaVA-13B and VILA are missing, and the selection of methods across different benchmarks lacks consistency.
3. While the paper asserts that DPA preserves general vision-language capabilities, the supporting evidence is limited. A broader evaluation across diverse benchmarks would help determine whether this approach impacts overall performance.
4. The authors highlight the limitations of existing methods, noting they “require massive training data.” However, the proposed DPA also introduces additional training requirements, which suggests a tradeoff between efficiency and effectiveness.

**Questions:**

1. How does DPA perform on other types of hallucinations beyond object hallucination, such as attribute or location hallucinations?
2. The reported results for some baseline methods, such as VCD, differ from those in the original papers. Did you directly test VCD, or were the results extracted from their papers? If the latter, the comparison may not be entirely fair, as it mixes experimental results with reported findings from other sources.
3. This paper augments the training data with hallucinated responses by substituting terms in both open-set and closed-set cases. However, simple substitution could potentially impact fluency and grammatical accuracy. Could this approach compromise data quality and, in turn, affect model performance?

---

> ### Author Response · Authors · 2024-11-18
> **Rebuttal response part 1**
>
> W: Weakness, Q: Question
>
> > **W1. Significance and novelty of DPA in advancing the field of hallucination mitigation research**
>
> The concept of maximizing the probability of a preferred sample through pairwise comparisons was introduced in the Bradley-Terry model \[R1\], which is a common concept across various preference optimization methods (e.g., RLHF, DPO), as well as in our DPA. However, the novelty of DPA lies in its loss formulation. The DPA loss is designed with the understanding that hallucinations typically occur locally and can be pinpointed at the subsequence level, such as words or phrases unlike other alignment problems e.g., helpfulness. For example,  it is difficult to pinpoint the lack of helpfulness in a response to a particular word, whereas hallucinated objects can be directly identified. As shown in Figure 1,  “tooth-pick” is the hallucinated word in the entire response generated by LLaVA 1.5: *In the image, there are four different types of utensils:  a fork, a knife, a spoon, and a tooth-pick.* However, existing methods do not leverage this fact and instead attempt to mitigate hallucinations using sequence-level loss the same way every other reward (such as helpfulness) is handled. In contrast, DPA introduces a fine-grained mechanism to mitigate hallucinations that allows to tackle hallucinations while not hurting the general capabilities of the model due to the localized nature of the training. An in-depth discussion highlighting the key differences between existing fine-tuning methods and ours is provided in Appendix B.
>
> As demonstrated through strong performance across various benchmarks, DPA is effective in addressing various forms of object hallucinations such as object existence, attributes, and relations, in both generative and discriminative tasks (see Tables 1-4). None of the existing methods exhibit such all-round improvements in various forms of object hallucinations. Moreover, even though not explicitly trained for it, DPA can generalize to other types of unseen hallucinations that may occur due to visual illusions and complex charts among others (see Table 5). Furthermore, DPA retains the general capabilities of the base model, unlike existing methods.
>
> > **W2. Additional results and comparisons**
>
> We have now added MEMVR \[R2\], AGLA \[R3\], ARA \[R4\], and CODE \[R5\] as additional baselines, in addition to previously used baselines (HA-DPO, EOS, OPERA, Woodpecker, HACL, VCD, RLHF-V, and LLaVA-RLHF). These new results are included in the updated manuscript in Tables 1, 2, and 4\. A high-level overview of new comparisons is also added below. Please note that some of these methods were released after the ICLR submission deadline (e.g., MEMVR), and many have not yet published their code or weights. Therefore, we compare them based on the reported results from common evaluation benchmarks, since evaluation protocol remains consistent. These additional comparisons further demonstrate the superior performance of our proposed DPA compared to prior and contemporary works.
>
> CHAIR:
>
> | Method | CHAIR\_I ($\\downarrow$)  | CHAIR\_S ($\\downarrow$) |
> | :---- | :---- | :---- |
> | MEMVR | 13.0 | 46.6 |
> | AGLA | 14.1 | 43.0 |
> | **HALVA (Ours)** | **11.7** | **41.4** |
>
> MME-Hall:
>
> | Method | Score ($\\uparrow$) |
> | :---- | :---- |
> | MEMVR | 648.3 |
> | ARA | 648.3 |
> | AGLA | 640.0 |
> | **HALVA (Ours)** | **665.0** |
>
> MMHal Bench:
>
> | Method | Score ($\\uparrow$) | Hallucination Rate ($\\downarrow$) |
> | :---- | :---- | :---- |
> | CODE | 2.49 | 0.51 |
> | **HALVA (Ours)** | **2.58** | **0.45** |
>
> **Detailed results:** Kindly note that some of the detailed results were provided in the appendix due to limited space in the main manuscript. Detailed results corresponding to Tables 2, 5, and 6 can be found in Tables S4, S5, and S6, respectively. If you kindly point out any specific result we may have missed, we can provide them during the remainder of the rebuttal period.
>
> **Selection of methods:** Many methods do not release code and checkpoints, which prevents us from evaluating them on all the benchmarks we used unless they had already conducted those evaluations and reported the results in their paper. For example, EOS 13B is only included in Table 1, or HACL is only included in Table 5, with the numbers taken from their paper (since evaluation protocols remain consistent), and we could not include it in the other tables as their weights were not available. Since the weights of EOS 7B and HA-DPO 7B are available, we were able to compare them across all benchmarks.

---

> ### Author Response · Authors · 2024-11-18
> **Rebuttal response part 2**
>
> > **W3. Additional results on general vision-language benchmarks showing the effectiveness of DPA in preserving the general capabilities of the base model.**
>
> We have now evaluated HALVA on another popular general vision-language benchmark, *LLaVA-Bench-in-the-wild*, in addition to the 4 general vision-language benchmarks we previously evaluated: VQAv2, MM-Vet, TextVQA, and MME. As shown, DPA improves accuracy by 1.8% and 0.2% on the 7B and 13B variants, respectively. These new results are also added to Table 6 in the manuscript.
>
> | Model | LLaVA-Bench-in-the-wild ($\\uparrow$) |
> | :---- | :---- |
> | LLaVA v1.5 7B | 65.4 |
> | **HALVA 7B (Ours)** | **67.2** (+1.8) |
> | LLaVA v1.5 13B | 72.5 |
> | **HALVA 13B (Ours)** | **72.7** (+0.2) |
>
> Please note that the 5 benchmarks in Table 6 comprehensively evaluate the general vision-language capabilities of the multimodal LLMs, which are briefly summarized below. We hope this response addresses the reviewer’s concern regarding general vision-language benchmarks and would be happy to provide additional clarifications.
>
> | Benchmark | Sub-categories |
> | :---- | :---- |
> | LLaVA-Bench-in-the-wild | Conversation, reasoning, and detail image descriptions. |
> | VQAv2 | Open-ended questions about images that require an understanding of vision, language and commonsense knowledge to answer. |
> | MMVet | Recognition, knowledge, OCR, spatial awareness, detailed answer generation, and math. |
> | TextVQA | Reasoning about the text in the images |
> | MME | Coarse-grained perception such as existence, count, position, and color; Fine-grained perceptions such as poster, scene, celebrity, landmark, and artwork; OCR |
>
> > **W4. Tradeoff between efficiency and effectiveness compared to pretraining-based methods**
>
> HACL is a pretraining-based method while DPA is a finetuning method. As such, our approach allows us to directly build on off-the-shelf multimodal LLMs, which then requires an additional **5 hours** of fine-tuning/training using **21.5K** samples on a single A100 80GB GPU. However, HACL requires retraining the base model from scratch using a total of **1.2M** samples, which we estimate to take approximately **154 hours** of training on the same GPU. Moreover, HACL also requires negative responses at the pretraining stage, similar to those produced by our generative data-augmentation. The general performance of our approach vs. HACL is presented in the table below. We observe that despite significantly less training requirements, our method achieves competitive performance. We also note that the two approaches are complementary and there is generally a need for innovating new approaches for alignment in all stages of training.
>
> | Method | Training time ($\\downarrow$) | MMHal-Bench (Score  $\\uparrow$) | MME (Score $\\uparrow$) | MM-Vet (Acc. $\\uparrow$) |
> | :---- | :---- | :---- | :---- | :---- |
> | LLaVA v1.5 7B (Baseline) | \- | 2.11 | 1510.7 | 31.1 |
> | HACL 7B | 154 hrs | 2.13 | **1530.10** | 30.4 |
> | **HALVA 7B (Ours)** | **5 hrs** | **2.25** | 1527.00 | **32.1** |

---

> ### Author Response · Authors · 2024-11-18
> **Rebuttal response part 3**
>
> > **Q1. Performance of DPA on other types of hallucination beyond object hallucination:**
>
> The suggested hallucinations by the reviewer (e.g., attributes) are indeed considered as part of the broader category of object hallucination and DPA is effective in those scenarios. For instance, DPA improves F1 score on attribute hallucination mitigation by \~11% on both 7B and 13B variants of LLaVA v1.5 (Table 3 column F1\_A in the manuscript). Additionally we present a number of qualitative examples in Figures 6B, S21, S24, and S25 in the paper.
>
> To further study the impact of DPA on other forms of vision-language hallucinations, for example on  visual illusion or complex charts or tables, we evaluate DPA on HallusionBench \[R6\]. The results are presented in Tables 5 and S6, with a few qualitative examples provided in Figures 6D, S22, S23, and S26. We observe from this analysis that DPA improves the overall accuracy by up to 2.2% across all 3 multimodal LLMs, confirming the effectiveness of our approach beyond object hallucination, even though it is not explicitly trained for them.
>
> > **Q2. Source of VCD results:**
>
> We found that the value reported in the VCD paper for LLaVA 1.5 was slightly lower than what we and original LLaVA 1.5 paper had produced, which we suspect could be a minor discrepancy between the VCD paper and the original LLaVA 1.5 (and by extension, as well as ours). However, we have run the VCD model using their provided implementation and observe that in fact the results for the method are very close to those reported in the paper. Please see the table below. Accordingly, in our paper, we reported the original results for VCD.
>
> |  | MME-Hall ($\\uparrow$) |
> | :---- | :---- |
> | Reported results from VCD Paper | 604.66 |
> | Results reproduced by us | 602.22 |
>
> > **Q3. Fluency and grammatical accuracy:**
>
> This is an interesting question. The negative data augmentation could indeed lead to sequences that are less fluent and less grammatical. Note that our loss is designed to decrease the probability of those hallucinated phrases compared to the correct ones. The correct responses are written by Gemini and hence are expected to be fluent and grammatically correct. Moreover, our loss is only applied locally to the hallucinated tokens making it less likely to deteriorate the general capabilities of the model, including fluency and grammatical accuracy. Finally, we apply a KL regularizer to keep the model drifting away from the base model (e.g., LlaVA 1.5) which also helps to retain the general capabilities (including fluency and grammatical correctness).
>
> To address your question, we have now evaluated the linguistic quality of the responses using four aspects of response quality: grammatical correctness, fluency, detailedness, and choice of words. Since there is no standard benchmark for these tasks, we use 100 detailed image descriptions (a subset from the AMBER image description task) generated by LLaVA 1.5 and HALVA, with GPT-4o-mini as the judge to rate them on a scale of 0 to 10\. As shown below, overall HALVA exhibits the same performance as LLaVA 1.5.
>
> |  | Grammatical Correctness ($\\uparrow$) | Fluency ($\\uparrow$) | Detailedness ($\\uparrow$) | Choice of Words ($\\uparrow$) |
> | :---- | :---- | :---- | :---- | :---- |
> | LLaVA 1.5 7B | 9.90士0.30 | 9.64士0.52 | 8.37士0.48 | 8.93士0.26 |
> | **HALVA 7B (Ours)** | 9.99士0.10 | 9.51士0.50 | 8.35士0.48 | 8.99士0.23 |
>
>
> **References:**
>
> \[R1\] Rank Analysis of Incomplete Block Designs: I. The Method of Paired Comparisons; link: https://www.jstor.org/stable/2334029?seq=1
> \[R2\] Look Twice Before You Answer: Memory-Space Visual Retracing for Hallucination Mitigation in Multimodal Large Language Models; link: https://arxiv.org/abs/2410.03577
> \[R3\] ARA: Alleviating hallucination in large vision-language models with active retrieval augmentation; link: https://arxiv.org/abs/2408.00555
> \[R4\] AGLA: Mitigating Object Hallucinations in Large Vision-Language Models with Assembly of Global and Local Attention; link: https://arxiv.org/abs/2406.12718
> \[R5\] CODE: Contrasting Self-generated Description to Combat Hallucination in Large Multi-modal Models; link: https://arxiv.org/abs/2406.01920
> \[R6\] HallusionBench: An Advanced Diagnostic Suite for Entangled Language Hallucination and Visual Illusion in Large Vision-Language Models; link: https://arxiv.org/abs/2310.14566

---

> ### Author Response · Authors · 2024-11-23
> **Request for feedback**
>
> Dear Reviewer EoXH,
>
> Thank you for your thoughtful feedback on our paper. We kindly request that you confirm whether our responses have adequately addressed your questions. If there are any additional questions remaining, please do not hesitate to let us know\!
>
> Additionally, if our rebuttal has addressed your comments, we would be most grateful if you could consider updating your scores to reflect that.
>
> We sincerely value your time and consideration and look forward to your feedback.
>
> Thank you once again\!
> Authors

---

> ### Author Response · Authors · 2024-11-25
>
> Dear Reviewer EoXH,
>
> As the discussion period nears its end, we wanted to provide a summary of the discussion phase.
>
> In response to your questions, we have expanded our comparisons (Tables 1, 2, and 4\) with more concurrent and prior works (MEMVR, AGLA, ARA, and CODE), including papers that appear after the ICLR submission deadline. These additional comparisons further demonstrate the superior performance of our method compared to others.
>
> Moreover, we have included additional results on another general vision-language benchmark (LLaVA-Bench-in-the-wild in Table 6\) confirming the effectiveness of DPA in maintaining or improving the general capabilities. Our new analysis on linguistic quality of responses (Table S9) further confirms that our method retains fluency and grammatical accuracy of the responses, on par with the base model.
>
> Additionally, we have clarified that some of the fine-grained and detailed results are presented in the appendix due to space constraints (Tables S4, S5, and S6), and the effectiveness of our method beyond object hallucination is presented in Table 5, among others.
>
> We have also added a new discussion to the Introduction highlighting the significance and novelty of our phrase-level alignment loss. Simply put, hallucinations are typically localized to specific words or phrases, however, existing methods rely on sequence-level loss, which provides noisy signals and degrades the model's general vision-language capabilities. In contrast, phrase-level alignment loss is a fine-grained mechanism to mitigate hallucinations, enabling our method to address hallucinations without compromising the general capabilities of the model, thanks to the localized nature of the training. We are pleased to note that a similar question raised by Reviewer WxnC was well received, leading to an updated score indicating acceptance.
>
> We kindly ask if our responses have sufficiently resolved your concerns. If so, we would be grateful if you could consider updating your score to reflect this.
>
> Thank you again for your time and effort in reviewing our work.
>
> Best regards, \
> Authors

---

> ### Author Response · Authors · 2024-11-28
> **Additional results and statistical analysis**
>
> Given the extension provided for the rebuttal phase, we have now evaluated  HALVA 13B/384 on all 5 general vision-language benchmarks and performed a thorough statistical analysis of our method in both retaining the general capabilities of the base model and mitigating hallucination. In particular, we perform a one-to-one comparison between the base models and their finetuned models with hallucination mitigation methods. These statistical analyses are based on the reported results from Tables 1 through 6 from the paper, *plus the new experiment that we have now performed on general tasks using HALVA 13B/384 which is based on VILA 1.5 13B/384*.
>
> We have added an Excel file detailing this analysis in our anonymous repository (originally submitted with the paper). The file can be located in [https://anonymous.4open.science/r/HALVA](https://anonymous.4open.science/r/HALVA) under the name `HALVA-Statistical-Analysis.xlsx`. For optimal viewing, please download the file, as the Anonymous GitHub repository may not have a built-in viewer for Excel files. The direct download link is:
> [https://anonymous.4open.science/api/repo/HALVA/file/HALVA-Statistical-Analysis.xlsx?v=edb870d4\&download=true](https://anonymous.4open.science/api/repo/HALVA/file/HALVA-Statistical-Analysis.xlsx?v=edb870d4&download=true).
> We follow similar setups that have been used in prior works such as \[1, 2\] for statistical validation when comparing model performances. These findings will also be added to the final version of the paper, although we could not include them in the current PDF due to the closure of the update window.
>
> Below is a summary of our findings:
>
> **New experimental results:** We perform new experiments to evaluate the general vision-language capabilities of HALVA 13B/384 based on VILA 1.5 13B/384. These results are presented in *Sheet 1 (General vision language benchmark)*. We observe that HALVA retains the same performance as the base model on VQAv2 and MMVet and also shows improvement on MME and LLaVA-Bench-in-the-wild, while experiencing marginal drops within the confidence intervals (insignificant) on TextVQA.
>
> **Statistical analysis on general tasks:** Please see *Sheet 1 (General vision language benchmark)* of the Excel file. We observe that finetuning-based hallucination mitigation methods such as HA-DPO and EOS show statistically significant performance drops on general tasks. In contrast, our proposed DPA does not exhibit such deterioration.
>
> **Statistical analysis on hallucination tasks:** Please see *Sheet 2 (Hallucination benchmark)* of the Excel file. We observe that HALVA 7B shows statistically significant improvements in CHAIR, AMBER generative, and AMBER discriminative tasks. The same holds true for HA-DPO 7B and EOS 7B. Both of our 13B variants (HALVA 13B and HALVA 13B/384) show improvements across all setups, with the improvements on AMBER discriminative tasks being statistically significant. Unlike HALVA, existing methods, such as HA-DPO and EOS, exhibit performance deterioration in 2 out of 6 hallucination tasks compared to the base model, where the performance drop for EOS 7B on MME-Hall is statistically significant.
>
> We believe this new analysis further demonstrates the significance and contribution of our results in this area, and hope that it addresses the concern of the reviewer. If so, we would appreciate it if the reviewer would kindly consider taking this into account and updating their score.
>
> Please do not hesitate letting us know in case any questions remain.
>
> References:
>
> \[1\] Extending the WILDS Benchmark for Unsupervised Adaptation, ICLR 2022; [https://openreview.net/forum?id=z7p2V6KROOV](https://openreview.net/forum?id=z7p2V6KROOV)
> \[2\] Uncovering the Hidden Dynamics of Video Self-supervised Learning under Distribution Shifts, NeurIPS 2023; [https://openreview.net/forum?id=bKqrWLCMrX](https://openreview.net/forum?id=bKqrWLCMrX)

---

> > ### Comment · Reviewer_EoXH · 2024-11-29
> > **Thanks for your response.**
> >
> > Thank you for providing such detailed responses. I sincerely appreciate the authors' thorough and patient approach in addressing the review comments. Most of my concerns, including W3 and all Questions, have been comprehensively addressed. While I remain neutral on some responses (W1 and W4), I maintain my concern regarding W2, as the inconsistency among different benchmarks has not been fully explained.
> > Regarding the experimental results, I find the evaluation of LLaVA 1.5 7B to be solid and sufficient.
> > However, I notice two issues that warrant attention: First, the number of experimental results for LLaVA 1.5 13B, LLaVA, and VILA-v1.5 13B/384 is comparatively less comprehensive than those presented for LLaVA 1.5 7B. Second, there is inconsistency in the selection of baselines across different benchmarks.
> > To address these concerns, I strongly recommend including the complete results in the final appendix.
> > In conclusion, considering the authors' detailed responses and revisions, I am pleased to upgrade my rating to 5.

---

> ### Author Response · Authors · 2024-11-29
> **Thank you for increasing the score**
>
> Dear Reviewer EoXH,
>
> We thank you for confirming that most of your concerns have been resolved and truly appreciate increasing the score to reflect this. Below, we provide additional clarifications regarding your remaining concerns:
>
>
> > **Comment**: First, the number of experimental results for LLaVA 1.5 13B and VILA-v1.5 13B/384 is comparatively less comprehensive than those presented for LLaVA 1.5 7B.
>
> We believe the reviewer refers to the larger number of baselines included for LLaVA 1.5 7B compared to LLaVA 1.5 13B and VILA-v1.5 13B/384. This is due to several reasons:
>
> 1\. Many existing studies, such as HA-DPO, Opera, DOLA, MEMVR, AGLA, ARA, VCD, Woodpecker, and HACL, have validated their hallucination mitigation methods on LLaVA 1.5 7B but not on LLaVA 1.5 13B. It was infeasible for us to reimplement and *tune* all these methods on LLaVA 1.5 13B for a fair comparison due to practical constraints, including the unavailability of code/data and limited resources.
>
> 2\. Some prior works which have evaluated their methods on LLaVA 1.5 13B (e.g., EOS 13B) did not share their model weights, preventing us from including them on all benchmarks unless the results were reported in their papers.
>
> 3\. The weights for VILA 1.5 13B were released in July 2024\. We evaluated our method on VILA 1.5 to demonstrate its effectiveness on newer and more powerful models. At the time of this research (or prior to the ICLR submission deadline), we could not find any prior works that adopted this model for hallucination mitigation.
>
> Despite these challenges, we compared our 13B models against several baselines such as RLHF-V 13B, EOS 13B, CODE 13B, LLaVA-SFT 13B, LLaVA-RLHF 13B, MiniGPT-4 13B, InstructBLIP 13B, LLaVA 13B, SPHINX 13B, and Muffin 13B.
>
> If the reviewer’s question relates to results for any of the 13B models, we confirm that we have evaluated all three HALVA models across all **10 benchmarks** used in the paper. The corresponding results are provided in the main paper, appendix, and the updated Excel file we shared. Please let us know if we have misunderstood your question and if you have any other baselines in mind that we could compare against.
>
>
> > **Comment**: Second, there is inconsistency in the selection of baselines across different benchmarks.
>
> As noted above, our selection of baselines was determined by their availability. We strived to compare against **all available baselines** for every benchmark. However, we were unable to compare with methods that had not released checkpoint weights or reported their results (e.g., EOS 13B, HACL 7B) for specific benchmarks. Having said that, we computed results for several models, such as HA-DPO 7B, EOS 7B, LLaVA-RLHF 7B, RLHF-V 13B, and LLaVA-RLHF 13B on various benchmarks, as their weights were available. If there are any other particular models that the reviewer has in mind, please let us know, and we would be happy to include them.
>
> Lastly, we will ensure to include the new statistical analysis and new results on HALVA 13B/384 on general tasks (provided in the Excel file) in the final version of the manuscript. We believe the new analysis and other changes have greatly helped improve the paper.
>
> We kindly ask if our responses above have sufficiently addressed your concerns. If so, we would be grateful if you could reconsider updating your score to reflect this.
>
> Best regards,
> Authors

---

### Official Review · Reviewer_Qf9P · 2024-11-02

**Soundness:** 3
**Presentation:** 3
**Contribution:** 3
**Rating:** 6
**Confidence:** 4

**Summary:**

This paper proposes Data-augmented Phrase-level Alignment (DPA), a novel loss function designed to reduce object hallucinations in multimodal large language models (MLLMs) while preserving their general vision-language capabilities. By generating pairs of hallucinated and correct responses through data augmentation, DPA trains MLLMs to distinguish hallucinated phrases from correct ones. Experimental results show that MLLMs fine-tuned with DPA achieve significant improvements, reducing hallucination rates and enhancing performance on visual question-answering and image description tasks.

**Strengths:**

* The DPA loss function is an innovative approach that mitigates object hallucinations by targeting phrase-level distinctions, offering a focused solution for multimodal hallucination issues.

* The data augmentation method is straightforward yet effective, generating hallucinated-correct response pairs that enable the model to learn nuanced differences with minimal complexity.

* DPA demonstrates significant performance gains

**Weaknesses:**

The core idea of the paper is to generate correct-hallucinated data pairs through data augmentation. However, I have three questions about this process.
1. In hallucination-related datasets, object hallucination does not frequently occur, raising questions about the validity of replacing every possible object and attribute with hallucinations. Since models seldom generate such hallucinations, this augmentation strategy might introduce excessive "non-realistic" hallucination cases, leading to a mismatch between training and real-world distributions, potentially impacting the model's generalization.
2. The method’s effectiveness may be limited by the diversity of data augmentation. Since hallucinated data generation relies on a finite set of replacements, it may not fully cover the types of hallucinations that could appear in practical applications, limiting the model’s ability to handle unseen hallucinations.
3. The data augmentation strategy itself lacks independent experimental evaluation. The experiments mainly focus on improvements in model performance across different benchmarks, without assessing the augmentation strategy’s generalization effect and impact on model training stability across tasks.

**Questions:**

Please see the weakness part.

---

> ### Author Response · Authors · 2024-11-18
> **Rebuttal response**
>
> > **W1. Frequency of hallucinated concepts in hallucinated response**
>
> We thank the reviewer for their insightful question. While the example depicted in Figure 3 illustrates the replacement of every ground-truth object or attribute with a hallucinated one, this is not always the case. The average number of objects per image is 35, but we substituted a random subset of ground-truth objects in the response, resulting in an average of 8 hallucinated concepts per image as mentioned in Table S2. We replaced multiple objects in the response to improve sample efficiency. Moreover, our method is effective not only on various forms of object hallucinations (see Table 1-4) but also on other types of vision-language hallucinations (e.g., visual illusions), even though it was not explicitly trained for them (Table 5). We have also evaluated our method on non-hallucination benchmarks, and as shown in Table 6, the model retains or improves the base model’s performance, suggesting no adverse effect on the model's general capabilities due to training on DPA.
>
> > **W2. Diversity of hallucination concepts**
>
> We sample the hallucination concepts based on object co-occurrences in Visual Genome dataset which consists of 3.8M object instances, 34K unique object categories, and an average of 35 objects per image. Furthermore, to enhance the diversity of the responses, we use a powerful multimodal LLM, i.e. Gemini-Vision-Pro, to generate an additional set of hallucination concepts. In total, our training set consists of 28K unique pairs of correct-hallucinated phrases based on 5k unique hallucinated objects.
>
> As demonstrated through strong performance across various benchmarks, DPA is effective in addressing various forms of object hallucinations such as object existence, attributes, and relations, in both generative and discriminative tasks (Tables 1-4). None of the existing methods exhibit such all-round improvements in various forms of object hallucinations. Moreover, even though not explicitly trained for it, DPA can generalize to other types of unseen hallucinations that may occur due to visual illusions and complex charts among others (Table 5).
>
> > **W3. Analysis on data augmentation**
>
> **Training stability for different data augmentations:** To address the reviewer's comment we have now added the training curves for different generative data augmentations (please see the new Figure S5) which demonstrates stability during training across various data splits.
>
> **Impact of data augmentations applied independently:** In Table S1 we present an ablation study analyzing the impact of independent data augmentations.  We observe that while combining closed-set, open-set, and yes/no samples expectedly achieves an overall better performance across various benchmarks, each augmentation applied individually results in an improvement over the base model.
>
> **Effect of our generative data-augmentation on a different loss:** We study the impact of our generative data-augmentation on a loss (i.e., DPO) different from our DPA, and present the results in Table S7. The results show that while our data-augmentation exhibits some benefits even when applied on DPO, overall, the combination of our data-augmentation and our loss works better.

---

> ### Author Response · Authors · 2024-11-23
> **Request for feedback**
>
> Dear Reviewer Qf9P,
>
> Thank you for your thoughtful feedback on our paper. We kindly request that you confirm whether our responses have adequately addressed your questions. If there are any additional questions remaining, please do not hesitate to let us know\!
>
> Additionally, if our rebuttal has addressed your comments, we would be most grateful if you could consider updating your scores to reflect that.
>
> We sincerely value your time and consideration and look forward to your feedback.
>
> Thank you once again\!
> Authors

---

> ### Author Response · Authors · 2024-11-26
>
> Dear Reviewer Qf9P,
>
> Thank you for taking the time to review our paper and providing thoughtful feedback. We appreciate your comments on data augmentations and have attempted to address your concerns in our rebuttal.
>
> We kindly ask if our responses have sufficiently resolved your concerns. If so, we would be grateful if you could consider updating your scores to reflect this.
>
> Thank you again for your time and effort in reviewing our work.
>
> Best regards,
> Authors

---

### Author Response · Authors · 2024-11-18
**Overall response and summary of changes**

We sincerely thank the review committee for their time and for providing constructive feedback. We are happy to see the overall engaging comments by all the reviewers and are glad to note that the reviewers find this work a simple yet effective solution (**Qf9P**, **WxnC**) for mitigating hallucinations and recognize our work as a significant contribution in this area (**mbRw**). We have carefully addressed all the comments in the individual response section. Below, we provide a brief overview of the changes made in the manuscript in response to the reviewer comments.

**Motivation behind phrase-level loss**: The motivation behind phrase-level alignment is attributed to the observation that hallucinations typically occur locally and can be pinpointed to words or phrases. However, existing methods based on preference optimization techniques (e.g., RLHF, DPO) do not leverage this and instead attempt to mitigate hallucinations using a sequence-level loss. Sequence-level losses provide coarse and noisy signals, making them less effective in mitigating hallucinations, and can also lead to a decline in general vision-language capabilities. Please see page 2 in the Introduction for changes in describing the motivation behind our work.

**Additional comparisons**: We have included more comparisons with recent methods in Tables 1, 2, and 4, some of which appeared after the ICLR submission deadline. These additional comparisons further demonstrate the superior performance of our proposed method compared to prior and contemporary works.

**Additional results on general vision-language benchmarks**: We have now evaluated HALVA on another general vision-language benchmark, LLaVA-Bench-in-the-wild. The results are added to Section 4.3, Table 6\. HALVA improves accuracy by 1.8% and 0.2% on the 7B and 13B variants, showing effectiveness of our method in preserving the general capabilities of the base model.

**Training stability for different data augmentations**: In addition to the quantitative results presented earlier, we have now added the training curves for different generative data augmentations in Figure S5, which demonstrate stability during training across various data splits.

**Fluency and grammatical accuracy**: We have now evaluated the linguistic quality of the responses on four aspects of response quality: grammatical correctness, fluency, detailedness, and choice of words. HALVA exhibits the same performance as the base model, LLaVA 1.5, confirming that there is no deterioration in language generation due to DPA training. Please see Appendix C.9 for changes.

**Others**: A discussion on Figure 5 (Right) is added to Section 4.4 and mentions the source of the VCD results in Table 4. The suggested reference and related discussions are added in Section 5.

---

### Author Response · Authors · 2024-12-03
**Summary of rebuttal phase discussions**

Dear Review Committee,

As the discussion period nears its end, we would like to provide a summary of the discussion phase.

We would like to express our gratitude to all reviewers for their thoughtful and thorough reviews, and for the opportunity to engage with them to provide clarifications and further evidence. We are pleased to see the reviewers' engagement during this period and thank them for confirming that our rebuttal has resolved their concerns. We also thank the reviewers for their encouraging comments and support, and appreciate the score increase by Reviewers EoXH and WxnC. Reviewers Qf9P and WxnC found our method to be an effective solution, Reviewer EoXH recognized that HALVA 7B experimental results are solid, and Reviewer mbRw acknowledged our work as a significant contribution in this area.

The following changes are done during the rebuttal phase:

- We updated the Introduction (lines 078-087) to include additional discussions on the motivation behind the phrase-level alignment loss, which has been well received by both Reviewers EoXH and WxnC.
- We added new results on another general vision-language benchmark (LLaVA-Bench-in-the-wild), see Table 6\.
- We included results of HALVA 13B/384 on all general vision-language tasks and conducted a statistical analysis on all three HALVA models across all evaluation benchmarks used in our work. These results are shared through our anonymous GitHub repository as the PDF update window was closed and will be added to the final version. These results further demonstrate that the performance drops observed in other finetuning-based hallucination mitigation methods are statistically significant, while our method does not exhibit such deterioration.
- We performed additional experiments showcasing that our method retains fluency and grammatical accuracy of the responses, on par with the base model, see Appendix C.9.
- Finally, we increased the number of hallucination mitigation baselines for comparison, further demonstrating the superior performance of our proposed method compared to prior and concurrent works, see Tables 1, 2, and 4\.

Thank you once again for reviewing our work and we believe these changes indeed helped us to improve the paper.

Best regards,

Authors

---

### Meta-Review · Area_Chair_Qq3v · 2024-12-20

**Metareview:**

This work investigated the problem of object hallucination of multimodal large language models to mitigate hallucinations, while preserving
their general vision-language capabilities. This work further proposed a phrase-level alignment method to reduce the likelihood of hallucinated tokens. Extensive experiments conducted on multiple benchmarks verified that the proposed method can reduce hallucination while maintaining high accuracy. The final ratings are 8, 6, 6, 6, which are all positive.

Strengths: (1) The research problem of mitigating object hallucinations of MLLMs is important. (2) The proposed phrase-level alignment is straightforward and effective. (3) The experiments on base models and benchmarks are comprehensive to validate the effectiveness of the proposed method. (4) The overall writing quality is great.

Weaknesses: (1) The motivation of phrase-level loss was not introduced very clearly. The authors provided further explanations to make the motivation clearer. (2) Experiments with some related methods and benchmarks were missing, which were added during rebuttal and discussion.

Most important reasons for accept:

**Additional Comments On Reviewer Discussion:**

The major concerns are on unclear motivation and missing experiments. After rebuttal and discussion, the concerns on motivation and further experiments are well addressed, and all the reviewers gave positive ratings as final scores.

---

### Decision · Program_Chairs · 2025-01-22

Accept (Poster)